# Application of sequential cyclic compression on cancer cells in a flexible microdevice

**Sevgi Onal**[1,2]*, **Maan M. Alkaisi**[1,2], **Volker Nock**[1,2,3]*

**1** Electrical and Computer Engineering, University of Canterbury, Christchurch, New Zealand, **2** MacDiarmid Institute for Advanced Materials and Nanotechnology, Wellington, New Zealand, **3** Biomolecular Interaction Centre, University of Canterbury, Christchurch, New Zealand

* sevgi.onal@pg.canterbury.ac.nz (SO); volker.nock@canterbury.ac.nz (VN)

## Abstract

Mechanical forces shape physiological structure and function within cell and tissue microenvironments, during which cells strive to restore their shape or develop an adaptive mechanism to maintain cell integrity depending on strength and type of the mechanical loading. While some cells are shown to experience permanent plastic deformation after a repetitive mechanical tensile loading and unloading, the impact of such repetitive compression on deformation of cells is yet to be understood. As such, the ability to apply cyclic compression is crucial for any experimental setup aimed at the study of mechanical compression taking place in cell and tissue microenvironments. Here, we demonstrate such cyclic compression using a microfluidic compression platform on live cell actin in SKOV-3 ovarian cancer cells. Live imaging of the actin cytoskeleton dynamics of the compressed cells was performed for varying pressures applied sequentially in ascending order during cell compression. Additionally, recovery of the compressed cells was investigated by capturing actin cytoskeleton and nuclei profiles of the cells at zero time and 24 h-recovery after compression in end point assays. This was performed for a range of mild pressures within the physiological range. Results showed that the phenotypical response of compressed cells during recovery after compression with 20.8 kPa differed observably from that for 15.6 kPa. This demonstrated the ability of the platform to aid in the capture of differences in cell behaviour as a result of being compressed at various pressures in physiologically relevant manner. Differences observed between compressed cells fixed at zero time or after 24 h-recovery suggest that SKOV-3 cells exhibit deformations at the time of the compression, a proposed mechanism cells use to prevent mechanical damage. Thus, biomechanical responses of SKOV-3 ovarian cancer cells to sequential cyclic compression and during recovery after compression could be revealed in a flexible microdevice. As demonstrated in this work, the observation of morphological, cytoskeletal and nuclear differences in compressed and non-compressed cells, with controlled micro-scale mechanical cell compression and recovery and using live-cell imaging, fluorescent tagging and end point assays, can give insights into the mechanics of cancer cells.

**Data Availability Statement:** All relevant data are within the paper and its Supporting information files.

**Funding:** Financial support was provided by the MacDiarmid Institute for Advanced Materials and

Nanotechnology and the Biomolecular Interaction Centre. Additional funding for V.N. was provided by Rutherford Discovery Fellowship RDF-19-UOC-019. S.O. also thank University of Canterbury for Faculty of Engineering PhD Publishing Scholarship. The funders had no role in study design, data collection and analysis, decision to publish, or preparation of the manuscript.

**Competing interests:** The authors have declared that no competing interests exist.

## Introduction

While some cells experience permanent plastic deformation after a repetitive mechanical tensile loading and unloading, the impact of such repetitive compression on deformation of cells remains largely unknown [1, 2]. As such, the ability to apply cyclic compression is crucial for any experimental setup aimed at the study of mechanical compression taking place in cell and tissue microenvironments [3–5]. For instance, in ovarian cancer, cells are exposed to chronic compressive stress from different sources such as tumor growth, displacement within stromal tissue and hydrostatic pressure out of ascitic fluid in peritoneal cavity [3, 6]. Ascites-induced compression exists in the peritoneal microenvironment due to the ascites volume that can reach >2L and increases ovarian cancer cell adhesion to peritoneum, shown by Asem *et al.* using *in vivo* assay [7]. This peritoneal microenvironment is continuously affected by the movements of the surrounding organs, musculoskeletal dynamics, and gravity [6]. Thus, ovarian cancer cells might experience cyclic compression profiles during chronic exposure to compression from different sources and due to anatomical location of the ovary. Mimicking such compression profiles and physiological pressure values in an *in vitro* dynamic compression system is necessary to further study impact of compressive stress in cancer.

Compressive stress is estimated to reach 18.9 kPa for human tumors and can exceed 20 kPa based on the experimental data from murine tumors, while those values can be even higher in situ with the impact of the surrounding extracellular matrix in tumor microenvironment [3, 8]. Compared to other cancer types such as breast and metastatic bone niches, compression studies have been limited for ovarian cancer, including for investigating the impact of various amounts of applied pressures [6]. Pressure values existing in literature that are applied for ovarian cancer have been in a range from $\sim$3 kPa to 6.5 kPa [3, 7, 9]. Ovarian cancer cell response at higher applied pressures that can occur in the body need further investigation.

In the presence of a mechanical stress, cells strive to restore their shape to maintain cell integrity after the removal of mechanical loading. However, cells may show a plastic response in the form of cytoskeletal bond ruptures and an incomplete cell shape recovery [1]. Meanwhile, cells produce an adaptive mechanism to reduce the mechanical cell stress and thus protect themselves against mechanical damage, while subsequently deforming under the mechanical load [1, 10]. Such cell response and adaptive mechanism have been shown by Bonakdar *et al.* for cyclic mechanical tensile loading [1], but comparable investigations into compressive loading remain outstanding. In particular in relation to cancer, a knowledge gap exists in regards to cell deformation and recovery in response to cyclic mechanical compressive loading on living cells.

Asem *et al.* have recently shown that human peritoneal mesothelial cells drastically change their morphology under compression, as evidenced by formation of sub-micrometer scale intercellular projections with changes in the actin cytoskeleton and a resulting more mesenchymal phenotype [7]. Depending on the compression strength and duration, such as 3 kPa for 24-hour, mesothelial cells were observed to exhibit modified nanoscale cell surface projections and to form actin-based tunneling nanotubes (TNTs) used to physically interact with ovarian cancer cells [7].

In general, cells under compression display a change in cell height and thus cell deformation in vertical direction. The nuclei region is the most protruding section of the cell *in vivo* and contributes most to the overall cell height *in vitro* for cells adhered on a substrate. Thus, when cells are compressed, the nuclei and actin cytoskeleton components supporting the cell body are highly mechanoresponsive to the applied compression. Ho *et al.* performed volumetric scanning of the single cells being flatten and observed such a decrease in cell height as per increasing pressures applied externally through a compression control valve [2].

In itself, the nucleus is a mechanosensitive organelle [11–13] that can respond to solid stress as we have recently shown by investigating nuclear deformations in ovarian cancer cells exposed to compression at increasing pressures [5]. Solid stress-induced nuclear deformations can change the activity of nuclear pore complexes and associated proteins. Such nuclear perturbations in turn modulate the nuclear import of transcription factors, which can alter gene expression and induction of DNA repair programs [14].

Morphological changes in nuclear structure, such as increased nuclear size, irregular shape by grooving, convolutions and invaginations of the nuclear envelope and altered organization by disturbed chromatin distribution, are commonly used cancer markers by pathologists [15]. Mechanical compression through confinement aid in mechanical adaptation of the cell as shown with Hela-Kyoto cancer cells in which stretching of the nuclear envelope and upregulation of actomyosin contractility were observed [16, 17]. Furthermore, irregular nuclear morphology of cancer cells aligns with altered expression of nuclear envelope proteins such as lamins A/C or lamin B in cancers as detailed by Denais *et al.*, which has the capacity to promote metastatic processes [15]. Deformation of the large and stiff nucleus is required during the passage of metastatic cancer cells through the tight interstitial space and narrow capillaries in the microenvironment. Thus, mechanical properties of the cell nucleus, such as deformability and adaptation under compression, and its connection to the cytoskeleton may play a major role in cancer metastasis [15].

To date, dynamic compression experiments in literature include frequencies of 0.1–30 Hz with an applied stress of 5.1, 9.3, 12.9 and 18.7 kPa on breast cancer cells over short durations (30–300 s), as used by Takao *et al.*, who observed a mixed mode of apoptosis and necrosis-dominant cell death in mechanically compressed monolayers in a bulk compression platform [18]. In contrast, work by Novak *et al.* used a frequency of 0.05 Hz for cyclic compression applied at 3.9–6.5 kPa for 24 hours [3]. They observed an increase in proliferation capacity and decrease in apoptosis for OVCAR3 and OVSAHO ovarian cancer cells cultured in 3D hydrogel components in a bioreactor at millimeter scales. In another example, Ho *et al.* applied cyclic compression by alternating between external pressures of 10 and 15 psi (68.9 and 103.4 kPa) at 0.5 Hz for 6 minutes [2]. Using a single cell microfluidic compression platform, their application resulted in full recovery and thus no permanent deformation in MCF10A normal breast epithelial cells after compression [2]. In our work we add to these examples by using the capabilities of our compression platform to apply cyclic compression with a low frequency in microfluidic settings. In particular, we utilize this functionality to provide a first investigation of the effects of cell compression, deformation and recovery after compression on SKOV-3 ovarian cancer cells. As part of this we provide evidence that the amount of the pressure and duration of the application clearly affects morphology during cell compression and recovery. While compression frequencies in an *in vivo* tumor microenvironment are not yet fully known, cyclic compression is important for maintenance of cell culture during mechanical compressions in an *in vitro* experimental setup, in particular as it enhances media flow underneath the micro-pistons.

Thus, with the purpose of applying compression in a cyclic, dynamic and controlled manner for the investigation of cell deformation and recovery, we have recently developed a robust cyclic compression microfluidic method based on a flexible microdevice. Fabrication and characterization of the multilayer microfluidic platform, and the observation of directional growth of cells, viability and mechanical cell lysis as end point assays under static state and pressurized states within this platform were demonstrated previously [4, 5]. The developed platform was characterized to provide extensive control of the amount, duration, and mode within a pressure profile [5].

In this paper, we further extend the reproducibility and repeatability validation of our flexible microdevice-based compression method via comparisons of experimental and computational piston actuations for independent microdevices with different membrane thicknesses. We then extend the use of the platform to the application of cyclic compressions at and beyond physiological pressure values in a sequential fashion to study dynamic biomechanical processes by recording GFP-tagged actin dynamics of live cells under compression. To the best of our knowledge, this is the first demonstration of how live cell actin behave under dynamic compression applied over time in cyclic mode and sequentially at varying pressures in ascending order. Using time-lapse imaging, we thus show a sequence of actin deformation and disruption events based on the amount of the applied pressure in a flexible microdevice. The current work also expands applicability by providing a more physiologically relevant setting in form of a hydrogel coating within micro-piston device as shown in Fig 1. We showcase data of cellular deformations and recovery during and after cyclic compression at mild (e.g. 15.6 kPa) and upper mild (e.g. 20.8 kPa) physiological pressure values, obtained via end point assays of actin and nuclei deformations in compressed cells at zero time or at 24 h-recovery after compression. These experiments further prove flexibility and novel use of our microdevice for the applications of not only cell compression but also recovery. Differences observed between

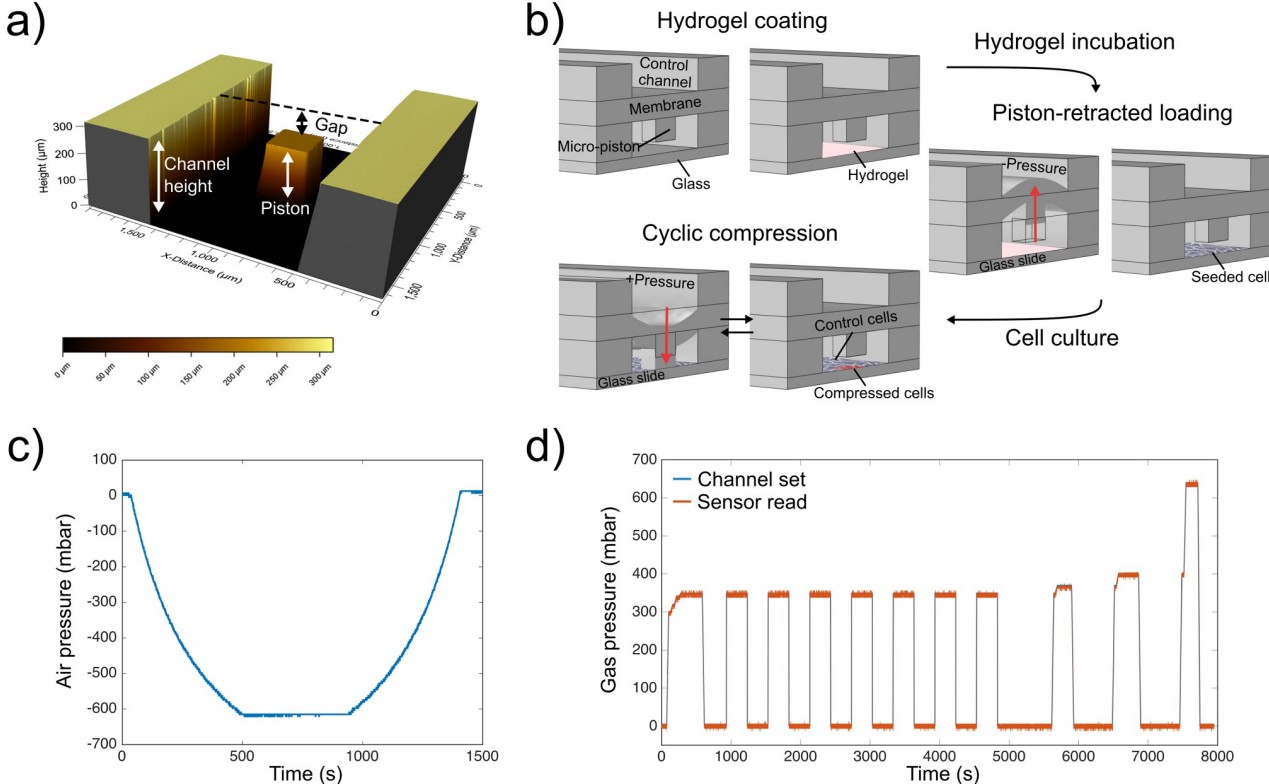

**Fig 1. Device design, operation and applied pressure profiles.** (a) 3D optical profilometer image of a micro-piston device showing the compartment including total height of the channel where the micro-piston is suspended, height of the micro-piston and the gap between the piston and surface of the channel. (b) Device operation steps from hydrogel coating, to piston-retracted loading of cells and cyclic compression on cells. The device was kept in static state during hydrogel loading and put into piston-retracted state for cell loading. In this state the membrane, and thus micro-piston, are retracted from the static position towards the top control channel by the applied negative pressure. After loading and settlement of the cells, flexible components were actuated down to their initial position. Cyclic compression of cells is illustrated by the membrane deflection and micro-piston brought onto cells by pressure applied through the control channel and retracted back after compression (repeated cyclically). Representative pressure profiles for piston-retracted loading, and sequential cyclic compression process are shown in (c) and (d), respectively.

compressed cells fixed at zero time or after 24 h-recovery suggest that SKOV-3 cells exhibit deformations at the time of the compression, a proposed mechanism cells use to prevent mechanical damage. The extent of recovery of compressed cells can give insights into the cell integrity and adaptation of cancer cells to restore their shape or acquire new one when exposed to mild or upper mild compressive stress at physiological levels. As demonstrated with SKOV-3 ovarian cancer cells, biomechanical responses of cells to sequential cyclic compression and during recovery after compression can be revealed in a flexible microdevice where cell deformation, cytoskeletal and nuclear changes and recovery in cells exposed to such compression can be readily captured.

## Materials and methods

### Device fabrication and characterization

Microdevice design, fabrication and characterization were explained in detail previously [5]. In brief, the multilayer flexible microdevice is composed of a control microchannel at the top, a polydimethylsiloxane (PDMS) membrane and monolithically attached micro-piston in the middle, suspended in a bottom microchannel enclosed by a glass substrate on which cells are cultured (Fig 1). The layers were fabricated out of PDMS via different soft-lithography methods including standard replica molding for the control layer, spin-coating for the membrane/micro-piston layer, and exclusion molding for the bottom layer. All flexible layers of the microdevice were fabricated from a 10:1 w/w mixture of PDMS (Sylgard 184, Dow) base and curing agent. Curing conditions for each layer, as well as spin-coating settings for PDMS membrane/micro-piston layer, are detailed in [5]. After the curing process was completed, individual layers were assembled via alignment marks in each layer and the aligned PDMS construct was plasma-bonded to the glass. Dimensions of the device compartments (Fig 1A) were measured with 3D optical profilometry (Profilm3D, Filmetrics) after each fabrication to ensure repeatability in predicting the external pressure amount via computational modelling. This ensured good control of piston contact pressures applied to cells on-chip during cell compression experiments.

### Computational model and experimental validation

A previously developed finite element method (FEM) based on hyperelastic Saint Venant–Kirchhoff theory for actuation of the flexible microdevice [5] was implemented in COMSOL Multiphysics (V5.5, COMSOL) and used with different membrane thicknesses obtained from experimental measurements of membranes within the fabricated devices. Vertical displacement and maximum piston contact pressure as per the applied external pressure were simulated and analysed.

### Cell culture, Matrigel coating and chip loading with cells

SKOV-3 ovarian cancer cells (kindly supplied by the Laboratory for Cell and Protein Regulation at the University of Otago, Christchurch) were cultured in Earle's salts and L-glutamine-positive MEM (Gibco) supplemented with 10% fetal bovine serum (FBS, Life Technologies), 1% of penicillin/streptomycin (Life Technologies), and 0.2% fungizone (Life Technologies) in a humidified atmosphere of 5% $CO_2$ at 37˚C. Prior to cell loading, the cell channel of the flexible microdevice was coated with 100 μg/ml Matrigel (Corning, 356237) for 60–70 minutes (Fig 1B). Due to the dilute Matrigel solution, the height of hydrogel layer formed inside the microchannel could not readily be measured, however, 100 μg/ml Matrigel was previously shown to facilitate cancer cell adhesion and mimic the extracellular matrix (ECM) microenvironment in

comparison to other substrates [19]. Matrigel is commonly used basement membrane extracellular matrix for providing improved microenvironmental and physiological conditions in *in vitro* culture assays of cancer cells by its composition [20–23]. Next, SKOV-3 cells at a seeding density of $1.5 \times 10^6$ cells/ml were loaded using piston-retracted loading. In this method of chip loading, the PDMS membrane with the attached micro-piston was first retracted out of the culture channel towards the top control channel using an applied negative pressure of 615 (-615) mbar to favour flow of seeding solution for homogenous distribution of cells. Seeded cells were cultured for 3–4 days on-chip until confluency (S1 Fig). One day before experiments, cell media in the channel was changed to $CO_2$-independent medium (Gibco) supplemented with 10% fetal bovine serum, 2% L-Glutamine (Gibco, 200 mM), 1% of penicillin/streptomycin, and 0.2% fungizone. Following this, cells were cultured overnight in the humidified atmosphere of a $CO_2$-free incubator at 37˚C until live cell compression and imaging on a custom microscope stage (LEC060, LabIVF) pre-warmed to 37˚C.

### Device operation for sequential cyclic compression

The flexible microdevices were operated with cyclic compression profiles in sequential manner. A pressure ladder with gradually increasing pressure was applied until the micro-piston reached the cells and started to apply contact pressure. The membrane, and thus micro-piston, were actuated with a pressure controller (OB1 Mk3+, Elveflow) coupled to pressure sensors (MSP4, 7 bar, Elveflow) and sensor reader (MSR, Elveflow) operated via Elveflow Smart Interface software (ESI, v3.04). This was used to set the $N_2$ external gas pressure values required at each step to apply the desired cyclic pressure profiles in a controlled and automated manner. Pressure values in this work are given in mbar for externally applied pressures, as per calibrated settings of the pressure pump, pressure sensor and sensor reader, while kPa is used for the piston contact pressures, which are based on conversion from values computed using mechanical modelling. Our piston contact pressures given in kPa were compared to the literature values on contact pressures which are also given in kPa in the associated references.

### Live imaging of GFP-tagged actin cytoskeleton of cells

The actin cytoskeleton profile was quantified by comparing cancer cells with and without applied force. Both, an end point assay and live imaging of the actin cytoskeleton of cells transduced with CellLight™ Actin-GFP, BacMam 2.0 (Invitrogen) were used to investigate the dynamic biomechanical processes in cancer cells under compression. The use of baculovirus for the Actin-GFP transduction was conducted under Environmental Protection Authority (EPA, New Zealand) approval number of GMD101847. Cells were seeded on-chip at a density of $1.5 \times 10^6$ cells/ml and cultured overnight, allowing sufficient time for cell adherence. Cells were then transduced on-chip for actin-GFP at 30 PPC (particle per cell) with CellLight™ particles. Following this, cells were incubated for an additional 3 days and the cell media containing the particles was replenished daily until the day of the compression application. The actin-GFP transduced cancer cells were dynamically stimulated and compressed with various intermediate actuation pressures in a sequential cyclic manner, namely following a sequence from Mild (15.6–15.9 kPa) to Intermediate 1 (23.8–26.8 kPa), Intermediate 2 (37.8–41.7 kPa) and Severe (127.8–140 kPa) piston contact pressures. The first two ranges of this sequence fell within physiological and upper physiological pressure values and cell viability was high (on average 94% for Mild and 77% for Intermediate 1), while Intermediate 2 and Severe levels constitute hyperphysiological values and cell viability decreased (on average 43% for Intermediate 2 and 20% for Severe) [5]. As shown by the representative sequence in Fig 1D, first, a pressure amount ladder was formed up to a pressure in the mild compression range, represented by an

externally applied pressure of 350 mbar and resulting in a piston contact pressure of 15.6 kPa, during which the piston reached the cells at the bottom surface and cells started to distinctly deform. Once this was observed, cyclic compression was applied at mild pressure for a total of 1 hour, with cells compressed for 5 min at 15.6 kPa, followed alternately by a rest for 5 min at 0 kPa. To track cell response, actin-GFP signals of the compressed cells were captured before compression, as well as during the first and last cycles of the cyclic compression stage. At the end of this sequence, the externally applied pressure was sequentially increased first to 370 mbar, then to 400 mbar and finally 640 mbar, resulting in the respective piston contact pressures of 23.8 kPa (Intermediate 1), 37.8 kPa (Intermediate 2) and 140 kPa (Severe). During this stage, cells were compressed for up to 2 min at each pressure (Fig 1D) and actin-GFP signals of the compressed cells were captured. Throughout the application of sequential cyclic compression, cells were allowed to adjust to each next higher applied pressure. As shown in the pressure amount ladder in Fig 1D, pressure was applied gradually in 30 s among rising steps of the ladder until the piston reached the focal plane of the cells and started cell deformation. After observing that cells were distinctly deformed at a mild compression level but not to a disruption or lysis level, compression was set to be in cyclic mode. Although the pressure was changing suddenly along the cycles as per the settings of the selected cyclic mode in the pressure controller, cells were not damaged when compressed as they resulted in being resistant at this physiological value. Next, pressure application was continued with gradual increase from 350 to 370 mbar, from 370 to 400 mbar and from 400 to 640 mbar in 30 s for each, as shown in Fig 1D. Thus, cells could have the time to adjust during the pressure increase and show their intrinsic response to the pressurized stage of each compression level.

## End point assay staining method

After compression experiments, cells were fixed on-chip with 4% paraformaldehyde (Alfa Aesar) for 30 min. The nuclei and actin cytoskeleton of the cells in the cell-culture channels of the micro-piston devices were stained using Hoechst 33342 (Thermo Scientific, 1 μg/ml) and CF 488 A phalloidin (Biotium, 1:40) according to the manufacturers' instructions, with on-chip dye incubation for 20 min and 1.5 h, respectively. Stained nuclei and actin cytoskeleton of the fixed cells were captured with an inverted fluorescence microscope (Nikon Eclipse Ti) equipped with Hamamatsu camera (ORCA-Flash4.0 V2) and HCImage software.

## Imaging and data analysis

Live imaging of cyclic compression was performed via time-lapse phase-contrast imaging at a rate of 1 frame per 100 ms. Fluorescence images for actin-GFP were acquired following each step of the sequential cyclic compression. Images were processed and analysed using ImageJ (Fiji) [24]. Regions of interest (ROIs) were drawn around the periphery of the micro-pistons. Corrected total cell fluorescence (CTCF) [25] was calculated as a function of integrated density, area of micro-piston ROI and mean fluorescence of background readings as follows:

$$CTCF = Integrated\ Density - \left(Area_{\text{micro-piston ROI}}\ x\ Mean_{\text{fluorescence background}}\right) \quad (1)$$

For the analysis of cell nuclear deformation, ImageJ was used to extract ROIs for the cell nuclear boundaries by applying the analyse particles function to the threshold images with Hoechst epi-fluorescence signals. ROIs obtained from the Hoechst-stained cell nuclei in this way were confirmed using corresponding phase-contrast images for validation of the nuclei segmentation. Area, circularity, and aspect ratio of the cell nuclei were measured from the ROIs. Two-hundred-fold magnification images were used during the analysis of Hoechst-

stained nuclei of control (non-compressed) cells in control regions (CRs) around the micro-piston and compressed cells under the micro-piston at zero time or at 24 h-recovery after compression.

Student's t-test was used to determine the statistical significance of live cell actin deformation across increasing pressure ranges, and nuclear deformations in compressed and recovered cells after compression. Statistical significance was taken as $p < 0.05$. Data were represented as mean ± standard error of mean (s.e.m.).

## Results and discussion

### Comparison of the experimental and computational results in micro-piston device

To extend on what is possible with existing systems and increase robustness of cyclic cell compression results, control of the pressure application was optimized by evaluating the effect of device fabrication variations. To recall dimensions of the micro-piston device we developed in [5], the height of the cell culture channel was on average 313 μm, while micro-pistons were measured as around 206 μm in PDMS replicas. Once assembled, the gap these layers created between the bottom of the micro-piston and the glass surface was measured as on average 108 μm. In the current work, the thickness of the PDMS membrane attached to micro-pistons varied between 211 μm and 258 μm. Based on the experimental data obtained from optical profilometer measurements of actuated micro-pistons (Fig 1A), COMSOL simulations were performed to predict the piston contact pressure (Fig 2) and these pistons were then used to perform cyclic cell compression. Details of the sequential cyclic compression process at mild pressures, up to cell lysis at severe pressures, could be recorded and analysed, together with cell viability [26]. Computational modelling used to calculate the maximum piston contact pressure (S2 Fig) was further validated experimentally using independent microdevices with different membrane thicknesses to enhance reproducibility and repeatability of our flexible microdevice-based compression method. In our previous work we showed that simulations were in good agreement with optical profilometer measurements of vertical displacement of the PDMS membrane and thus micro-piston at given externally applied pressures [5]. Here, we extended this by providing evidence of close correlation between the computational model and cell compression applications within independent micro-piston devices fabricated with different membrane thicknesses. In particular, when small variations resulted in spin-coated PDMS membrane thicknesses, our model was able to predict what external pressures needed to be applied through the control microchannel to obtain comparable piston contact pressures on cells, as demonstrated in Fig 2 and Table 1 for thicknesses of 246, 253 and 258 μm as measured from a PDMS membrane coated on a 4-inch photoresist master. The model could also predict the external pressure levels that needed to be applied when the same membrane thickness, such as 235 μm, was used to obtain different piston contact pressures on cells in independent microdevices (Table 1). Measurements further demonstrated that the amount of pressure required to deflect each membrane to bring the micro-piston into contact with cells was in accordance with the computationally-modelled external pressures (Fig 2A). A summary of the compression applied to cancer cells via different membrane thicknesses for externally applied pressures and the resulting maximum piston contact pressures is shown in Table 1. Fig 2B further shows that comparable piston contact pressures were achieved by mildly compressing cells with applied external pressures ($R^2 = 0.9967$), which were experimentally applied according to the pressure amounts in simulations (Fig 2A). This indicates that the current model is indeed robust across different membrane thicknesses and independent devices, and can be used to predict the piston contact pressure resulting from externally applied pressures.

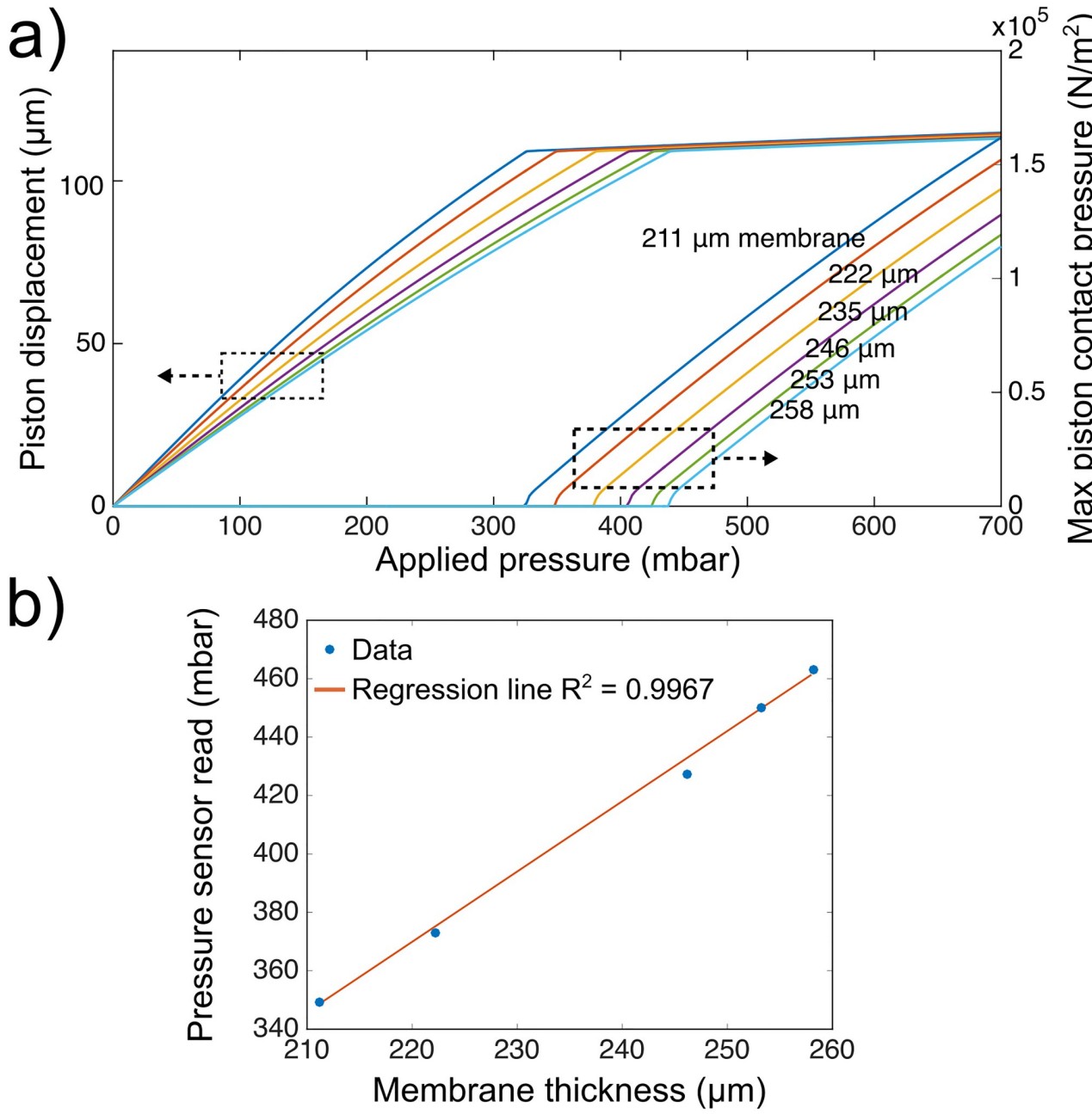

**Fig 2. Computational and experimental validation of the micro-piston contact pressures across different membrane thicknesses.** (a) Simulated piston displacement and maximum piston contact pressures at different membrane thicknesses as function of applied pressure. The left pattern of curves refers to the left y-axis (piston displacement ($\mu$m)), while the curve pattern on the right refers to the right y-axis (max piston contact pressure (N/m$^2$)), resulted at each externally applied pressure shown in x-axis. At a certain applied pressure (x-axis), micro-piston moves towards the bottom substrate and thus shows a displacement (left y-axis) depending on membrane thickness and applies a corresponding contact pressure (right y-axis). Colors of the curves refers to the membrane thicknesses indicated on the graph. (b) Correlation among independent experiments with the pressure sensor read for the externally applied pressures applied based on simulations in (a) run for the corresponding membrane thicknesses, resulting in a similar piston contact pressure.

**Table 1. Compression applied to cancer cells via fabricated different membrane thicknesses, showing externally applied pressures with sensor readings (in parentheses) from experiments and the predicted resulting internal maximum piston contact pressures.**

| Membrane thickness (μm) | Applied pressure External (mbar) | Piston contact pressure Internal (kPa) |
|---|---|---|
| 211 | 354 (350.01) | 15.6 |
| 222 | 380 (373.74) | 15.9 |
| 235 | 400 (397.47) | 12.5 |
| 235 | 410 (409.34) | 17.9 |
| 235 | 420 (415.27) | 20.7 |
| 246 | 430 (427.94) | 15.1 |
| 253 | 450 (450.87) | 15.6 |
| 258 | 465 (463.61) | 15.6 |
| 258 | 470 (468.68) | 17.9 |

While inferring these results, the cell compression process was imaged in detail in Matrigel-coated microchannels at a speed of 1 frame per 100 ms, as shown in S1 Movie. As illustrated by this time-lapse video, the platform allowed for the temporal evolution of the dynamic cell compression to be investigated in each cycle while the piston was contacting cells and applying compression. Distinct cell membrane bulges could be observed to form during such compression at piston contact pressure of 15.6 kPa and recovery of cell membranes from these bulges was visible when pistons were lifted off during rest stages. Changes in cell membrane integrity and redistribution of the actin cytoskeleton under stress could affect cell adhesion and migration, integration of cadherin adhesion and cytoskeleton at adherens junctions at cell-cell contacts [27–29] and interactions between different cell types such as ovarian cancer cells and mesothelial cells [7]. Although the platform is capable of temporal control of cell compression (e.g. to decrease, increase or remove the pressure at any certain time interval), duration of each stage in a cycle was set to 5 minutes, corresponding to a 5-min compressed stage followed by 5-min rest stage in each cycle. This was chosen to allow cells enough time during each stage to complete bulge formation or recovery, processes which were considered important for the investigation of cell deformation and recovery after compression. Sensor readings in S1 Movie showed that the micro-piston actuated according to externally applied pressures with reliable control and could be operated in short durations gradually or suddenly when needed.

Cyclic compression in the study of Ho *et al.* was applied only for 6 minutes in total [2], significantly shorter than the entire duration of a cyclic compression in this work, and results presented here suggest that cell deformation at compression stage and recovery at rest stage depends on the temporal length of the compression on cells. Particularly observable in S1 Movie, cell membrane bulge formation slowly developed during the compression stage while cells were responding to contact pressure at the interface with the PDMS piston. This was despite the fact that the piston was actuated onto the cells rapidly during cyclic compression as directed by the external pressure controller. As shown by the pressure sensor readings in the same movie, PDMS membrane and piston were consistently responding to amount, duration and mode of the applied pressure. Thus, the gradual deformation of cells and bulge formation during compression stages must originate in the response type of the cells to mechanical load when in solid contact with the PDMS piston. During rest stages, where the compression was lifted, cell bulges did not fully recover right after compression was removed, suggesting that the cells require a certain amount of time to recover compressed cellular parts (S1 Movie). Based on these observations of temporal evolution of the cell compression application, and

comparison of application durations in this study with the study by Ho *et al.* [2], we hypothesize that this recovery also depends on the duration and number of the cycles.

### Live cell actin profiles of cancer cells during sequential cyclic compression

In addition to the observation of cell membrane bulges, we were further able to track how the application of sequential cyclic compression led to dynamic changes in GFP-tagged live cell actin as a function of applied pressure (see S2 Movie). As mentioned in the section Live imaging of GFP-tagged actin cytoskeleton of cells, for these experiments SKOV-3 ovarian cancer cells were transduced on-chip with CellLight Actin-GFP, BacMam 2.0 baculovirus to visualize the dynamic biomechanical processes during compression (Fig 3A). Specifically, the actin cytoskeleton profile was quantified while cells were in compressed and rest stages (Figs 3B, 3C and 4).

Observations were quantified based on CTCF as given in Eq (1) and results are plotted as shown in Fig 4. A clear correlation between the actin-GFP signal of the cells compressed under the piston and the applied pressure could be observed, with the signal decreasing with increasing pressure. As such, the results illustrate how the actin cytoskeleton of cancer cells changes in response to compressive stress under live-cell conditions, from deformation to disruption, depending on the strength of compression (also see S2 Movie). The actin cytoskeleton of cells, after being flattened under the micro-piston during compression stages (Fig 3C), was undergoing a process of recovery during rest stages (Fig 3B). Throughout a cycle of compression and rest stages, actin structures rearranged themselves and a few new extensions could be observed forming, for instance, during the first cycle of applied mild pressures (15.6–15.9 kPa), while disruption and ruptures became apparent during cycles of higher pressures such as Intermediate 2 (37.8–41.7 kPa) and Severe (127.8–140 kPa) levels (see S2 Movie and Fig 3B and 3C). As such, Fig 3B clearly exemplifies cell actin changes from deformation to disruption over the indicated pressure ranges.

In Fig 4, the change in actin signal for compressed cells under micro-piston was statistically different compared to non-compressed cells in control regions around the micro-piston at all applied compression levels ($p < 0.05$, see S1 File). Actin signal changes in the control regions on the other hand, remained statistically insignificant ($p > 0.05$). Thus, the GFP-tagged actin signal in control regions was robust during the entire process, as expected [30]. This can be reference to that fluorescence changes in the region under micro-piston (i.e. decrease in actin signal) was due to the impact of mechanical compression. At the higher pressures of Intermediate 2 and Severe, actin of most of the cells was disrupted compared to in the initial stages of Mild compression ($p < 0.05$, see S1 File). Interestingly, a few cells on the glass surface usually remained resistant to the increasing pressure ranges, as evidenced by their retained fluorescent signal while compressed under the micro-piston. During the entire process, wide-field imaging was used and the focus was kept on one focal plane on the cells adhered to the glass surface. At compressed stages, glass and PDMS piston surfaces were in contact and at the same focal plane. At the rest stages when pressure is released and piston is retracted back, to be able to gain the signal from piston surface there would be need to bring the focal plane to the suspended piston level that is on average 108 μm above the glass surface, which we did not employ in this application. Thus, we retained the same focal plane and wide-field imaging condition across all stages of the sequential compression on GFP-tagged actin of cells. In the region under the micro-piston, there seem to be variations in gained actin signal between the compressed and rest stages of each cycle (Fig 4), but these were not statistically significant ($p > 0.05$, see S1 File). Thus, no significant amount of cell artifacts was attached to piston surface.

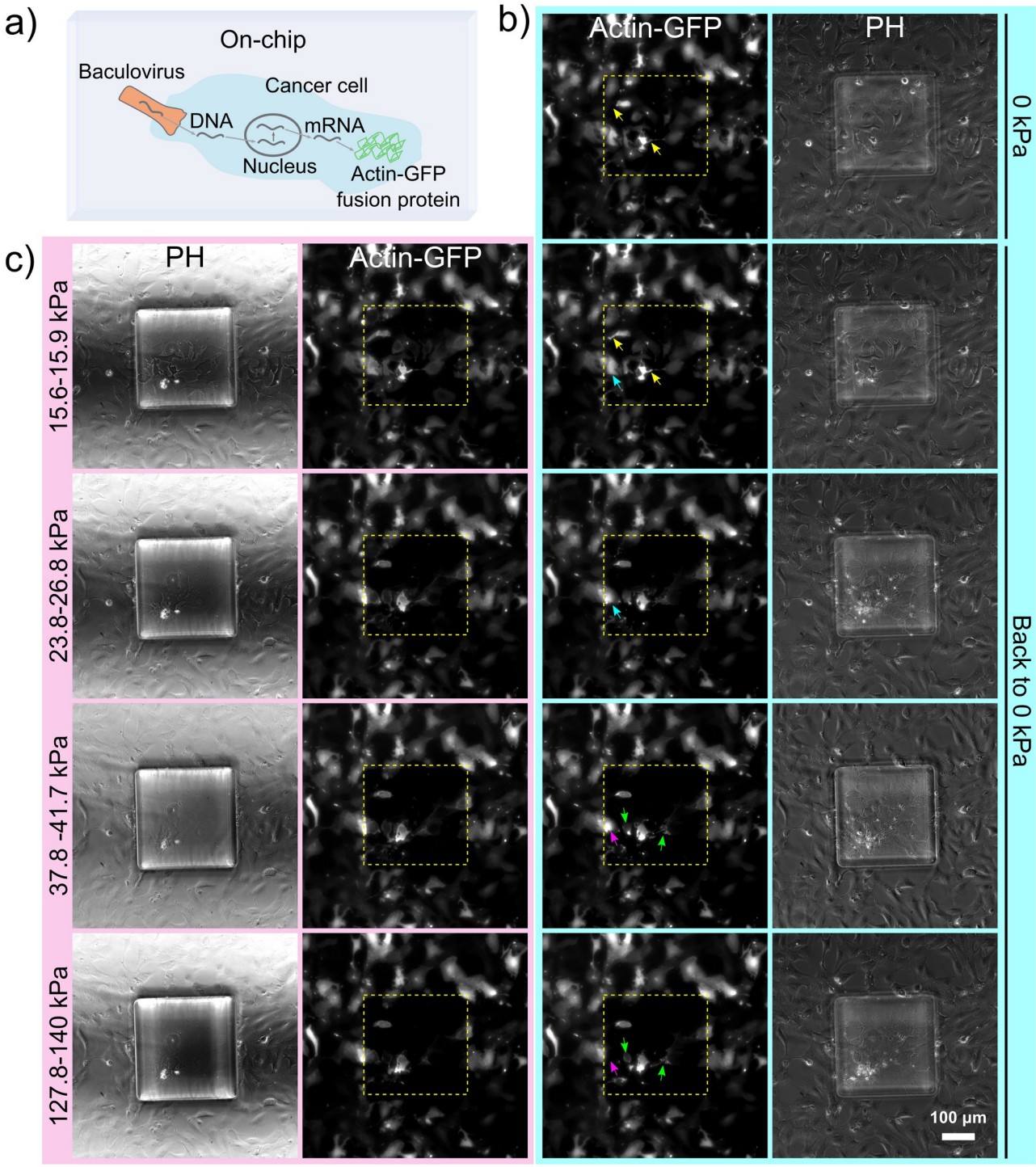

**Fig 3. Representative fluorescent microscopy images of live imaging of GFP-tagged actin cytoskeleton of cells.** (a) Schematic of the on-chip BacMam transduction into cancer cells for the expression of actin-GFP fusion protein. (b) Phase contrast (right) and fluorescent (left) images of the GFP-tagged actin expressing SKOV-3 ovarian cancer cells in static state (0 kPa) or retracted back to static state (0 kPa) after compression for the given pressure range. Yellow arrows indicate example cells exhibiting rearrangement of actin structures with formation of new extensions, blue arrows for actin deformation with rearrangement of existing actin structures, green arrows for actin disruption with raptures and pink arrows for complete actin disruption. (c) Phase contrast (left) and fluorescent (right) images showing the change in GFP-tagged actin in the cells as a result of the applied pressure in the compressed state. Pressure was applied sequentially in ascending order.

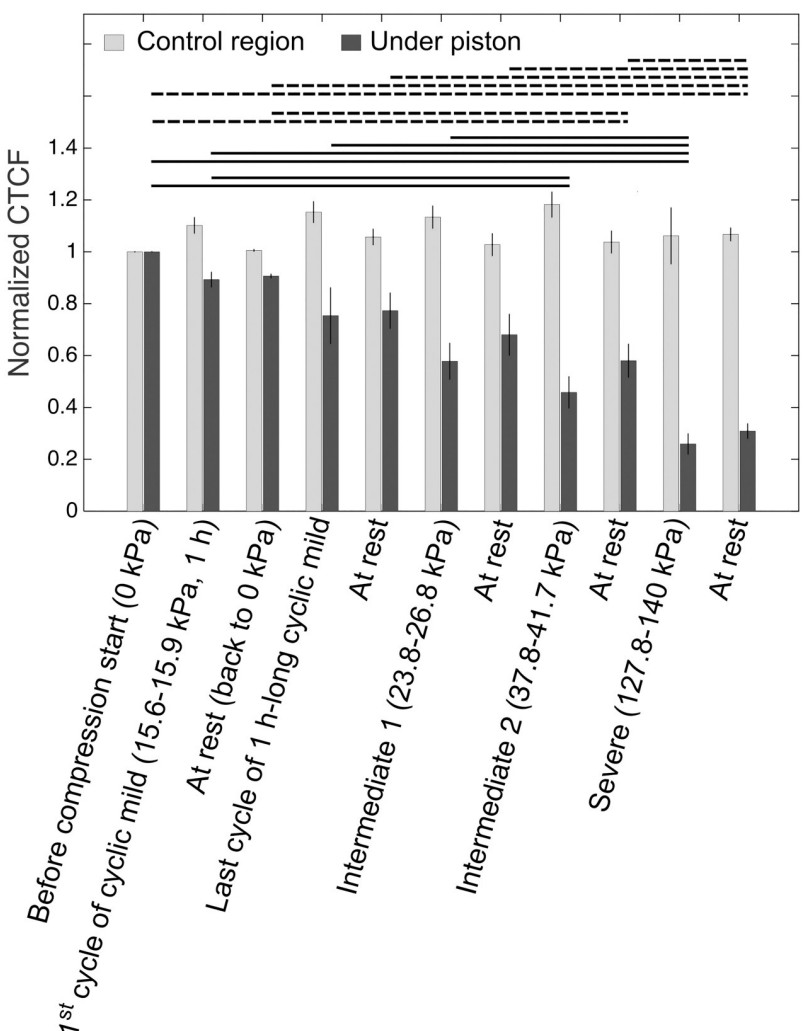

**Fig 4. Corrected total cell fluorescence (CTCF) calculated from images of GFP-tagged actin cytoskeleton of cells during sequential cyclic compression at the indicated pressures.** Student's t-test was used to determine the statistical significance of live cell actin deformation across increasing pressure ranges. Continuous horizontal bars show significant differences between compressed cell groups under micro-pistons for images taken at the time of compression at the indicated pressure ranges. Dashed horizontal bars show significant differences between the cell groups at rest under micro-piston for images taken after compression stages at the corresponding pressure ranges of the cycles. Results represent at least three independent experiments.

CTCF during the rest stage after upper mild compression at Intermediate 2 level was statistically significantly different from the stage before compression (p = 0.0229) and the rest stage after first cycle of Mild compression (p = 0.0352), which indicates that Intermediate 2 level caused a disruption of cell actin. This disruption was significantly higher again at Severe level compared to other levels (p <0.05, see S1 File). As such, these results expand on the work by Ho *et al.*, who used a more limited pressure and frequency range, and observed formation of new actin stress fibers and thickening of a pre-existing actin structure in a very limited number of cells [2]. In contrast, the current work observed live-cell actin in the cytoskeleton from deformation to ruptures at a wider range of applied pressures across several stages of the sequential cyclic compression. The observed disruption of actin cytoskeleton in these experiments indicates for cell plasticity during a cyclic compression in ovarian cancer when varying

higher pressures occur in the microenvironment. Such a plastic response could originate from the cytoskeletal bond ruptures, as suggested by Bonakdar *et al*. who applied mechanical tensile loading on cells [1]. There is need for more studies in literature to compare our results on cell plasticity after cyclic or dynamic compression applied sequentially up to high pressures. For cell actin deformation at milder pressures, however, a distinct change in actin cytoskeleton morphology was observed by Asem *et al*. for mesothelial cells, interacting with ovarian cancer cells, under compression that is statically applied at $\sim 3$ kPa for 24 h in a bulk compression system. As a result, the interaction between ovarian cancer cells and mesothelial cells by enhanced formation of actin-based tunneling nanotubes (TNTs), and hence metastatic progression in ovarian cancer, was promoted under compression [7].

The current device also allowed the response of individual cells at the interface between compressed and non-compressed regions in the periphery of the micro-piston to be studied, thus providing a degree of control of the localization of compression (see S2 Movie). As depicted in the video, parts of cells at the interface were subject to compression, while the rest remained uncompressed. Thus, the cell actin was either deformed or damaged in the compressed part depending on the magnitude of the applied pressure, whereas it retained more of its original shape in the non-compressed part. This functionality which is unique to our platform by having control and compressed groups in the same microdevice, is useful to mimic partial cellular deformation or damage that can occur *in vivo* in presence of a localized mechanical loading.

## Actin cytoskeleton and nuclei profiles of cancer cells after compression and recovery

Applicability of the cyclic compression on the platform was further demonstrated by recording actin and nuclei deformation as an end point assay in cancer cells fixed at zero time or after 24 h-recovery following 1 hour-long cyclic compressions. To this end, we show progression of the compressed cells when live culture was maintained for 24 h-recovery after compression (S3 Fig). Fig 5A shows the distinct actin deformation by the increased fluorescence signals at the edges of the cells in the compressed groups under the micro-piston at zero time after compression, compared to the control (non-compressed) groups around the micro-piston (S5 Fig). Thus, membrane bulges formed during cell compression at 15.6 kPa as observed in S1 Movie and fixed directly after compression appeared as increased actin signal at cell edges (Fig 5A). Conversely, at 24 h-recovery after compression, cell actin deformation seems recovered in the group under the micro-piston as appeared similar to the control group (Fig 5A and S5 Fig).

Nuclear shape analysis was performed on fluorescence images of control (non-compressed) and compressed cells at zero time or 24 h-recovery after compression (S4 and S5 Figs and Fig 5A). No significant change was observed between the control and compressed cells at 15.6 kPa for areal and axial cell nuclei deformation, while compressed cells had larger nuclei than 24 h-recovered compressed cells (Fig 5B and 5D). Conversely, circularity of the cell nuclei was significantly reduced in the compressed cells compared to the control (non-compressed) at zero time after compression, while not significant in the 24 h-recovered compressed group compared to control (Fig 5C). Differences between the compressed cells fixed at zero time or after 24 h-recovery suggest that cells exhibit deformations at the time of the compression, a proposed mechanism cells use to prevent mechanical damage [1, 13]. These deformations were not permanent morphologically however, and could be recovered at mild pressures, such as 15.6 kPa. The absence of statistical difference in nuclei circularity between the 24 h-recovered compressed cells under micro-pistons and that of the non-compressed cells in the control

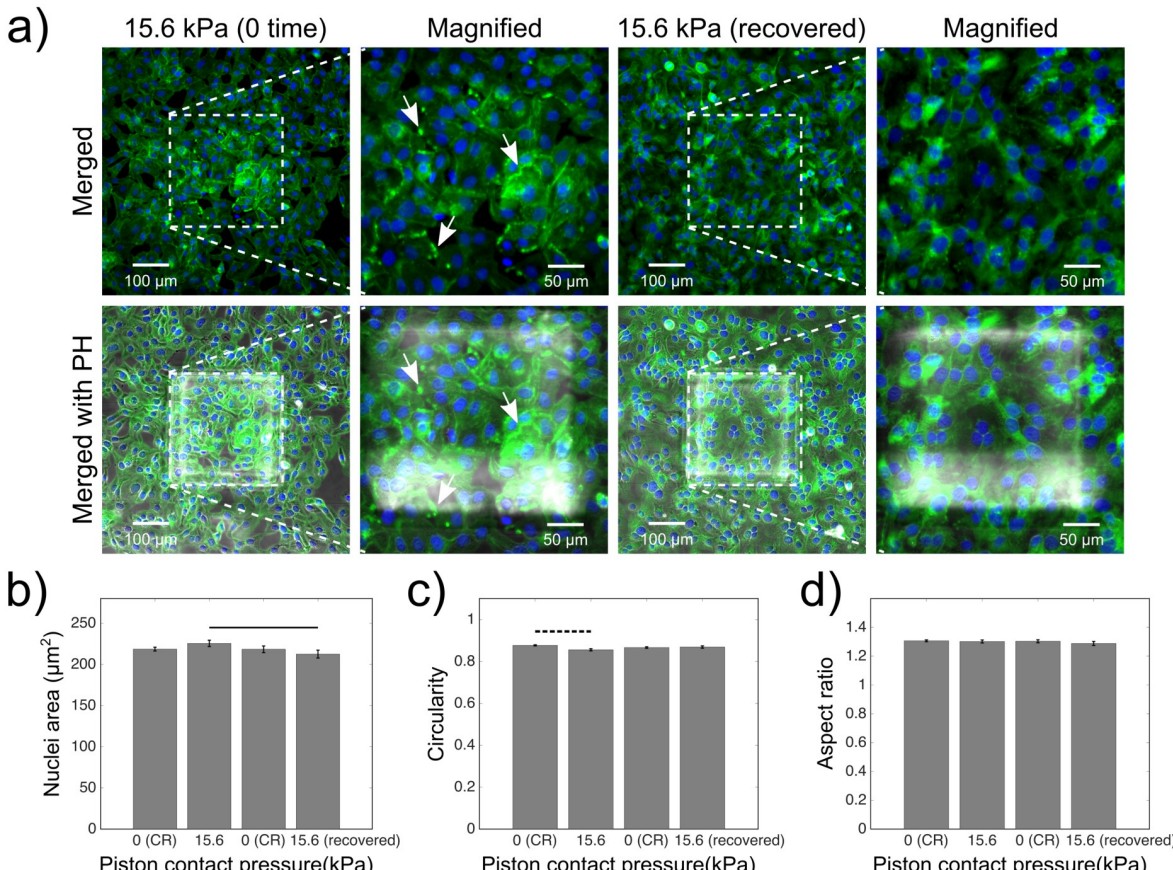

**Fig 5. End point assay staining and analysis for actin and nuclei of cancer cells fixed at zero time or after 24 h-recovery, following 1 hour-long cyclic compression between 0 and 15.6 kPa.** (a) Representative images of control and compressed cell groups stained for actin (green) and nuclei (blue) for their shape at zero time and 24 h-recovery after cyclic compression. Dashed areas are under piston, while the surrounding is control region (CR). Merged: merged form of the phalloidin (stain for actin) and Hoechst (stain for nuclei) epi-fluorescence images; merged with PH: merged form of the epi-fluorescence images with the corresponding phase-contrast (PH) image; magnified: two-hundred-fold magnification images of the region under micro-piston. The unevenness of brightness in phase contrast images is due to that imaging focal plane was on cells cultured on glass while the micro-piston, brought back to static state after compression, was suspended on average 108 μm above the glass. Representative arrows (white) show the distinct actin deformation by the increased fluorescence signals at the edges of the cells in the compressed groups under the micro-piston at zero time after compression. Nuclei deformation in control cells at control regions (CR) versus in compressed cells under piston at zero time or 24 h-recovery after cyclic compression, was measured for area (b), circularity (c) and aspect ratio (d) of the nuclei (mean ± s.e.m. n = 588, 230, 358, 174 cell nuclei from at least two or three independent experiments per each group). Student's t-test was used to determine the statistical significance of nuclear deformations in compressed and recovered cells after compression. Dashed horizontal bars show significant differences between the control region (CR) group and compressed group under micro-piston. Continuous horizontal bars show the comparison pairs between the cell groups under the micro-piston at zero time and 24-recovery after compression. The statistical significance levels among all groups are given in S2 File.

region within those recovery samples (Fig 5C) shows that 24 h-recovered compressed cells started to behave like non-compressed cells during the recovery after compression at 15.6 kPa. Nonetheless, the absence of statistical difference for nuclei circularity between the zero-time and 24 h-recovered cell groups after compression suggests that cells exists that show no full recovery. This divergence might emerge from the variations in decrease of cell volume under compression and nuclear envelope wrinkling and ability of each cell to recover these changes after mild compression [13, 31]. Although the developed platform is not a single-cell compression platform, cells could be assessed as individual cells for their nuclei despite overlapping cellular parts due to dense culture formation on Matrigel coatings. At the same time, the platform

could be considered superior compared to single-cell systems in that it allows for cell-to-cell communication within cell monolayers and cultures on Matrigel coatings on-chip, with conditions in the presence of mechanical compression. Overall, the above experiments constitute a first determination of ovarian cancer cell response to lower physiological compression values, such as 15.6 kPa, applied in a cyclic manner and over 1 hour. To further understand whether cells exhibit different cell recovery profiles after compression, upper physiological values were tested as well, results of which are discussed in the following.

Utilising the full pressure range capabilities of the platform, we also investigated cell recovery after compression at upper physiological values, such as 20.8 kPa (S3 Fig). For this, cells that were compressed at upper mild pressures were assessed for zero time and 24 h-recovery after the compression and compared to non-compressed control cells in each group (Fig 6 and

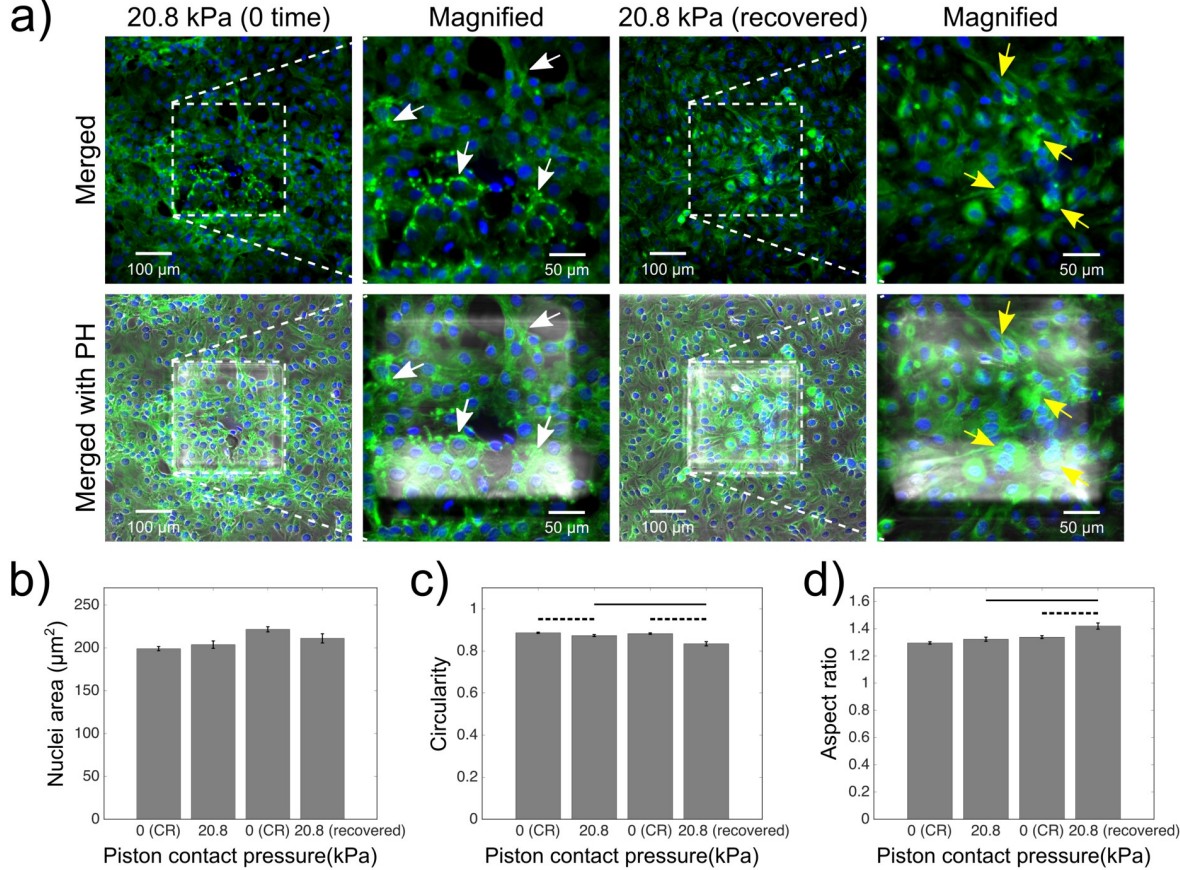

**Fig 6. End point assay staining and analysis for actin and nuclei of cancer cells fixed at zero time or after 24 h-recovery, following 1 hour-long cyclic compression between 0 and 20.8 kPa.** (a) Control and compressed cell groups stained for actin (green) and nuclei (blue) for their form at zero time and 24 h-recovery after cyclic compression. Dashed areas are under piston, while the surrounding is control region (CR). Merged: merged form of the phalloidin (stain for actin) and Hoechst (stain for nuclei) epi-fluorescence images; merged with PH: merged form of the epi-fluorescence images with the corresponding phase-contrast (PH) image; magnified: two-hundred-fold magnification images of the region under micro-piston. Representative white arrows show distinct actin deformation in form of increased fluorescence signals at the edges of the cells in the compressed groups under the micro-piston at zero time after compression, while yellow arrows indicate no full recovery and thus an altered actin profile of cells at 24 h-recovery after compression. Several cells can be observed to have their actin disrupted over the nuclei and cytoplasm, with highly increased stress fibers present at the membrane. Nuclei deformation in control cells at control regions (CR) versus in compressed cells under piston at zero time or 24 h-recovery after cyclic compression, measured for area (b), circularity (c) and aspect ratio (d) of the nuclei (mean ± s.e.m. n = 406, 162, 342, 143 cell nuclei from at least two or three independent experiments per each group). Student's t-test was used to determine the statistical significance of nuclear deformations in compressed and recovered cells after compression. Dashed horizontal bars show significant differences between the control region (CR) group and compressed group under micro-piston. Continuous horizontal bars show the comparison pairs between the cell groups under the micro-piston at zero time and 24-recovery after compression. The statistical significance levels among all groups are given in S2 File.

S6 Fig). Cells treated this way developed distinct actin deformation, as shown by the increased fluorescence signals at the edges of the compressed cells under the micro-piston at zero time after cyclic compression at 20.8 kPa for 1 hour. Some cells had their actin visibly disrupted over the nuclei and cytoplasm, while highly increased stress fibers presented at the cell membrane (Fig 6A). This result is consistent with changes in actin cytoskeleton and formation of cell surface projections observed in compressed peritoneal mesothelial cells in the study by Asem *et al.* [7]. Interestingly, in the current work cell actin signal was localized mostly in compressed cells under micro-piston at 24 h-recovery after cyclic compression (Fig 6A), compared to control cells (S6 Fig). For compressed cells this outcome indicates no full recovery from the impact of the compression at 20.8 kPa, and thus an altered actin cytoskeleton profile. Shape descriptors of the cell nuclei were measured for zero-time and 24 h-recovery after the cyclic compression. There was no statistical difference in nuclei area between compressed or 24 h-recovered compressed groups and with their control groups (Fig 6B). There is statistically significant change in nuclei area between the control groups of zero-time and of 24 h-recovery after compression (see S2 File). Although these are not exactly the same control groups in the same conditions and we intend to compare the test group (e.g. zero-time or 24 h-recovery after compression) with their own control groups, we consider that the observed significant difference could be due to cell-cell signaling between the cells in compressed and control groups during that 24 h-recovery period after compression. In the groups of zero-time after compression, we did not allow control and compressed cells enough time to further interact with each other as they were fixed right after the compression. While we expect a statistically significant change between the compressed and its control group at zero-time after compression at 20.8 kPa similar to the results observed in our previous report [5], we attribute the absence of such a significant change to cells being cultured on Matrigel-coating here as opposed to PLL-coating used in the previous experiments. Since Matrigel is expected to better represent the cancer cell microenvironment, it thus aided in cell growth into denser monolayers and formation of supporting extracellular matrix layer, that can alter sensing of the applied stress for the compressed cells.

On the other hand, circularity of the compressed and 24 h-recovered compressed cells statistically significantly decreased compared to their respective control groups (Fig 6C). Circularity of the 24 h-recovered compressed cells was also statistically less than the compressed group at zero-time after compression (Fig 6C). Furthermore, the aspect ratio of the 24 h-recovered compressed cell nuclei was statistically significantly higher than its 24 h-recovered control group and compressed group at zero time after compression (Fig 6D). These results suggest that cells did not fully recover after cyclic compression at upper mild physiological pressure values such as 20.8 kPa. The phenotypical response of compressed cells during recovery after compression at 20.8 kPa differed observably from that for 15.6 kPa, demonstrating the capability of the flexible microdevice to capture differences in cell behaviour after being compressed at various pressures in a physiologically relevant manner. In fact, such changes in cell actin and nuclei shape during recovery after upper mild compression as observed here may indicate formation of more aggressive and metastatic cells [32].

In the study by Ho *et al.*, the height of breast epithelial cells recovered fully after 6 minutes of cyclic compression applied between the external pressures of 10 and 15 psi at 0.5 Hz [2]. No statistical difference in cell height before and after compression was observed, indicating that the breast epithelial cells did not exhibit permanent deformation. Measurement of the viscoelastic relaxation of the cells is difficult however, especially when vertical and temporal resolution in imaging are limited, and long image acquisition times required for 3D visualization might prevent capturing the changes in cell height at the time of compression. In the current study, we thus deliberately limited cell deformation assessment to planar dimensions including

area, circularity, and aspect ratio [24], as wide-field imaging is considered fast enough (e.g. milliseconds long) to capture cell states during the steps of the compression process compared to real-time volumetric imaging (e.g. minutes long) using confocal microscopy [2, 5, 33–36].

In summary, our platform was shown to be able to control the strength and duration of cyclic compression, all while enabling the observation of morphological, cytoskeletal and nuclear changes in cells. It thus provides a powerful new tool for the study of mechanobiological processes in cancer and cell biology. For instance, we propose that the presented sequential cyclic compression and cell recovery after compression method should be employed in the future to study the effects of compression and decompression in intratumor vessels or endothelial cells to help understand the structure and functionality of decompressed vessels, importance of which was shown in a study by Padera *et al.* [37]. Furthermore, the data obtained in this work suggest that ovarian and likely other cancer cells can exhibit a varying cell response to applied sequential cyclic compression. Cells in the current work were able to do so by deformations and perturbations in actin cytoskeleton and nuclei. It is proposed that cells exhibit such deformations at the time of compression to compensate for the applied compressive stress and protect themselves against mechanical damage. Those deformations were recovered fully or partially depending on the amount of applied pressure. At upper mild pressures, such as 20.8 kPa, cells appear to develop and use an adaptive mechanism through incomplete shape recovery. Whereas recovery after mechanical loading is important to maintain cell integrity, an adaptive mechanism in the form of incomplete shape recovery might indicate the emergence of more aggressive and metastatic cancer cells.

## Conclusions

We have demonstrated coating of cell culture chambers in micro-piston devices with hydrogels and successful culture of SKOV-3 ovarian cancer cells therein, mimicking the *in vivo* cellular microenvironment. Similar to Matrigel coating used in this study, our platform and established chip loading method would allow coatings of other hydrogel types such as collagen and fibronectin. Furthermore, micro-piston devices fabricated with different membrane thicknesses were used to cyclically compress cells with controlled micro-piston contact pressures, verified using FEM simulations to account for possible membrane thickness variations. Computational results showed good agreement with experimentally applied cyclic compressions. Sequential cyclic compression profiles were applied to the actin-GFP transduced cancer cells. Thus, live cell actin dynamics was recorded based on the cell response to pressure profiles applied in cyclic mode at a mild pressure level and sequentially at intermediate and severe pressures in ascending order. While actin deformation at mild pressures (e.g. 15.6 kPa) in cyclic mode indicates for cell elasticity, the resultant actin disruption indicates for cell plasticity that can emerge when ovarian cancer cells are exposed to varying and higher pressures in a microenvironment. The applications were extended to recording of the long-term cellular changes in mechanosensitive subcellular structures, such as actin cytoskeleton and nuclei, while allowing cells to recover for 24 hours after compression. Details of morphological cell deformations during sequential cyclic compression were captured via high-speed imaging. Membrane bulges in cells fixed directly after compression appeared as increased actin signal at cell edges. We also conclude that recovery of the compressed live cells was dependent on magnitude of the applied compression. The extent of recovery of compressed cells can give insights into the cell integrity and adaptation of cancer cells while preventing mechanical damage when exposed to mild or upper mild compressive stress at physiological levels. As demonstrated here with controlled micro-scale mechanical cell compression and recovery in a flexible microdevice, our comprehensive method will provide a more accurate replication of

cell-physiological mechanisms to study both short- and long-term effects of compression in cellular microenvironments.

## Supporting information

**S1 Fig. On-chip growth of SKOV-3 cells on Matrigel-coated glass surfaces in micro-piston devices.** Representative images of two samples during culture from zero time to Day 4 after piston-retracted loading of cells at -615 mbar.
(TIF)

**S2 Fig. Illustrations of the piston contact pressure area by the symmetrical half-cell of the piston and cell-culture chamber in the computational model.** They show the distribution of contact pressure (in N/m$^2$) under piston for the selected applied external pressures (in mbar). Red dashed line of model symmetry drawn on the illustrations at different pressures point out the middle side of the piston. The contact pressure is slightly higher at the edges of the piston towards the side channel walls where the attached membrane distance between piston and side wall is shorter. Thus, there seem to be some variation due to how the piston bottom gets squashed in these sides of the piston, but it is minimal compared to the overall area. This is demonstrated by the color being relatively even over the whole area, especially for externally applied lower pressures such as 0–420 mbar.
(TIF)

**S3 Fig. Examples of 24 h-recovery of mildly and upper-mildly compressed SKOV-3 cells on Matrigel-coated glass surfaces in micro-piston device.**
(TIF)

**S4 Fig. Image analysis of nuclear shape after cell compression on Matrigel-coated surfaces.** Regions of interest (ROIs) extracted for the cell nuclear boundaries in Hoechst (stain for nuclei) epi-fluorescence channel (blue in merged/composite frames) were used to measure area, circularity, and aspect ratio of the cell nuclei, as independent of phalloidin (stain for actin) epi-fluorescence channel (green in merged/composite frames). Merged—200X: merged form of the phalloidin and Hoechst epi-fluorescence images obtained at two-hundred-fold magnification; merged with PH—200X: merged form of two-hundred-fold magnification epi-fluorescence images with the corresponding phase-contrast (PH) image for the region under micro-piston; merged with PH—100X: merged form of one-hundred-fold magnification epi-fluorescence images with the corresponding phase-contrast (PH) image for the region under and around micro-piston in channel.
(TIF)

**S5 Fig. End point assay staining and analysis for actin and nuclei of cancer cells in control region (CR) fixed at zero time or after 24 h-recovery, following 1 hour-long cyclic compression between 0 and 15.6 kPa.** Control cell groups stained for actin (green) and nuclei (blue) for their form at zero time and 24 h-recovery. Merged: merged form of the phalloidin (stain for actin) and Hoechst (stain for nuclei) epi-fluorescence images; merged with PH: merged form of the epi-fluorescence images with the corresponding phase-contrast (PH) image; Random region 200X: two-hundred-fold magnification images of control cells in random region as part of the control region.
(TIF)

**S6 Fig. End point assay staining and analysis for actin and nuclei of cancer cells in control region (CR) fixed at zero time or after 24 h-recovery, following 1 hour-long cyclic compression between 0 and 20.8 kPa.** Control cell groups stained for actin (green) and nuclei

(blue) for their form at zero time and 24 h-recovery. Merged: merged form of the phalloidin (stain for actin) and Hoechst (stain for nuclei) epi-fluorescence images; merged with PH: merged form of the epi-fluorescence images with the corresponding phase-contrast (PH) image; Random region 200X: two-hundred-fold magnification images of control cells in random region as part of the control region.
(TIF)

**S1 Movie. Temporal evolution of the dynamic cell compression within Matrigel-coated micro-piston devices.** Time-lapse live cell imaging with 100 ms per frame was recorded for ladder pressure increase from 0 kPa up to 15.6 kPa and a short cycle at 15.6 kPa with gradual increase (from 0 kPa to 15.6 kPa) and decrease (from 15.6 kPa to 0 kPa) in 30 s for each. These steps were sequentially followed by 1 h-long cyclic compression between 0 kPa and 15.6 kPa (piston contact pressure from simulation). Cells were compressed in the temporal evolution of the dynamic pressure control within Matrigel-coated micro-piston devices and dynamic changes in cells during the cyclic compression were captured in detail at a rate of 100 ms per frame.
(MP4)

**S2 Movie. Representative data recording of the dynamic changes of GFP-tagged live cell actin during application of sequential cyclic compression at various applied pressures.** SKOV-3 ovarian cancer cells were transduced on-chip with CellLight Actin-GFP, BacMam 2.0 baculovirus (Invitrogen) to study the dynamic biomechanical processes under compression. The actin cytoskeleton profile was quantified for the cancer cell response under applied forces at compressed and rest stages during the compression process.
(MP4)

**S1 File. Corrected total cell fluorescence (CTCF) data for actin integrated density and statistical test results.**
(XLS)

**S2 File. Statistical significance levels among all groups, obtained with Student's t-test run for nuclear deformations in compressed and recovered cells after compression presented in Figs 5 and 6.**
(XLSX)

## Acknowledgments

The authors thank Helen Devereux, Gary Turner, and Nicole Lauren-Manuera for assistance. We also thank Mathieu Sellier for help with COMSOL, and Campbell Sheen for assistance with the EPA protocol to use the Actin-GFP BacMam construct.

## Author Contributions

**Conceptualization:** Sevgi Onal, Volker Nock.

**Data curation:** Sevgi Onal.

**Formal analysis:** Sevgi Onal.

**Funding acquisition:** Sevgi Onal, Volker Nock.

**Investigation:** Sevgi Onal, Volker Nock.

**Methodology:** Sevgi Onal, Volker Nock.

**Project administration:** Volker Nock.

**Resources:** Volker Nock.

**Supervision:** Maan M. Alkaisi, Volker Nock.

**Validation:** Sevgi Onal.

**Visualization:** Sevgi Onal, Volker Nock.

**Writing – original draft:** Sevgi Onal, Volker Nock.

**Writing – review & editing:** Sevgi Onal, Maan M. Alkaisi, Volker Nock.

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
