## [Decision Letter · Decision Letter 0]

23 May 2022

PONE-D-22-08675Application of sequential cyclic compression on cancer cells in a flexible microdevicePLOS ONE

Dear Dr. Onal,

Thank you for submitting your manuscript to PLOS ONE. After careful consideration, we feel that it has merit but does not fully meet PLOS ONE’s publication criteria as it currently stands. Therefore, we invite you to submit a revised version of the manuscript that addresses the points raised during the review process. 2 reviewers recommended rejection of the paper, but I feel that their comment might be addressed in a revised version of the article. The 3 reviewers are concerned about the novelty of the device and do not understand the relevance of the cyclic compression. These are 2 main claims of the paper and these 2 points absoultely need to be clarified. Shoud you decide to submit a revised version, it will be reevaluated and I can not guarantee a positive answer. If you decide to submit a revised manuscript please submit by Jul 07 2022 11:59PM. If you will need more time than this to complete your revisions, please reply to this message or contact the journal office at plosone@plos.org. Please include the following items when submitting your revised manuscript:A rebuttal letter that responds to each point raised by the academic editor and reviewer(s). You should upload this letter as a separate file labeled 'Response to Reviewers'.A marked-up copy of your manuscript that highlights changes made to the original version. You should upload this as a separate file labeled 'Revised Manuscript with Track Changes'.An unmarked version of your revised paper without tracked changes. You should upload this as a separate file labeled 'Manuscript'.

We look forward to receiving your revised manuscript.

Kind regards,

Etienne Dague, PhD

Academic Editor

PLOS ONE

Journal Requirements:

[The authors thank Helen Devereux, Gary Turner, and Nicole Lauren-Manuera for assistance. We also thank Mathieu Sellier for help with COMSOL, and Campbell Sheen for assistance with the EPA protocol to use the Actin-GFP BacMam construct. Financial support was provided by the MacDiarmid Institute for Advanced Materials and Nanotechnology and the Biomolecular Interaction Centre. Additional funding for V.N. was provided by Rutherford Discovery Fellowship RDF-19-UOC-019. We also thank University of Canterbury for Faculty of Engineering PhD Publishing Scholarship to S.O.]

 [Financial support was provided by the MacDiarmid Institute for Advanced Materials and Nanotechnology and the Biomolecular Interaction Centre. Additional funding for V.N. was provided by Rutherford Discovery Fellowship RDF-19-UOC-019. S.O. also thank University of Canterbury for Faculty of Engineering PhD Publishing Scholarship.]

Reviewers' comments:

Reviewer's Responses to Questions

**Comments to the Author**

1. Is the manuscript technically sound, and do the data support the conclusions?

Reviewer #1: No

Reviewer #2: No

Reviewer #3: Yes

2. Has the statistical analysis been performed appropriately and rigorously? 

Reviewer #1: I Don't Know

Reviewer #2: No

Reviewer #3: Yes

3. Have the authors made all data underlying the findings in their manuscript fully available?

Reviewer #1: Yes

Reviewer #2: Yes

Reviewer #3: Yes

4. Is the manuscript presented in an intelligible fashion and written in standard English?

Reviewer #1: Yes

Reviewer #2: Yes

Reviewer #3: Yes

5. Review Comments to the Author

Reviewer #1: Cancer mechanics are important for tumor biology and progression, but the effects of mechanical forces on cancer cells are poorly understood. The authors present methodology for applying compressive forces to cancer cells using a microfabricated platform. They show that cyclic compression affects the intensity of actin-GFP and nuclear morphology in ovarian cancer cells. Although this is an interesting system, there are a number of issues related to novelty, biological relevance and interpretation that need to be addressed.

Specific comments:

1) In the introduction, it is mentioned that "...cells are exposed to chronic compressive stress from different sources such as tumour growth, displacement within stromal tissue and hydrostatic pressure out of ascitic fluid in peritoneal cavity." This is not completely accurate, as hydrostatic pressure will not deform the cell in the same way as anisotropic compression. The forces would be isotropic for fluid pressure.

2) The rationale for using matrigel is not clear. Collagen and/or fibronectin would be more representative.

3) Does the disruption of the cytoskeleton depend on the rate of application of the force? If the piston is applied more slowly, do the cells have time to adjust and avoid damage?

4) In Figure 3, it looks like some labeled cells are actively moving away from the area of the piston. Did this affect the measurements? Related to this, the authors might refer to doi.org/10.1073/pnas.1118910109.

5) Are the differences in cell response (and death) seen in a given compression experiment related to cell cycle?

6) In Fig 6 b, it appears that there is a difference between the two "0 (CR)" values. Was this significant, and can the authors discuss the details of these samples?

7) The meaning of "plasticity" should be defined more clearly. Does plasticity refer to mechanical lack of elasticity? If so, why would we expect this to be related to loss of actin-GFP? Also, plasticity would be reflected by a lack of recovery of the cell to its original state. Or are the authors simply analyzing biological processes that restore the cell after stress?

8) Similarly, the meaning "recovery" should be defined more clearly. The authors write "The extent of recovery of compressed cells, inferred by imaging cell membrane bulges and actin cytoskeleton and measuring the shape descriptors of cell nuclei, can give insights into the plasticity of cancer cells." However, there are no measurements of cell membrane bulges, and no spatial analyses of actin organization. How do we know the cells have "recovered"?

9) Other groups have published similar pressure-actuated devices for compressing cells, so there should be more discussion of these studies and comparison with the presented technology. In general, a more extensive literature review on in vitro analyses of cancer cell solid stress would be useful.

10) Given the relatively rapid application and removal of compressive forces, there must be some interesting fluid dynamics happening between the cells and the piston. Can the authors estimate the changes in fluid pressure and shear stress on the cells? This could also affect the cell biology.

11) Further justification should be provided for the cyclic stress schedule chosen. Musculo-skeletal tissues would be expected to experience such forces, but this is not clear for ovarian cancer.

Reviewer #2: The following are the attached Comments for Authors.

In this manuscript, authors characterized changes and recovery in actin fluorescence signal and nucleus shape after applying cyclic compression to ovarian cancer cells. However, there are several major issues with the current manuscript: 1) the introduction lacks information and citations of previous work and does not address the importance of this work, i.e., why apply cyclic compression on ovarian cancer cells; 2) Key parts of the experimental design lack rationale, i.e. why 5 min pressure 5 min recovery, how large is the piston/cell contact area, variability of the pressure within contact area, etc. 3) Several results are questionable, as detailed the following comments; and 4) Several results lack information on significance tests and number of independent studies. As such, the reviewer recommends rejection.

1. The introduction is not clear what the similarities and differences are between chronic and cyclic compression, and why cyclic compression is important to study in ovarian and other cancer cells.

2. Page 2, line 9-11: “Such compression is estimated to reach 18.9 kPa for human tumours and can exceed 20 kPa based on the experimental data from murine tumours [3, 7].” Given the wide range of compressive stresses in different tumor models & species, the authors should specify the physiological range of compression for ovarian cancer.

3. The introduction is not clear about the similarities and differences between chronic and cyclic compression, and why cyclic compression is important to study in ovarian and other cancer cells.

4. The innovations in the microdevice authors developed are not clear. What has been developed in the past and what is new should be summarized in the Introduction.

5. References supporting nucleus as mechanosensory are lacking. Kirby (2018) Nature Cell Biology, Fu (2012) Lab on a Chip, etc. Additionally, past findings on nucleus deformation in ovarian cancer cells and its functional effects should be introduced as background to this study

6. Page 7, Fig 2(b): it is not clear which are experimental data and which are simulated pressures on the graph, and how well they match quantitatively.

7. Page 6, line 198: “maximum piston contact pressure”…the authors should include how much variation in pressure exists for the entire area under the piston, and address how this variation may affect results

8. Page 8, line 234: It is not clear why authors chose 5 min pressure stage followed by 5 min rest stage. Experimental characterization of nucleus deformation recovery time in relationship to pressure AND/OR theoretical support is needed.

9. Page 9: Panels a-c in Figure 3 were not referenced in the text.

10. Page 11, figure 4 should include levels of significance for intended comparison.

11. Page 9, line 295-297: It is not clear the lower actin signal from higher pressure groups is due to higher pressure or accumulated stress from the sequential testing. Independent experiments should be carried out for each pressure..

12. Page 9, Line 304-306: Were these “resistant” cells in contact with the piston? Characterization of the interface between the piston and the cell substrate AND/OR other metrics (ie., nucleus shape/area) is needed to rule out artifact.

13. Page 10-11: the authors should consider discussing impact of the observed disruption of actin /no recovery due to cyclic compression in ovarian cancer and the relevance to cell plasticity

14. Page 13, figure 5 b-d needs to include details of significance tests conducted and the significance levels detected.

15. Page 12, line 352: it is not clear what specific heterogeneity the authors mean here

16. Page 12, line 391: minute differences (?)

17. Page 13: in Fig 5(a) 15.6 kPa (o time), actin signal looks higher in compressed area compared to cells in control area? The authors should also explain the unevenness of brightness in phase contrast images

18. Fig. 5 & 6 quantification of nucleus metrics: how many independent experiments were conducted?

19. Page 12, line 380: authors should explain why no significant changes in nucleus area were observed, as it would be expected given the high stress cells were experiencing. The authors should also explain the visible difference in two “0 (CR)” groups in Figure 6 (b)

Comments to the Editor:

This study lacks novelty.

RECOMMENDATION:

Reject.

Reviewer #3: Onal and colleagues present a previously published method to dynamically compress a monolayer of cells. In this article, they use this device to briefly study how compression can impact actin expression, through a live reporter.

The study is technically sound. However, there are some points that need to be addressed before I can reach a decision:

1/ The claim of novelty of the device needs to be taken away. The device does not seem different from the one published by the authors in 2021 in Frontiers in Physics, and is inspired by the one published by Mishra et al (PNAS 2017), which should be acknowledged. In their 2021 paper, the authors notably already perform cyclic compression. Can the authors comment on what is really new, and if nothing is, can they transform the abstract and parts of introduction / conclusion to have it more focused on the study on actin?

2/ I was surprised that the authors present their method as a great strategy to apply cyclic compressive stresses, but did not really present results on this part (playing with frequency for instance, etc). It is discussed on p.15, but no experiments are presented. Have the authors played with this parameter? If not, maybe it would be better to put less emphasis on this in the manuscript. Also, can the authors provide a rationale for the frequency of the oscillations used in this study?

3/ I have some difficulties with the motivations in the introduction for the choice of ovarian cancer cells. In cancer, compressive stress is very slowly increasing, and not cyclically changing. Dynamic stress, though, can be found in the heart or during breathing, or also through the peristaltic motion during digestion. These examples, which would be great to motivate the dynamic capabilities of the device, are not presented, and instead, the focus is made on cancer. Moreover, I would not call 16 or 21 kPa as “mild” compressive stresses: these are rather important. The authors may wish to re-write their introduction if they want to introduce the device through its dynamic capabilities. But in this case, ovarian cancer cells may not be the best model (although, the authors can just say they use these cells as a model cell line for this test study).

4/ The actin intensity result seems correlated to survival, oddly. Can the authors plot these two parameters alongside as a function of pressure on same plot? Can the authors speculate on the decrease observed under pressure? Finally, have the authors checked for bleaching, by changing the sequence of compression for instance?

Minor points:

1/The paper is too affirmative at parts (for instance: p.9, “the results illustrate how the actin cytoskeleton […] changes in response to compressive stress” is overstated as the authors show a correlation, and do not provide a clear mechanism). The authors need to be more careful on their claims.

2/ Could the authors put all pressure values in the same unit? We sometimes read kPa, mbar or psi (which is by its very name not a pressure unit and not in the international system of units), and a harmonization would help the reader.

3/ Part at the end of page 8 read more like introduction and the end of page 14 / beginning of page 15 read more like a discussion. Maybe they would be better placed in these specific sections to avoid confusion.

6. PLOS authors have the option to publish the peer review history of their article (what does this mean?). If published, this will include your full peer review and any attached files.

Reviewer #1: No

Reviewer #2: No

Reviewer #3: No

---

## [Author Response · Author response to Decision Letter 0]

7 Jul 2022

Reviewer #1: Cancer mechanics are important for tumor biology and progression, but the effects of mechanical forces on cancer cells are poorly understood. The authors present methodology for applying compressive forces to cancer cells using a microfabricated platform. They show that cyclic compression affects the intensity of actin-GFP and nuclear morphology in ovarian cancer cells. Although this is an interesting system, there are a number of issues related to novelty, biological relevance and interpretation that need to be addressed.

Specific comments:

1) In the introduction, it is mentioned that "...cells are exposed to chronic compressive stress from different sources such as tumour growth, displacement within stromal tissue and hydrostatic pressure out of ascitic fluid in peritoneal cavity." This is not completely accurate, as hydrostatic pressure will not deform the cell in the same way as anisotropic compression. The forces would be isotropic for fluid pressure.

Asem et al. (2020) Scientific Reports, show that ascites-induced compression exist in the peritoneal microenvironment due to the ascites volume that can reach >2L and cause intraperitoneal pressure (IPP) to rise from normal values of 5 mmHg to as high as 22 mmHg (~3 kPa). Ovarian cancer cell adhesion to peritoneum increased due to the impact of ascites-induced compression, shown in in vivo assay.

Asem et al. also applied a pressure of ∼3 kPa on OVCAR5 or OVCAR8 cells added atop the mesothelial surface of murine peritoneal explants ex vivo for 24 hours in a bulk compression system. As a result, they observed enhanced interaction between peritoneal mesothelial cells and cancer cells via induction of tunneling nanotubes (TNT), which later lead to metastatic ovarian cancer progression. These compression-induced TNTs are actin based and aid in trafficking of mitochondria between ovarian cancer cells and mesothelial cells.

The other consequence of the ascites-induced compression through hydrostatic pressure in the peritoneal cavity has been a more linear anisotropic alignment of peritoneal collagen fibers. Such alignment of collagen is a common indication for enhanced invasion of solid tumors.

Overall, Asem et al. have found that ascites-induced compression promotes metastatic ovarian cancer progression. 

This information has been summarized and added in the first paragraph of the Introduction: “…….Ascites-induced compression exists in the peritoneal microenvironment due to the ascites volume that can reach >2L and increases ovarian cancer cell adhesion to peritoneum, shown by Asem et al. using in vivo assay [7]. This peritoneal microenvironment is continuously affected by the movements of the surrounding organs, musculoskeletal dynamics, and gravity [6].…… ”

2) The rationale for using matrigel is not clear. Collagen and/or fibronectin would be more representative.

In the Materials and methods section “Cell culture, Matrigel coating and chip loading with cells” it was noted that “…… Matrigel was previously shown to facilitate cancer cell adhesion and mimic the extracellular matrix (ECM) microenvironment in comparison to other substrates ….. .”

It has been also added into this section and further referenced that “Matrigel is commonly used basement membrane extracellular matrix for providing improved microenvironmental and physiological conditions in in vitro culture assays of cancer cells by its composition [20-23].”, as shown by such as Kleinman et al (1982) Biochemistry, Mullen et al (1996) Int J Cancer, Kleinman et al (2005) Seminars in Cancer Biology, and Vaillant et al (2011) Breast Cancer.

Major components of Matrigel are laminin, collagen IV, heparan sulphate proteoglycans, entactin and nidogen (Kleinman et al.,1982, Biochemistry). It also contains various growth factors such as TGF-�, epidermal growth factor, fibroblast growth factor and tissue plasminogen activator, insulin like growth factor which are naturally occurring in tumor microenvironment (Corning Inc.). Thus, Matrigel is rich in extracellular matrix proteins and polymerises under normal physiological conditions to produce a reconstituted, biologically active matrix material which is effective for the attachment and differentiation of cellular material as shown by Kleinman et al (1982) Biochemistry.

It has been also added in Conclusions “Similar to Matrigel coating used in this study, our platform and established chip loading method would allow coatings of other hydrogel types such as collagen and fibronectin”, which could be tried as in the method shown in Figure 1(b). 

3) Does the disruption of the cytoskeleton depend on the rate of application of the force? If the piston is applied more slowly, do the cells have time to adjust and avoid damage?

Regarding deformation and disruption of actin cytoskeleton, Figure 3 and Figure 4 show how live cell actin cytoskeleton behave under dynamic compression applied over time in cyclic mode at a physiological value and sequentially in varying applied pressures in ascending order, and are out of at least 3 independent experiments. In text, while explaining the associated figures, the representative pressure profile for sequential cyclic compression process was indicated to be in Figure 1(d): “Similar as to the sequence shown in Fig 1(d), first, a pressure amount ladder was formed up to a pressure in the mild compression range, represented by an externally applied pressure of 350 mbar and resulting in a piston contact pressure of 15.6 kPa, during which the piston reached the cells at the bottom surface and cells started to distinctly deform.”

We think that this pressure application is gradual enough for cells to adjust to the varying pressures during sequential cyclic compression application. Mechanical cell damage that we see at higher pressures is intrinsic to cell response to higher amounts of applied pressure. The following explanation has been added at the end of the 1st paragraph in section “Live cell actin profiles of cancer cells during sequential cyclic compression”:

“Throughout the application of sequential cyclic compression, cells were allowed to adjust to each next higher applied pressure. As shown in the pressure amount ladder in Fig 1(d), pressure was applied gradually in 30 s among rising steps of the ladder until the piston reached the focal plane of the cells and started cell deformation. After observing that cells were distinctly deformed at a mild compression level but not to a disruption or lysis level, compression was set to be in cyclic mode. Although the pressure was changing suddenly along the cycles as per the settings of the selected cyclic mode in the pressure controller, cells were not damaged when compressed as they resulted in being resistant at this physiological value. Next, pressure application was continued with gradual increase from 350 to 370 mbar, from 370 to 400 mbar and from 400 to 640 mbar in 30 s for each, as shown in Fig 1(d). Thus, cells could have the time to adjust during the pressure increase and show their intrinsic response to the pressurized stage of each compression level.”

4) In Figure 3, it looks like some labeled cells are actively moving away from the area of the piston. Did this affect the measurements? Related to this, the authors might refer to doi.org/10.1073/pnas.1118910109.

For CTCF calculations as shown in Eq (1) (format of the equation has been revised) in methods section “Imaging and data analysis”, while analysing the cells under micro-piston we must draw an ROI and set it as threshold that separates the analysis area (Area of micro-piston ROI) from the rest which is control group. We set an ROI from the beginning and then we do not change its boundaries during the frames taken along the stages of the dynamic compression process. The way we decided the micro-piston ROI is shown as in the ROI (yellow frame) drawn the videos in the links, we accept slightly larger area which is drawn based on the pressurized from of the micro-piston. 

Phase contrast with micro-piston ROI: https://www.dropbox.com/s/sixcpmtw2abteth/S300um_3pistonSKOV-3%20actin-GFP_phasecontrast_withROI.avi?dl=0

Actin-GFP with micro-piston ROI: https://www.dropbox.com/s/1ww7p90el7t3ej1/S300um_3pistonSKOV-3%20actin-GFP_withROI.avi?dl=0

Cells are partially compressed partially not under the micro-piston periphery (boundaries of the ROI), so in CTCF calculations the compressed and then buckled back parts of the cells were taken into account. This was explained as in the last paragraph in section “Live cell actin profiles of cancer cells during sequential cyclic compression”: “The current device also allowed the response of individual cells at the interface between compressed and non-compressed regions in the periphery of the micro-piston to be studied, thus providing a degree of control of the localization of compression (see S2 Movie). As depicted in the video, parts of cells at the interface were subject to compression, while the rest remained uncompressed. Thus, the cell actin was deformed and damaged only partially in the compressed part depending on the magnitude of the applied pressure, whereas it remained more intact in the non-compressed part. This functionality which is unique to our platform by having control and compressed groups in the same microdevice, is useful to mimic partial cellular deformation and damage that can occur in vivo in presence of a localized mechanical loading.”

Cells under the periphery of the micro-piston are mainly changing shape in partial and some are being pushed away at compressed stage due to the physical presence of the micro-piston edges and accumulated cell debris towards the end of process when higher pressures were applied. For the cells that might be moving away from the area of the micro-piston, such as the cell at the top right corner of the representative actin-GFP video in S2 Movie (in the links here as well), which looks pulled its body part and changed its shape to move away from the contacting piston, the reviewer is right, such cells might get impacted by the pressurized pressure and sensing the mechanical stress loading and move away from the stress. However, to better understand this sensing and movement of the cells away from the stress (whether this is a mechanism for the cell to protect itself from the mechanical stress) we need to observe more cells doing that. On the other hand, we are more confident on that majority of the cells were deforming, being pushed when the piston was contacting and then buckling back when the stress was removed and remained there as partially compressed and damaged (at high pressures towards the end of the process) and partially intact at the control region (that was outside the micro-piston ROI as threshold area). 

Regarding movement of the cells away from under the micro-piston periphery, we suspect that confluency would be important as well. We expect that if the culture is confluent enough cells might not find enough space to move away, while in the less confluent cultures cells would be able to show their response better to decide whether to stay partially under the pressure or move away to a potentially available empty space nearby. We see the reviewer’s observation invaluable to be further investigated in such confluency-controlled samples under compression with tracking of the cells at the periphery of the micro-piston in the future work.

5) Are the differences in cell response (and death) seen in a given compression experiment related to cell cycle?

We have not looked at the impact of compression on cell cycle in our experiments in this manuscript. However, we have extensively studied impact of compression on cell viability, running multiple assays in our previous reports (e.g. Onal et al 2021 Frontiers in Physics). Pressures that cause cell death and thus decrease in cell viability were well characterized there and thus these were referenced here at the beginning of the section “Live cell actin profiles of cancer cells during sequential cyclic compression”: “For this, the actin-GFP transduced cancer cells were dynamically stimulated and compressed with various intermediate actuation pressures in a sequential cyclic manner, namely following a sequence from Mild (15.6–15.9 kPa) to Intermediate 1 (23.8–26.8 kPa), Intermediate 2 (37.8–41.7 kPa) and Severe (127.8–140 kPa) piston contact pressures. The first two ranges of this sequence fell within physiological and upper physiological pressure values and cell viability was high (on average 94 % for Mild and 77 % for Intermediate 1), while Intermediate 2 and Severe levels constitute hyperphysiological values and cell viability decreased (on average 43 % for Intermediate 2 and 20 % for Severe) [5].” 

Thus, our previous report showed mechanical stress induced cell damage and death towards higher pressures so categorized as Intermediate 2 and Severe pressures, providing the degree of the damage in detail in Onal et al 2021 Frontiers in Physics. Cell viability results for the applied pressure ranges were reminded from Onal et al, 2021, Frontiers in Physics to give an idea at what viability states the cells were at these pressure ranges while investigating live cell actin cytoskeleton under dynamic compression in the current manuscript.

For the cell cycle itself e.g. impact of mechanical compression on mitosis, or other cell cycle phases, this looks beyond the scope of current work, yet, worth looking at in future work. 

6) In Fig 6 b, it appears that there is a difference between the two "0 (CR)" values. Was this significant, and can the authors discuss the details of these samples?

All the statistical significance values for nuclear deformations in compressed and recovered cells after compression are noted in S2_File, newly added as supplementary file. 

S2 File caption “Statistical significance levels among all groups, obtained with Student's t-test run for nuclear deformations in compressed and recovered cells after compression presented in Fig 5 and Fig 6.” has been added. Fig 5 and Fig 6 captions have been also updated.

We added the following result and discussion for the difference between the two "0 (CR)" values: “There is statistically significant change in nuclei area between the control groups of zero-time and of 24 h-recovery after compression (see S2 File). Although these are not exactly the same control groups in the same conditions and we intend to compare the test group (e.g. zero-time or 24 h-recovery after compression) with their own control groups, we consider that the observed significant difference could be due to cell-cell signaling between the cells in compressed and control groups during that 24 h-recovery period after compression. In the groups of zero-time after compression, we did not allow control and compressed cells enough time to further interact with each other as they were fixed right after the compression.”

We further added the following discussion for the absence of statistical change in nucleus area between the control and compressed groups at zero-time after compression, in the associated paragraph: “While we expect a statistically significant change between the compressed and its control group at zero-time after compression at 20.8 kPa similar to the results observed in our previous report [5], we attribute the absence of such a significant change to cells being cultured on Matrigel-coating here as opposed to PLL-coating used in the previous experiments. Since Matrigel is expected to better represent the cancer cell microenvironment, it thus aided in cell growth into denser monolayers and formation of supporting extracellular matrix layer, that can alter sensing of the applied stress for the compressed cells.”

7) The meaning of "plasticity" should be defined more clearly. Does plasticity refer to mechanical lack of elasticity? If so, why would we expect this to be related to loss of actin-GFP? Also, plasticity would be reflected by a lack of recovery of the cell to its original state. Or are the authors simply analyzing biological processes that restore the cell after stress?

For analysis of actin-GFP signal we had done CTCF calculations and have now improved the graph to show the statical significances in Fig 4 in the revised manuscript. When there was statistically significant loss in actin-GFP signal and as seen on S2 Movie, actin disruption happens and that indicates for plastic response of the cells at those higher pressures. We added a discussion in the last paragraph of section “Live cell actin profiles of cancer cells during sequential cyclic compression”: “The observed disruption of actin cytoskeleton in these experiments indicates for cell plasticity during a cyclic compression in ovarian cancer when varying higher pressures occur in the microenvironment. There is need for more studies in literature to compare our results on cell plasticity after cyclic or dynamic compression applied sequentially up to high pressures. For cell actin deformation at milder pressures, however, a distinct change in actin cytoskeleton morphology was observed by Asem et al. for mesothelial cells, interacting with ovarian cancer cells, under compression that is statically applied at ∼3 kPa for 24 h in a bulk compression system. As a result, the interaction between ovarian cancer cells and mesothelial cells by enhanced formation of actin-based tunneling nanotubes (TNTs), and hence metastatic progression in ovarian cancer, was promoted under compression [7].” 

In the section Conclusions it has been also added that “While actin deformation at mild pressures (e.g. 15.6 kPa) in cyclic mode indicates for cell elasticity, the resultant actin disruption indicates for cell plasticity that can emerge when ovarian cancer cells are exposed to varying and higher pressures in a microenvironment.”

The reviewer is right on that plasticity refers to that mechanical lack of elasticity. If there is a lack of recovery of the cell to its original state due to, for instance, actin disruption, we think this is because of cell plasticity (plastic response of cells), as well. However, the cell being not recovering to its original state does not always refer to cell plasticity (mechanical lack of elasticity), especially at lower intermediate pressure values such as 20.8 kPa. The cell could be adapting to this pressure level and acquire a new phenotype while it is still elastic despite that there is an incomplete shape recovery to its original state.

We have gone through the entire manuscript and checked the parts where elasticity, plasticity and recovery words are used and revised the associated parts accordingly to prevent confusions in case of that the terms had been used interchangeably. 

Thus, wherever further slight changes were done, those are also highlighted in the revised sentences as in the following.

The following sentence in the last paragraph of Results and discussion has been revised:

“Furthermore, the data obtained in this work suggest that ovarian and likely other cancer cells can exhibit a plastic cell response to applied sequential cyclic compression.”

revised to 

“Furthermore, the data obtained in this work suggest that ovarian and likely other cancer cells can exhibit a varying cell response to applied sequential cyclic compression.”

In the paragraph starting as in the following in Introduction:

“In the presence of a mechanical stress, cells strive to restore their shape to maintain cell integrity after the removal of mechanical loading. However, cells may show a plastic response in the form of cytoskeletal bond ruptures and an incomplete cell shape recovery [1]. Meanwhile, cells produce an adaptive mechanism via these bond ruptures to reduce the mechanical cell stress and thus protect themselves against mechanical damage, while subsequently deforming under the mechanical load [1,8].”

3rd sentence has been revised to

“Meanwhile, cells produce an adaptive mechanism to reduce the mechanical cell stress and thus protect themselves against mechanical damage, while subsequently deforming under the mechanical load [1,10].” 

Plastic word has been removed in the following sentence in Introduction:

“Using a single cell microfluidic compression platform, their application resulted in full recovery and thus no permanent plastic deformation in MCF10A normal breast epithelial cells after compression [2].”

Plastic word has been removed in the following sentence in Results and discussion section “Actin cytoskeleton and nuclei profiles of cancer cells after compression and recovery”:

“No statistical difference in cell height before and after compression was observed, indicating that the breast epithelial cells did not exhibit permanent plastic deformation.”

8) Similarly, the meaning "recovery" should be defined more clearly. The authors write "The extent of recovery of compressed cells, inferred by imaging cell membrane bulges and actin cytoskeleton and measuring the shape descriptors of cell nuclei, can give insights into the plasticity of cancer cells." However, there are no measurements of cell membrane bulges, and no spatial analyses of actin organization. How do we know the cells have "recovered"?

Our response to this question is in agreement with the response to previous (7th) question of the reviewer. 

Indeed, we used the recovery word in this manuscript for the investigation of cell response after compression, whether they fully restore their shape or not, allowing cells enough time. Thus, it refers to a process to investigate what happens to cells after being exposed to compression. 

To prevent confusion the sentence in the last paragraph of Introduction has been revised to a clear form: “The extent of recovery of compressed cells can give insights into the cell integrity and adaptation of cancer cells to restore their shape or acquire new one when exposed to mild or upper mild compressive stress at physiological levels.”

Cell membrane bulges formed at compressed stages are distinct in S1 Movie which include huge amount of information per time-lapse imaging video. While membrane bulges in cells observed in phase contrast live cell imaging are fixed directly after compression and distinctly appeared as increased actin signal at cell edges when fluorescently stained, their further analysis out of time-lapse videos would require an advanced and automated image analysis method as label-free and high-content. Spatial analyses of actin organization (in addition to CTCF calculations) would require imaging data taken at higher resolution, via a super resolution microscopy for instance. 

9) Other groups have published similar pressure-actuated devices for compressing cells, so there should be more discussion of these studies and comparison with the presented technology. In general, a more extensive literature review on in vitro analyses of cancer cell solid stress would be useful.

The following paragraph that was in Results and discussion has been moved to Introduction and slightly revised:

“To date, dynamic compression experiments in literature include frequencies of 0.1 – 30 Hz with an applied stress of 5.1, 9.3, 12.9 and 18.7 kPa on breast cancer cells over short durations (30 – 300 s), as used by Takao et al., who observed a mixed mode of apoptosis and necrosis-dominant cell death in mechanically compressed monolayers in a bulk compression platform [18]. In contrast, work by Novak et al. used a frequency of 0.05 Hz for cyclic compression applied at 3.9 – 6.5 kPa for 24 hours [3]. They observed an increase in proliferation capacity and decrease in apoptosis for OVCAR3 and OVSAHO ovarian cancer cells cultured in 3D hydrogel components in a bioreactor at millimeter scales. In another example, Ho et al. applied cyclic compression by alternating between external pressures of 10 and 15 psi (68.9 and 103.4 kPa) at 0.5 Hz for 6 minutes [2]. Using a single cell microfluidic compression platform, their application resulted in full recovery and thus no permanent plastic deformation in MCF10A normal breast epithelial cells after compression [2]. In our work we add to these examples by using the capabilities of our compression platform to apply cyclic compression with a low frequency in microfluidic settings. In particular, we utilize this functionality to provide a first investigation of the effects of cell compression, deformation and recovery after compression on SKOV-3 ovarian cancer cells. As part of this we provide evidence that the amount of the pressure and duration of the application clearly affects morphology during cell compression and recovery. While compression frequencies in an in vivo tumor microenvironment are not yet fully known, cyclic compression is important for maintenance of cell culture during mechanical compressions in an in vitro experimental setup, in particular as it enhances media flow underneath the micro-pistons.”

10) Given the relatively rapid application and removal of compressive forces, there must be some interesting fluid dynamics happening between the cells and the piston. Can the authors estimate the changes in fluid pressure and shear stress on the cells? This could also affect the cell biology.

This is very good point. We have not particularly looked at the shear stress effect. We do not have horizontal, continuous flow on the cells in the channel, which means the cultures were in static media, but while moving the piston at vertical direction, the media also displace and that might create a negligible shear stress. However, we believe that the solid stress that we applied with physically sharply contacting surface of the micro-piston dominates the effect of compressive stress on cells, not that of shear stress. Considering other stresses from the very beginning, we applied the pressure gradually on the cells as demonstrated at the beginning of S1 Movie while increasing the pressure at a gradual rate shown on the pressure ladder on sensor reading graphs. Cells were experiencing the changing pressure mostly at vertical directions in the form of compression. 

There could be hydrostatic pressure which also translate into compressive stress while deflecting the membrane, so we expect that hydrostatic pressure to be all around the channel. Comparing the viability results among the different shapes and among different compression states, studied extensively in Onal et al. 2021 Frontiers in Physics, the cells in the control regions around the micro-piston were intact and alive throughout the applications as well as that the cells under micro-piston responded according to the applied pressure. This shows that other stress factors seem to not have an influence on cells in terms of viability. 

If we had a horizontal cell compression setup as Paggi et al. (2020, Sensors Actuators B: Chemical) had in their design with a vertical membrane adjacent to the cell culture chamber that is deflected at horizontal direction, we would need to be looking at multi-modality of the forces and the combination of compressive and shear stress as they did in their work. However, in our design, the membrane is located horizontal to the cell culture chamber, and we have a piston to physically contact the cells once compressed. Thus, we deflect the membrane vertically and the forces are mainly in compressing mode at the vertical direction in the cell culture chamber. 

Nevertheless, we believe that the reviewer made an interesting point here, worth looking at combination of shear stress and compressive stress effects on cells further in the scope of a next piece of work within cell compression field.

11) Further justification should be provided for the cyclic stress schedule chosen. Musculo-skeletal tissues would be expected to experience such forces, but this is not clear for ovarian cancer.

The cyclic stress schedule (i.e. 5-min compressed stage followed by 5-min rest stage in each cycle) was chosen to allow cells enough time during each stage to complete bulge formation or recovery, processes which were considered important for the investigation of cell deformation and recovery after compression. Membrane bulge formation and recovery processes are clearly observable in S1 Movie. This was explained in the following paragraph in section “Comparison of the experimental and computational results in micro-piston device”

“While inferring these results, the cell compression process was imaged in detail in Matrigel-coated microchannels at a speed of 1 frame per 100 ms, as shown in S1 Movie. As illustrated by this time-lapse video, the platform allowed for the temporal evolution of the dynamic cell compression to be investigated in each cycle while the piston was contacting cells and applying compression. Distinct cell membrane bulges could be observed to form during such compression at piston contact pressure of 15.6 kPa and recovery of cell membranes from these bulges was visible when pistons were lifted off during rest stages. Although the platform is capable of temporal control of cell compression (e.g. to decrease, increase or remove the pressure at any certain time interval), duration of each stage in a cycle was set to 5 minutes, corresponding to a 5-min compressed stage followed by 5-min rest stage in each cycle. This was chosen to allow cells enough time during each stage to complete bulge formation or recovery, processes which were considered important for the investigation of cell deformation and recovery after compression. Sensor readings in S1 Movie showed that the micro-piston actuated according to externally applied pressures with reliable control and could be operated in short durations gradually or suddenly when needed.”

Further explanation and discussion have been also added in this section as in the following paragraph:

“Cyclic compression in the study of Ho et al. was applied only for 6 minutes in total [2], significantly shorter than the entire duration of a cyclic compression in this work, and results presented here suggest that cell deformation at compression stage and recovery at rest stage depends on the temporal length of the compression on cells. Particularly observable in S1 Movie, cell membrane bulge formation slowly developed during the compression stage while cells were responding to contact pressure at the interface with the PDMS piston. This was despite the fact that the piston was actuated onto the cells rapidly during cyclic compression as directed by the external pressure controller. As shown by the pressure sensor readings in the same movie, PDMS membrane and piston were consistently responding to amount, duration and mode of the applied pressure. Thus, the gradual deformation of cells and bulge formation during compression stages must originate in the response type of the cells to mechanical load when in solid contact with the PDMS piston. During rest stages, where the compression was lifted, cell bulges did not fully recover right after compression was removed, suggesting that the cells require a certain amount of time to recover compressed cellular parts (S1 Movie). Based on these observations of temporal evolution of the cell compression application, and comparison of application durations in this study with the study by Ho et al. [2], we hypothesize that this recovery also depends on the duration and number of the cycles.”

Regarding sources of compression ovarian cancer cells are exposed to, further information has been added in Introduction:

“While some cells experience permanent plastic deformation after a repetitive mechanical tensile loading and unloading, the impact of such repetitive compression on deformation of cells remains largely unknown [1,2]. As such, the ability to apply cyclic compression is crucial for any experimental setup aimed at the study of mechanical compression taking place in cell and tissue microenvironments [3–5]. For instance, in ovarian cancer, cells are exposed to chronic compressive stress from different sources such as tumor growth, displacement within stromal tissue and hydrostatic pressure out of ascitic fluid in peritoneal cavity [3,6]. Ascites-induced compression exists in the peritoneal microenvironment due to the ascites volume that can reach >2L and increases ovarian cancer cell adhesion to peritoneum, shown by Asem et al. using in vivo assay [7]. This peritoneal microenvironment is continuously affected by the movements of the surrounding organs, musculoskeletal dynamics, and gravity [6]. Thus, ovarian cancer cells might experience cyclic compression profiles during chronic exposure to compression from different sources and due to anatomical location of the ovary. Mimicking such compression profiles and physiological pressure values in an in vitro dynamic compression system is necessary to further study impact of compressive stress in cancer.

Compressive stress is estimated to reach 18.9 kPa for human tumors and can exceed 20 kPa based on the experimental data from murine tumors, while those values can be even higher in situ with the impact of the surrounding extracellular matrix in tumor microenvironment [3,8]. Compared to other cancer types such as breast and metastatic bone niches, compression studies have been limited for ovarian cancer, including for investigating the impact of various amounts of applied pressures [6]. Pressure values existing in literature that are applied for ovarian cancer have been in a range from ∼3 kPa to 6.5 kPa [3,7,9]. Ovarian cancer cell response at higher applied pressures that can occur in the body need further investigation.”

Furthermore, it was noted in our manuscript that “While compression frequencies in an in vivo tumor microenvironment are not yet fully known, cyclic compression is important for maintenance of cell culture during mechanical compressions in an in vitro experimental setup, in particular as it enhances media flow underneath the micro-pistons.”

Reviewer #2: The following are the attached Comments for Authors.

In this manuscript, authors characterized changes and recovery in actin fluorescence signal and nucleus shape after applying cyclic compression to ovarian cancer cells. However, there are several major issues with the current manuscript: 1) the introduction lacks information and citations of previous work and does not address the importance of this work, i.e., why apply cyclic compression on ovarian cancer cells; 2) Key parts of the experimental design lack rationale, i.e. why 5 min pressure 5 min recovery, how large is the piston/cell contact area, variability of the pressure within contact area, etc. 3) Several results are questionable, as detailed the following comments; and 4) Several results lack information on significance tests and number of independent studies. As such, the reviewer recommends rejection.

1. The introduction is not clear what the similarities and differences are between chronic and cyclic compression, and why cyclic compression is important to study in ovarian and other cancer cells.

Chronic compression in this manuscript and in literature was meant to be that ovarian cancer cells are under a dynamic compression from different sources. 

Chronic word was removed from the following sentence to prevent confusion, as it was meant to be that higher pressure amounts (total perceived stress which can originate from chronic exposure to compression from different sources) and profiles that can be experienced by ovarian cancer cells need to be mimicked in an in vitro dynamic compression system and the impact on cells need to be studied further:

“Mimicking such chronic compression profiles and physiological pressure values in an in vitro dynamic compression system is necessary to further study impact of compressive stress in cancer.”

The first two paragraphs of the Introduction have been revised as follows:

“While some cells experience permanent plastic deformation after a repetitive mechanical tensile loading and unloading, the impact of such repetitive compression on deformation of cells remains largely unknown [1,2]. As such, the ability to apply cyclic compression is crucial for any experimental setup aimed at the study of mechanical compression taking place in cell and tissue microenvironments [3–5]. For instance, in ovarian cancer, cells are exposed to chronic compressive stress from different sources such as tumor growth, displacement within stromal tissue and hydrostatic pressure out of ascitic fluid in peritoneal cavity [3,6]. Ascites-induced compression exists in the peritoneal microenvironment due to the ascites volume that can reach >2L and increases ovarian cancer cell adhesion to peritoneum, shown by Asem et al. using in vivo assay [7]. This peritoneal microenvironment is continuously affected by the movements of the surrounding organs, musculoskeletal dynamics, and gravity [6]. Thus, ovarian cancer cells might experience cyclic compression profiles during chronic exposure to compression from different sources and due to anatomical location of the ovary. Mimicking such compression profiles and physiological pressure values in an in vitro dynamic compression system is necessary to further study impact of compressive stress in cancer.

Compressive stress is estimated to reach 18.9 kPa for human tumors and can exceed 20 kPa based on the experimental data from murine tumors, while those values can be even higher in situ with the impact of the surrounding extracellular matrix in tumor microenvironment [3,8]. Compared to other cancer types such as breast and metastatic bone niches, compression studies have been limited for ovarian cancer, including for investigating the impact of various amounts of applied pressures [6]. Pressure values existing in literature that are applied for ovarian cancer have been in a range from ∼3 kPa to 6.5 kPa [3,7,9]. Ovarian cancer cell response at higher applied pressures that can occur in the body need further investigation.”

The following sentence that was in Results and discussion has been moved to Introduction: “While compression frequencies in an in vivo tumor microenvironment are not yet fully known, cyclic compression is important for maintenance of cell culture during mechanical compressions in an in vitro experimental setup, in particular as it enhances media flow underneath the micro-pistons.”

2. Page 2, line 9-11: “Such compression is estimated to reach 18.9 kPa for human tumours and can exceed 20 kPa based on the experimental data from murine tumours [3, 7].” Given the wide range of compressive stresses in different tumor models & species, the authors should specify the physiological range of compression for ovarian cancer.

Regarding ovarian cancer, compression sources and current studies in literature and expected compression values are now explained in the first two paragraphs of the Introduction in our revised manuscript. Further compression values and findings for ovarian cancer cell compression based on the ovarian cancer cell response to the applied pressures in our study are presented in Results and discussion in this manuscript.

3. The introduction is not clear about the similarities and differences between chronic and cyclic compression, and why cyclic compression is important to study in ovarian and other cancer cells.

Same as the 1st question. 

4. The innovations in the microdevice authors developed are not clear. What has been developed in the past and what is new should be summarized in the Introduction.

Last two paragraphs of Introduction included information on what has been developed in past and what is new. Now the two paragraphs are revised, and the last paragraph has been further improved to better reflect on what is new. 

“Thus, with the purpose of applying compression in a cyclic, dynamic and controlled manner for the investigation of cell deformation and recovery, we have recently developed a robust cyclic compression microfluidic method based on a flexible microdevice. This platform provides extensive control of the amount, duration, and mode within a pressure profile. Fabrication and characterization of the multilayer microfluidic platform, and the observation of directional growth of cells, viability and mechanical cell lysis as end point assays under static state and pressurized states within this platform were demonstrated previously [4,5].”

“In this paper, we further extend the reproducibility and repeatability validation of our flexible microdevice-based compression method via comparisons of experimental and computational piston actuations for independent microdevices with different membrane thicknesses. We then use the microfluidic platform for the in vitro application of cyclic cell compressions that mimic biologically relevant compression profiles occurring in cellular microenvironments. In particular, we demonstrate applicability of the platform for the chronic exposure of ovarian cancer cells to repetitive compressive stress. As such, the current work extends the use of the platform to the application of cyclic compressions at and beyond physiological pressure values in a sequential fashion to study dynamic biomechanical processes by recording GFP-tagged actin dynamics of live cells under compression. In relation to our previous work which discussed initial findings on cellular deformations observed at cyclically applied ascending pressures, the current work also expands applicability by providing a more physiologically relevant setting in form of a hydrogel coating within micro-piston device. We showcase data of cellular deformations and recovery during and after cyclic compression at mild (e.g. 15.6 kPa) and upper mild (e.g. 20.8 kPa) physiological pressure values, obtained via end point assays of actin and nuclei deformations in compressed cells at zero time or at 24 h-recovery after compression, showing flexibility and novel use of our microdevice for the applications of not only cell compression but also recovery. We demonstrate that the platform can control the strength and duration of cyclic compression, thus providing a powerful new tool for the study of mechanobiological processes, by which cell deformation, cytoskeletal and nuclear changes and recovery in cells exposed to such compression can be readily captured.”

Revised to 

“Thus, with the purpose of applying compression in a cyclic, dynamic and controlled manner for the investigation of cell deformation and recovery, we have recently developed a robust cyclic compression microfluidic method based on a flexible microdevice. Fabrication and characterization of the multilayer microfluidic platform, and the observation of directional growth of cells, viability and mechanical cell lysis as end point assays under static state and pressurized states within this platform were demonstrated previously [4,5]. The developed platform was characterized to provide extensive control of the amount, duration, and mode within a pressure profile [5].

In this paper, we further extend the reproducibility and repeatability validation of our flexible microdevice-based compression method via comparisons of experimental and computational piston actuations for independent microdevices with different membrane thicknesses. We then extend the use of the platform to the application of cyclic compressions at and beyond physiological pressure values in a sequential fashion to study dynamic biomechanical processes by recording GFP-tagged actin dynamics of live cells under compression. To the best of our knowledge, this is the first demonstration of how live cell actin behave under dynamic compression applied over time in cyclic mode and sequentially at varying pressures in ascending order. Using time-lapse imaging, we thus show a sequence of actin deformation and disruption events based on the amount of the applied pressure in a flexible microdevice. The current work also expands applicability by providing a more physiologically relevant setting in form of a hydrogel coating within micro-piston device as shown in Fig 1. We showcase data of cellular deformations and recovery during and after cyclic compression at mild (e.g. 15.6 kPa) and upper mild (e.g. 20.8 kPa) physiological pressure values, obtained via end point assays of actin and nuclei deformations in compressed cells at zero time or at 24 h-recovery after compression. These experiments further prove flexibility and novel use of our microdevice for the applications of not only cell compression but also recovery. Differences observed between compressed cells fixed at zero time or after 24 h-recovery suggest that SKOV-3 cells exhibit deformations at the time of the compression, a proposed mechanism cells use to prevent mechanical damage. The extent of recovery of compressed cells can give insights into the cell integrity and adaptation of cancer cells to restore their shape or acquire new one when exposed to mild or upper mild compressive stress at physiological levels. As demonstrated with SKOV-3 ovarian cancer cells, biomechanical responses of cells to sequential cyclic compression and during recovery after compression can be revealed in a flexible microdevice where cell deformation, cytoskeletal and nuclear changes and recovery in cells exposed to such compression can be readily captured.”

5. References supporting nucleus as mechanosensory are lacking. Kirby (2018) Nature Cell Biology, Fu (2012) Lab on a Chip, etc. Additionally, past findings on nucleus deformation in ovarian cancer cells and its functional effects should be introduced as background to this study

We thank the reviewer for letting us know about these references. We have looked at and cited them in the respective part of the Introduction “In itself, the nucleus is a mechanosensitive organelle [11-13] …..”, along with Nava et al (2020) Cell. 

We are not aware of ovarian cancer cell-specific cell nucleus deformation by mechanical compression and its functional affects if such study results exist in literature. However, we have added a literature search from various cancer types, supporting background and motivation of this study:

“Morphological changes in nuclear structure, such as increased nuclear size, irregular shape by grooving, convolutions and invaginations of the nuclear envelope and altered organization by disturbed chromatin distribution, are commonly used cancer markers by pathologists [15]. Mechanical compression through confinement aid in mechanical adaptation of the cell as shown with Hela-Kyoto cancer cells in which stretching of the nuclear envelope and upregulation of actomyosin contractility were observed [16,17]. Furthermore, irregular nuclear morphology of cancer cells aligns with altered expression of nuclear envelope proteins such as lamins A/C or lamin B in cancers as detailed by Denais et al., which has the capacity to promote metastatic processes [15]. Deformation of the large and stiff nucleus is required during the passage of metastatic cancer cells through the tight interstitial space and narrow capillaries in the microenvironment. Thus, mechanical properties of the cell nucleus, such as deformability and adaptation under compression, and its connection to the cytoskeleton may play a major role in cancer metastasis [15].”

6. Page 7, Fig 2(b): it is not clear which are experimental data and which are simulated pressures on the graph, and how well they match quantitatively.

Fig 2(b) was plotted for membrane thickness (on x-axis) vs. pressure sensor readings (on y-axis). Those values (both on x- and y-axis) are from experiments that resulted in a similar piston contact pressure. Before experiments those were computationally simulated as shown in Fig 2(a) to predict what external pressures need to be applied at different thicknesses to be able to get a similar piston contact pressure. Thus, while Fig 2(a) shows simulations, Fig 2(b) shows comparison of experimental values that were applied based on simulations. Because R2 resulted to be high among independent experiments, giving a good correlation along different thicknesses and applied external pressures, to obtain a similar piston contact pressure calculated based on simulations, we think the model is robust to predict what external pressures should be applied when there is a variation in membrane thickness.

Caption of Fig 2(b) has been revised as in the following: 

“(b) Correlation of independent experiments with the pressure sensor read for the externally applied pressure based on simulations run for the corresponding membrane thicknesses, resulting in a similar piston contact pressure.”

Revised to

“(b) Correlation among independent experiments with the pressure sensor read for the externally applied pressures applied based on simulations in (a) run for the corresponding membrane thicknesses, resulting in a similar piston contact pressure.”

The 2nd sentence in the following part at the end of the 1st paragraph in section “Comparison of the experimental and computational results in micro-piston device” has been revised to make it clear that the calculated R2 was among independent experiments with different membrane thicknesses operated under different amounts of externally applied pressures that were predicted with simulations: 

“A summary of the compression applied to cancer cells via different membrane thicknesses for externally applied pressures and the resulting maximum piston contact pressures is shown in Table 1. Fig 2b further shows that comparable piston contact pressures were achieved by mildly compressing cells with applied external pressures, which were experimentally matching the pressure amounts in simulations (R2 = 0.9967). ……” 

Revised to

“……. Fig 2(b) further shows that comparable piston contact pressures were achieved by mildly compressing cells with applied external pressures (R2 = 0.9967), which were experimentally applied according to the pressure amounts in simulations Fig 2(a). …….” 

7. Page 6, line 198: “maximum piston contact pressure”…the authors should include how much variation in pressure exists for the entire area under the piston, and address how this variation may affect results

New supplementary figure, S2 Fig in the revised manuscript, has been added to illustrate the distribution of contact pressure under piston. It is referenced in the manuscript text in “Computational modelling used to calculate the maximum piston contact pressure (S2 Fig) was further validated experimentally using independent microdevices with different membrane thicknesses to enhance reproducibility and repeatability of our flexible microdevice-based compression method.” 

Explanation for the piston contact pressure illustrations has been added in figure caption as follows: “S2 Fig. Illustrations of the piston contact pressure area by the symmetrical half-cell of the piston and cell-culture chamber in the computational model. They show the distribution of contact pressure (in N/m2) under piston for the selected applied external pressures (in mbar). Red dashed line of model symmetry drawn on the illustrations at different pressures point out the middle side of the piston. The contact pressure is slightly higher at the edges of the piston towards the side channel walls where the attached membrane distance between piston and side wall is shorter. Thus, there seem to be some variation due to how the piston bottom gets squashed in these sides of the piston, but it is minimal compared to the overall area. This is demonstrated by the color being relatively even over the whole area, especially for externally applied lower pressures such as 0-420 mbar.”

8. Page 8, line 234: It is not clear why authors chose 5 min pressure stage followed by 5 min rest stage. Experimental characterization of nucleus deformation recovery time in relationship to pressure AND/OR theoretical support is needed.

The cyclic stress schedule (i.e. 5-min compressed stage followed by 5-min rest stage in each cycle) was chosen to allow cells enough time during each stage to complete bulge formation or recovery, processes which were considered important for the investigation of cell deformation and recovery after compression. Membrane bulge formation and recovery processes are clearly observable in S1 Movie. This was explained in the following paragraph in section “Comparison of the experimental and computational results in micro-piston device”

“While inferring these results, the cell compression process was imaged in detail in Matrigel-coated microchannels at a speed of 1 frame per 100 ms, as shown in S1 Movie. As illustrated by this time-lapse video, the platform allowed for the temporal evolution of the dynamic cell compression to be investigated in each cycle while the piston was contacting cells and applying compression. Distinct cell membrane bulges could be observed to form during such compression at piston contact pressure of 15.6 kPa and recovery of cell membranes from these bulges was visible when pistons were lifted off during rest stages. Although the platform is capable of temporal control of cell compression (e.g. to decrease, increase or remove the pressure at any certain time interval), duration of each stage in a cycle was set to 5 minutes, corresponding to a 5-min compressed stage followed by 5-min rest stage in each cycle. This was chosen to allow cells enough time during each stage to complete bulge formation or recovery, processes which were considered important for the investigation of cell deformation and recovery after compression. Sensor readings in S1 Movie showed that the micro-piston actuated according to externally applied pressures with reliable control and could be operated in short durations gradually or suddenly when needed.”

Further explanation and discussion have been also added in this section as in the following paragraph:

“Cyclic compression in the study of Ho et al. was applied only for 6 minutes in total [2], significantly shorter than the entire duration of a cyclic compression in this work, and results presented here suggest that cell deformation at compression stage and recovery at rest stage depends on the temporal length of the compression on cells. Particularly observable in S1 Movie, cell membrane bulge formation slowly developed during the compression stage while cells were responding to contact pressure at the interface with the PDMS piston. This was despite the fact that the piston was actuated onto the cells rapidly during cyclic compression as directed by the external pressure controller. As shown by the pressure sensor readings in the same movie, PDMS membrane and piston were consistently responding to amount, duration and mode of the applied pressure. Thus, the gradual deformation of cells and bulge formation during compression stages must originate in the response type of the cells to mechanical load when in solid contact with the PDMS piston. During rest stages, where the compression was lifted, cell bulges did not fully recover right after compression was removed, suggesting that the cells require a certain amount of time to recover compressed cellular parts (S1 Movie). Based on these observations of temporal evolution of the cell compression application, and comparison of application durations in this study with the study by Ho et al. [2], we hypothesize that this recovery also depends on the duration and number of the cycles.”

9. Page 9: Panels a-c in Figure 3 were not referenced in the text.

Panel a of Figure 3 has been referenced in section “Live cell actin profiles of cancer cells during sequential cyclic compression”: “……., for these experiments SKOV-3 ovarian cancer cells were transduced on-chip with CellLight Actin-GFP, BacMam 2.0 baculovirus to visualize the dynamic biomechanical processes during compression (Fig 3(a)).” and panels b-c referenced in “…… and actin-GFP signals of the compressed cells were captured (Fig 3(b-c)).”

10. Page 11, figure 4 should include levels of significance for intended comparison.

Significance levels for intended comparison have been added onto the graph in Figure 4. Rest of all the significance test results (p-values) are in S1_File excel file. Figure 4 caption has been also updated:

“….. Student's t-test was used to determine the statistical significance of live cell actin deformation across increasing pressure ranges. Continuous horizontal bars show significant differences between compressed cell groups under micro-pistons for images taken at the time of compression at the indicated pressure ranges. Dashed horizontal bars show significant differences between the cell groups at rest under micro-piston for images taken after compression stages at the corresponding pressure ranges of the cycles. Results represent at least three independent experiments.”

11. Page 9, line 295-297: It is not clear the lower actin signal from higher pressure groups is due to higher pressure or accumulated stress from the sequential testing. Independent experiments should be carried out for each pressure..

This is very good point. We have not run such independent experiments for each pressure. From the very beginning we deliberately aimed for sequential cyclic compression profiles in a dynamic manner that can mimic varying pressures in the same microenvironment that can be applied on the same sample. The platform is suitable to run individual steps of sequential cyclic compression.

We revised Fig 4 to show statistical significance levels and those were given in detail in S1 File for all groups, and further explained and discussed in text as highlighted in the revised manuscript in section “Live cell actin profiles of cancer cells during sequential cyclic compression”. Based on these, there are no significant differences at rest stages (after completion of the compression stages) until Intermediate 2 level pressures which are considered high. Thus, although cells deform throughout 1-hour cyclic compression applied at Mild pressures, loss of actin signal was not statistically significant between 1st cycle (beginning of 1-hour) and last cycle (at the end of 1-hour). Then pressure was risen to Intermediate 1 to Intermediate 2 and Severe pressures in 30 s at each increase (gradual rate) and cells were compressed for up to 2 min at each pressure level. Those are relatively gradual and short durations at those higher pressures applied sequentially, following cyclic mode mild pressure that was applied for 1 hour. Statistically significant change was seen at rest stage after Intermediate 2 level of pressures at which we think there was a mechanical impact due to high pressure. Such an impact is statistically significant again at rest stage after applied Severe pressures compared to all other levels.

“…….. Similar as to the sequence shown in Fig 1(d), first, a pressure amount ladder was formed up to a pressure in the mild compression range, represented by an externally applied pressure of 350 mbar and resulting in a piston contact pressure of 15.6 kPa, during which the piston reached the cells at the bottom surface and cells started to distinctly deform. Once this was observed, cyclic compression was applied at mild pressure for a total of 1 hour, with cells compressed for 5 min at 15.6 kPa, followed alternately by a rest for 5 min at 0 kPa. To track cell response, actin-GFP signals of the compressed cells were captured before compression, as well as during the first and last cycles of the cyclic compression stage. At the end of this sequence, the externally applied pressure was sequentially increased first to 370 mbar, then to 400 mbar and finally 640 mbar, resulting in the respective piston contact pressures of 23.8 kPa (Intermediate 1), 37.8 kPa (Intermediate 2) and 140 kPa (Severe). During this stage, cells were compressed for up to 2 min at each pressure (Fig 1(d)) and actin-GFP signals of the compressed cells were captured (Fig 3(b-c)). Throughout the application of sequential cyclic compression, cells were allowed to adjust to each next higher applied pressure. As shown in the pressure amount ladder in Fig 1(d), pressure was applied gradually in 30 s among rising steps of the ladder until the piston reached the focal plane of the cells and started cell deformation. After observing that cells were distinctly deformed at a mild compression level but not to a disruption or lysis level, compression was set to be in cyclic mode. Although the pressure was changing suddenly along the cycles as per the settings of the selected cyclic mode in the pressure controller, cells were not damaged when compressed as they resulted in being resistant at this physiological value. Next, pressure application was continued with gradual increase from 350 to 370 mbar, from 370 to 400 mbar and from 400 to 640 mbar in 30 s for each, as shown in Fig 1(d). Thus, cells could have the time to adjust during the pressure increase and show their intrinsic response to the pressurized stage of each compression level.”

Further explanation and discussion after the CTCF calculations, have been also added in the next two paragraphs in text following the paragraph above in section “Live cell actin profiles of cancer cells during sequential cyclic compression”. Overall, the revised/added discussions (highlighted in the revised manuscript) include that the lower actin-GFP signal and disruption of actin cytoskeleton in these experiments indicates for cell plasticity during a cyclic compression in ovarian cancer when varying higher pressures occur in the microenvironment.

The following sentence has been also added to Conclusions: “While actin deformation at mild pressures (e.g. 15.6 kPa) in cyclic mode indicates for cell elasticity, the resultant actin disruption indicates for cell plasticity that can emerge when ovarian cancer cells are exposed to varying and higher pressures in a microenvironment.” 

Although we think that a mechanical compression process is dynamic inside the body and it can develop in a cumulative way (with increasing pressures over time), and thus our sequential cyclic compression can mimic such process, we also think that the reviewer is right on that actin profile investigation could be also done straight on a particular higher pressure in a discrete way, should such sudden high pressures emerge in the body. 

For cell viability investigation in Onal et al 2021 Frontiers in Physics, we extensively tried on multiple devices applying compression at 640 mbar (Severe pressure), and checked for viability at initial time of the compression and after 1 hour of continuous application at 640 mbar and then on other samples we tried sequential cyclic compression up to 640 mbar with the stage of 640 mbar itself being for 2 min. At all applications, cell viability result was similar for samples exposed to 640 mbar indicating that cell response was due to the mechanical compression at this high pressure rather than its independent experiment or sequential cyclic compression (that developed with varying pressures up to 640 mbar similar to profiles that were applied in live cell actin investigation here) or the durations of the applications. 

Thus, similar to those compression assays and cell viability response in Onal et al 2021 Frontiers in Physics, for high pressures we expect that actin disruptions still happens when the applications at each of Intermediate 2 or Severe level pressure ranges are done independently and actin-GFP signal is measured, in similar way as it happened in sequential cyclic compression shown in the current work. 

12. Page 9, Line 304-306: Were these “resistant” cells in contact with the piston? Characterization of the interface between the piston and the cell substrate AND/OR other metrics (ie., nucleus shape/area) is needed to rule out artifact.

The resistant cells remained on the glass surface. 

The relevant sentence has been slightly revised to prevent confusion: “Interestingly, a few cells on the glass surface usually remained resistant to the increasing pressure ranges, as evidenced by their retained fluorescent signal while compressed under the micro-piston.”

The following clarification has been also added: “During the entire process, wide-field imaging was used and the focus was kept on one focal plane on the glass surface where cells were initially adhered to. At compressed stages, glass and PDMS piston surfaces were in contact and at the same focal plane. At the rest stages when pressure is released and piston is retracted back, to be able to gain the signal from piston surface there would be need to bring the focal plane to the suspended piston level that is on average 108 �m above the glass surface. In the region under the micro-piston, there seem to be variations in gained actin signal between the compressed and rest stages of each cycle (Fig 4), but these were not statistically significant (p>0.05, see S1 File). Thus, no significant amount of cell artifacts was attached to piston surface.”

In terms of background (light artifacts), this was extracted from the images as shown in the equation in section “Imaging and data analysis”. Furthermore, the calculated actin signals were compared between the region under piston and control region around the piston. Actin signal in the control regions were also compared to each other across the cycles and remained statistically insignificant (S1 File) as it was indicated in the manuscript. The following discussion has been added to the text in the same paragraph as above in Results and discussion: “Thus, the GFP-tagged actin signal in control regions was robust during the entire process, as expected [27]. This can be reference to that fluorescence changes in the region under micro-piston (i.e. decrease in actin signal) was due to the impact of mechanical compression.” 

13. Page 10-11: the authors should consider discussing impact of the observed disruption of actin /no recovery due to cyclic compression in ovarian cancer and the relevance to cell plasticity

The following discussion has been added in section “Live cell actin profiles of cancer cells during sequential cyclic compression”:

“The observed disruption of actin cytoskeleton in these experiments indicates for cell plasticity during a cyclic compression in ovarian cancer when varying higher pressures occur in the microenvironment. There is need for more studies in literature to compare our results on cell plasticity after cyclic or dynamic compression applied sequentially up to high pressures. For cell actin deformation at milder pressures, however, a distinct change in actin cytoskeleton morphology was observed by Asem et al. for mesothelial cells, interacting with ovarian cancer cells, under compression that is statically applied at ∼3 kPa for 24 h in a bulk compression system. As a result, the interaction between ovarian cancer cells and mesothelial cells by enhanced formation of actin-based tunneling nanotubes (TNTs), and hence metastatic progression in ovarian cancer, was promoted under compression [7].” 

14. Page 13, figure 5 b-d needs to include details of significance tests conducted and the significance levels detected.

Intended comparison is shown on the figures. Figure caption is revised. 

“Fig 5. ……. Student’s t-test was used to determine the statistical significance of nuclear deformations in compressed and recovered cells after compression.”

Rest of all the significance test results (p-values) are in S2_File excel file, newly added supplementary file.

Same revisions have been applied for Figure 6 in caption and S2_File. 

15. Page 12, line 352: it is not clear what specific heterogeneity the authors mean here

Heterogeneity word was clarified to prevent confusion by revising the relevant sentence into “This divergence might emerge from the variations in decrease of cell volume under compression and nuclear envelope wrinkling and ability of each cell to recover these changes after mild compression [13,28].”

16. Page 12, line 391: minute differences (?)

“Minute” word in “……minute differences in cell behaviour…..” has been removed from line 391 and from the Abstract, to prevent any confusion. It was meant that the pressure change was minute (e.g. ~5 kPa difference between Fig 5 and Fig 6) but there were changes in cell response.

It was also removed from the sentence in the section “Actin cytoskeleton and nuclei profiles of cancer cells after compression and recovery”: “The phenotypical response of compressed cells during recovery after compression at 20.8 kPa differed observably from that for 15.6 kPa, demonstrating the capability of the flexible microdevice to capture minute differences in cell behaviour after being compressed at various pressures in a physiologically relevant manner.”

17. Page 13: in Fig 5(a) 15.6 kPa (o time), actin signal looks higher in compressed area compared to cells in control area? The authors should also explain the unevenness of brightness in phase contrast images

The distinct actin deformation by the increased fluorescence signals at the edges of the cells were shown on Fig 5(a) by representative arrows as it was noted on figure caption: 

“Representative arrows (white) show the distinct actin deformation by the increased fluorescence signals at the edges of the cells in the compressed groups under the micro-piston at zero time after compression.”

An explanation has been added in text: “Fig 5(a) shows the distinct actin deformation by the increased fluorescence signals at the edges of the cells in the compressed groups under the micro-piston at zero time after compression, compared to the control (non-compressed) groups around the micro-piston. Thus, membrane bulges formed during cell compression at 15.6 kPa as observed in S1 Movie and fixed directly after compression appeared as increased actin signal at cell edges. Conversely, at 24 h-recovery after compression, cell actin deformation seems recovered in the group under the micro-piston as appeared similar to the control group.”

A note about the unevenness of brightness in phase contrast images has been added to the caption of Fig 5(a): “The unevenness of brightness in phase contrast images is due to that imaging focal plane was on cells cultured on glass while the micro-piston, brought back to static state after compression, was suspended on average 108 �m above the glass.”

18. Fig. 5 & 6 quantification of nucleus metrics: how many independent experiments were conducted?

Figure captions are revised.

“Fig 5. …… (mean ± s.e.m. n = 588, 230, 358, 174 cell nuclei from at least two or three independent experiments per each group).”

“Fig 6. …… (mean ± s.e.m. n = 406, 162, 342, 143 cell nuclei from at least two or three independent experiments per each group).”

19. Page 12, line 380: authors should explain why no significant changes in nucleus area were observed, as it would be expected given the high stress cells were experiencing. The authors should also explain the visible difference in two “0 (CR)” groups in Figure 6 (b)

All the statistical significance values for nuclear deformations in compressed and recovered cells after compression are noted in S2_File, newly added as supplementary file. 

S2 File caption “Statistical significance levels among all groups, obtained with Student's t-test run for nuclear deformations in compressed and recovered cells after compression presented in Fig 5 and Fig 6.” has been added. Fig 5 and Fig 6 captions have been also updated.

We added the following result and discussion for the difference between the two "0 (CR)" values: “There is statistically significant change in nuclei area between the control groups of zero-time and of 24 h-recovery after compression (see S2 File). Although these are not exactly the same control groups in the same conditions and we intend to compare the test group (e.g. zero-time or 24 h-recovery after compression) with their own control groups, we consider that the observed significant difference could be due to cell-cell signaling between the cells in compressed and control groups during that 24 h-recovery period after compression. In the groups of zero-time after compression, we did not allow control and compressed cells enough time to further interact with each other as they were fixed right after the compression.”

We further added the following discussion for the absence of statistical change in nucleus area between the control and compressed groups at zero-time after compression, in the associated paragraph: “While we expect a statistically significant change between the compressed and its control group at zero-time after compression at 20.8 kPa similar to the results observed in our previous report [5], we attribute the absence of such a significant change to cells being cultured on Matrigel-coating here as opposed to PLL-coating used in the previous experiments. Since Matrigel is expected to better represent the cancer cell microenvironment, it thus aided in cell growth into denser monolayers and formation of supporting extracellular matrix layer, that can alter sensing of the applied stress for the compressed cells.”

Reviewer #3: Onal and colleagues present a previously published method to dynamically compress a monolayer of cells. In this article, they use this device to briefly study how compression can impact actin expression, through a live reporter.

The study is technically sound. However, there are some points that need to be addressed before I can reach a decision:

1/ The claim of novelty of the device needs to be taken away. The device does not seem different from the one published by the authors in 2021 in Frontiers in Physics, and is inspired by the one published by Mishra et al (PNAS 2017), which should be acknowledged. In their 2021 paper, the authors notably already perform cyclic compression. Can the authors comment on what is really new, and if nothing is, can they transform the abstract and parts of introduction / conclusion to have it more focused on the study on actin?

Multiple parts in the manuscript have been revised and novelty section has been improved as in the last paragraphs of the Introduction, as highlighted in the revised manuscript. Abstract and Conclusion have been also revised.

We thank the reviewer for letting us know about Mishra et al (PNAS 2017) which we were not aware of. We have gone through the details of the microfluidic device presented in the supplementary of the paper. The device includes 2 layers of PDMS with micro patch pad array of 15-30 �m height which can be obtained in a single master. Design and applications are more adapted to yeast-size cells. Thus, the two platforms developed by Onal et al 2021 Frontiers in Physics and Mishra et al 2017 PNAS are different in dimensions and fabrication methods. Our microfluidic platform is composed of 3 layers of PDMS with microchannels and micro-pistons of 200 – 300 �m height and we provide dynamic cell compression at mammalian cell-culture suited dimensions. The 3 layers must be fabricated separately as UV lithography of a single master with varying structures at 200-300 �m (i.e. a micro-channel of 300 mm with a suspended micro-piston of 200 �m height and also attached to a membrane of ~200 �m thick at the same time) for a fluidic layer is not feasible to pattern as in the method of Mishra et al. to obtain small structures in a patterned single fluidic layer. One of the advantages of our platform is that dimensions of any layer can be changed easily as the layer dimensions are not depending on each other by a single master. The spin coating method to obtain such large structures as monolithically attached to PDMS membrane in our platform has been also very novel to obtain structures of 200 �m height and 300 �m diameter/length micro-pistons as attached to varying membrane thicknesses (varied from 100 �m up to 345 �m) at high resolution and high fidelity on a 4-inch Si/photoresist master. Cell loading methods are also quite different between the two papers. We use piston-retracted loading, which is a dynamic method and pistons are retracted up towards control microchannel while loading cells into the bottom channel to ensure homogenous distribution of cells under and around micro-piston. About the compression values applied, Mishra et al. used 7 psi (~48 kPa) widely in their experiments and they mention that the pressure level they use is physiologically relevant, as yeast cells can experience up to about 100 psi (~690 kPa) when grown in confined geometries. These values, especially up to 690 kPa, used in yeast applications are quite high pressures compared to physiology of cancer cell compression microenvironments. Additionally, it is not clear whether 7 psi used Mishra et al. in their device is externally applied pressure or contact pressure. Since no contact pressures are mentioned in their paper (including for supplementary as well), we consider that the mentioned value is externally applied pressure and there would be need to predict what were the internal/contact pressures via more device characterization methods. In our studies we typically characterize both externally applied pressures by a pressure pump and internal/contact pressures experienced inside the microdevice as PDMS is a hyperelastic material. 

2/ I was surprised that the authors present their method as a great strategy to apply cyclic compressive stresses, but did not really present results on this part (playing with frequency for instance, etc). It is discussed on p.15, but no experiments are presented. Have the authors played with this parameter? If not, maybe it would be better to put less emphasis on this in the manuscript. Also, can the authors provide a rationale for the frequency of the oscillations used in this study?

S1 Movie and S2 Movie, and Fig 1(d) representing how the sequential cyclic compression application was done for S2 Movie and Fig 3, show temporal evolution of the dynamic cell compression and a clear control of the process. 

Further explanation of the profiles in Fig 1(d) has been added at the end of 1st paragraph of the section “Live cell actin profiles of cancer cells during sequential cyclic compression”: “Throughout the application of sequential cyclic compression, cells were allowed to adjust to each next higher applied pressure. As shown in the pressure amount ladder in Fig 1(d), pressure was applied gradually in 30 s among rising steps of the ladder until the piston reached the focal plane of the cells and started cell deformation. After observing that cells were distinctly deformed at a mild compression level but not to a disruption or lysis level, compression was set to be in cyclic mode. Although the pressure was changing suddenly along the cycles as per the settings of the selected cyclic mode in the pressure controller, cells were not damaged when compressed as they resulted in being resistant at this physiological value. Next, pressure application was continued with gradual increase from 350 to 370 mbar, from 370 to 400 mbar and from 400 to 640 mbar in 30 s for each, as shown in Fig 1(d). Thus, cells could have the time to adjust during the pressure increase and show their intrinsic response to the pressurized stage of each compression level.”

The application rates and durations of the process in S1 Movie was already shown on the moving graph in the movie itself and was also clearly noted on its caption: “Temporal evolution of the dynamic cell compression within Matrigel-coated micro-piston devices. Time-lapse live cell imaging with 100 ms per frame was recorded for ladder pressure increase from 0 kPa up to 15.6 kPa and a short cycle at 15.6 kPa with gradual increase (from 0 kPa to 15.6 kPa) and decrease (from 15.6 kPa to 0 kPa) in 30 s for each. These steps were sequentially followed by 1 h-long cyclic compression between 0 kPa and 15.6 kPa (piston contact pressure from simulation). Cells were compressed in the temporal evolution of the dynamic pressure control within Matrigel-coated micro-piston devices and dynamic changes in cells during the cyclic compression were captured in detail at a rate of 100 ms per frame.”

During the mechanical characterizations of the microdevice itself, multiple profiles that the platform could apply was shown in Onal et al. 2021 Frontiers in Physics.

Thus, it was discussed and noted in this manuscript, for the frequency of the oscillations as well in the Results and discussion, section “Comparison of the experimental and computational results in micro-piston device”: “Although the platform is capable of temporal control of cell compression (e.g. to decrease, increase or remove the pressure at any certain time interval), duration of each stage in a cycle was set to 5 minutes, corresponding to a 5-min compressed stage followed by 5-min rest stage in each cycle. This was chosen to allow cells enough time during each stage to complete bulge formation or recovery, processes which were considered important for the investigation of cell deformation and recovery after compression. Sensor readings in S1 Movie showed that the micro-piston actuated according to externally applied pressures with reliable control and could be operated in short durations gradually or suddenly when needed.”

This was further discussed with a next paragraph added following the paragraph above in the same section: “Cyclic compression in the study of Ho et al. was applied only for 6 minutes in total [2], significantly shorter than the entire duration of a cyclic compression in this work, and results presented here suggest that cell deformation at compression stage and recovery at rest stage depends on the temporal length of the compression on cells. Particularly observable in S1 Movie, cell membrane bulge formation slowly developed during the compression stage while cells were responding to contact pressure at the interface with the PDMS piston. This was despite the fact that the piston was actuated onto the cells rapidly during cyclic compression as directed by the external pressure controller. As shown by the pressure sensor readings in the same movie, PDMS membrane and piston were consistently responding to amount, duration and mode of the applied pressure. Thus, the gradual deformation of cells and bulge formation during compression stages must originate in the response type of the cells to mechanical load when in solid contact with the PDMS piston. During rest stages, where the compression was lifted, cell bulges did not fully recover right after compression was removed, suggesting that the cells require a certain amount of time to recover compressed cellular parts (S1 Movie). Based on these observations of temporal evolution of the cell compression application, and comparison of application durations in this study with the study by Ho et al. [2], we hypothesize that this recovery also depends on the duration and number of the cycles.”

And supported with the paragraph revised and moved to Introduction, while introducing the literature frequencies and this work:

“To date, dynamic compression experiments in literature include frequencies of 0.1 – 30 Hz with an applied stress of 5.1, 9.3, 12.9 and 18.7 kPa on breast cancer cells over short durations (30 – 300 s), as used by Takao et al., who observed a mixed mode of apoptosis and necrosis-dominant cell death in mechanically compressed monolayers in a bulk compression platform [18]. In contrast, work by Novak et al. used a frequency of 0.05 Hz for cyclic compression applied at 3.9 – 6.5 kPa for 24 hours [3]. They observed an increase in proliferation capacity and decrease in apoptosis for OVCAR3 and OVSAHO ovarian cancer cells cultured in 3D hydrogel components in a bioreactor at millimeter scales. In another example, Ho et al. applied cyclic compression by alternating between external pressures of 10 and 15 psi (68.9 and 103.4 kPa) at 0.5 Hz for 6 minutes [2]. Using a single cell microfluidic compression platform, their application resulted in full recovery and thus no permanent plastic deformation in MCF10A normal breast epithelial cells after compression [2]. In our work we add to these examples by using the capabilities of our compression platform to apply cyclic compression with a low frequency in microfluidic settings. In particular, we utilize this functionality to provide a first investigation of the effects of cell compression, deformation and recovery after compression on SKOV-3 ovarian cancer cells. As part of this we provide evidence that the amount of the pressure and duration of the application clearly affects morphology during cell compression and recovery. While compression frequencies in an in vivo tumor microenvironment are not yet fully known, cyclic compression is important for maintenance of cell culture during mechanical compressions in an in vitro experimental setup, in particular as it enhances media flow underneath the micro-pistons.”

3/ I have some difficulties with the motivations in the introduction for the choice of ovarian cancer cells. In cancer, compressive stress is very slowly increasing, and not cyclically changing. Dynamic stress, though, can be found in the heart or during breathing, or also through the peristaltic motion during digestion. These examples, which would be great to motivate the dynamic capabilities of the device, are not presented, and instead, the focus is made on cancer. Moreover, I would not call 16 or 21 kPa as “mild” compressive stresses: these are rather important. The authors may wish to re-write their introduction if they want to introduce the device through its dynamic capabilities. But in this case, ovarian cancer cells may not be the best model (although, the authors can just say they use these cells as a model cell line for this test study).

The first two paragraphs have been revised to better reflect on the importance of studying cyclic compression, and dynamic peritoneal environment of ovarian cancer cells. 2nd paragraph give insights on the physiological values that have been in the literature and further values that should be investigated: 

“While some cells experience permanent plastic deformation after a repetitive mechanical tensile loading and unloading, the impact of such repetitive compression on deformation of cells remains largely unknown [1,2]. As such, the ability to apply cyclic compression is crucial for any experimental setup aimed at the study of mechanical compression taking place in cell and tissue microenvironments [3–5]. For instance, in ovarian cancer, cells are exposed to chronic compressive stress from different sources such as tumor growth, displacement within stromal tissue and hydrostatic pressure out of ascitic fluid in peritoneal cavity [3,6]. Ascites-induced compression exists in the peritoneal microenvironment due to the ascites volume that can reach >2L and increases ovarian cancer cell adhesion to peritoneum, shown by Asem et al. using in vivo assay [7]. This peritoneal microenvironment is continuously affected by the movements of the surrounding organs, musculoskeletal dynamics, and gravity [6]. Thus, ovarian cancer cells might experience cyclic compression profiles during chronic exposure to compression from different sources and due to anatomical location of the ovary. Mimicking such compression profiles and physiological pressure values in an in vitro dynamic compression system is necessary to further study impact of compressive stress in cancer.

Compressive stress is estimated to reach 18.9 kPa for human tumors and can exceed 20 kPa based on the experimental data from murine tumors, while those values can be even higher in situ with the impact of the surrounding extracellular matrix in tumor microenvironment [3,8]. Compared to other cancer types such as breast and metastatic bone niches, compression studies have been limited for ovarian cancer, including for investigating the impact of various amounts of applied pressures [6]. Pressure values existing in literature that are applied for ovarian cancer have been in a range from ∼3 kPa to 6.5 kPa [3,7,9]. Ovarian cancer cell response at higher applied pressures that can occur in the body need further investigation.”

When it comes to categorization as Mild compression this was introduced and used so in Onal et al, 2021, Frontiers in Physics while investigating cell viability response to a wide range of pressures in 4 categories. In those previous experiments, we showed SKOV-3 ovarian cancer cells were highly viable as on average 94% for Mild (15.6–15.9 kPa) and 77% for Intermediate 1 (23.8–26.8 kPa). Thus, we preferred to continue to use the same categorization, as it was also used in this manuscript while applying sequential cyclic compression to study dynamics of live cell actin, from Mild to Intermediate 1 to Intermediate 2 and Severe pressures. 

Similarly, and actually beyond the mild and intermediate pressures we used in our study for ovarian cancer cells, Hosmane et al. (2011, Lab on a Chip) showed application of compression on single axons in the range of 0-250 kPa and categorized as mild (<55 kPa), moderate (55 - 95 kPa) and severe levels (>95 kPa). 

Cyclic compression has been also applied to breast cancer cells, normal breast epithelial cells, and other ovarian cancer cell lines, as revised to Introduction:

“To date, dynamic compression experiments in literature include frequencies of 0.1 – 30 Hz with an applied stress of 5.1, 9.3, 12.9 and 18.7 kPa on breast cancer cells over short durations (30 – 300 s), as used by Takao et al., who observed a mixed mode of apoptosis and necrosis-dominant cell death in mechanically compressed monolayers in a bulk compression platform [18]. In contrast, work by Novak et al. used a frequency of 0.05 Hz for cyclic compression applied at 3.9 – 6.5 kPa for 24 hours [3]. They observed an increase in proliferation capacity and decrease in apoptosis for OVCAR3 and OVSAHO ovarian cancer cells cultured in 3D hydrogel components in a bioreactor at millimeter scales. In another example, Ho et al. applied cyclic compression by alternating between external pressures of 10 and 15 psi (68.9 and 103.4 kPa) at 0.5 Hz for 6 minutes [2]. Using a single cell microfluidic compression platform, their application resulted in full recovery and thus no permanent deformation in MCF10A normal breast epithelial cells after compression [2]. In our work we add to these examples by using the capabilities of our compression platform to apply cyclic compression with a low frequency in microfluidic settings. In particular, we utilize this functionality to provide a first investigation of the effects of cell compression, deformation and recovery after compression on SKOV-3 ovarian cancer cells. As part of this we provide evidence that the amount of the pressure and duration of the application clearly affects morphology during cell compression and recovery. While compression frequencies in an in vivo tumor microenvironment are not yet fully known, cyclic compression is important for maintenance of cell culture during mechanical compressions in an in vitro experimental setup, in particular as it enhances media flow underneath the micro-pistons.” 

Thus, different to bulk compression platform by Takao et al. (2019, Biology Open), bioreactor compression platform at millimeter scales by Novak et al. (2020, Cancers) and single cell microfluidic compression platform by Ho et al. (2018, Frontiers in Bioeng and Biotech), in our platform with microchannels of 200 – 300 �m height we provide dynamic cell compression at mammalian cell-culture suited dimensions that can be adapted to 2D (cell monolayers, individual cells) and 2.5 (dense cultures with hydrogel coatings, cluster of cells) and in the future work 3D (spheroids and cell-embedded hydrogels) cultures.

As the reviewer pointed out, this method can be applied to all other cell types, as it was noted in Conclusions:

“As demonstrated here with controlled micro-scale mechanical cell compression and recovery in a flexible microdevice, our comprehensive method will provide a more accurate replication of cell-physiological mechanisms to study both short- and long-term effects of compression in cellular microenvironments.”

4/ The actin intensity result seems correlated to survival, oddly. Can the authors plot these two parameters alongside as a function of pressure on same plot? Can the authors speculate on the decrease observed under pressure? Finally, have the authors checked for bleaching, by changing the sequence of compression for instance?

Cell viability results for the applied pressure ranges were reminded from Onal et al, 2021, Frontiers in Physics to give an idea at what viability states the cells were at these pressure ranges while investigating live cell actin cytoskeleton under dynamic compression: “……. the actin-GFP transduced cancer cells were dynamically stimulated and compressed with various intermediate actuation pressures in a sequential cyclic manner, namely following a sequence from Mild (15.6–15.9 kPa) to Intermediate 1 (23.8–26.8 kPa), Intermediate 2 (37.8–41.7 kPa) and Severe (127.8–140 kPa) piston contact pressures. The first two ranges of this sequence fell within physiological and upper physiological pressure values and cell viability was high (on average 94% for Mild and 77% for Intermediate 1), while Intermediate 2 and Severe levels constitute hyperphysiological values and cell viability decreased (on average 43% for Intermediate 2 and 20% for Severe) [5].” 

Thus, our previous report showed mechanical stress induced cell damage and death towards higher pressures so categorized as Intermediate 2 and Severe pressures, providing the degree of the damage as per the applied pressures via viability assays conducted in detail in Onal et al 2021 Frontiers in Physics.

The calculated actin signals were compared between the region under piston and control region around the piston. Actin signal in the control regions were also compared to each other across the cycles and remained statistically insignificant (S1 File) as it was indicated in the manuscript in Results and discussion in section “Live cell actin profiles of cancer cells during sequential cyclic compression”: “The change in actin signal for compressed cells under micro-piston was statistically different compared to non-compressed cells in control regions around the micro-piston at all applied compression levels (p <0.05, see S1 File). Actin signal changes in the control regions on the other hand, remained statistically insignificant (p >0.05).”

The following discussion has been added in the same paragraph: “Thus, the GFP tagged actin signal in control regions was robust during the entire process, as expected [27]. This can be reference to that fluorescence changes in the region under micro-piston (i.e. decrease in actin signal) was due to the impact of mechanical compression.” 

Thus, given that there was no statistically significant difference between the control groups throughout the application, we do not have signs indicating for photobleaching. 

The following sentence was noted in the same paragraph”: 

“At the higher pressures of Intermediate 2 and Severe, actin of most of the cells was disrupted compared to in the initial stages of Mild compression (p <0.05, see S1 File).”

The next sentence in the same paragraph has been slightly revised to: “Interestingly, a few cells on the glass surface usually remained resistant to the increasing pressure ranges, as evidenced by their retained fluorescent signal while compressed under the micro-piston.”

The following discussion as relevant to the observation sentence above has been added: “During the entire process, wide-field imaging was used and the focus was kept on one focal plane on the glass surface where cells were initially adhered to. At compressed stages, glass and PDMS piston surfaces were in contact and at the same focal plane. At the rest stages when pressure is released and piston is retracted back, to be able to gain the signal from piston surface there would be need to bring the focal plane to the suspended piston level that is on average 108 �m above the glass surface. In the region under the micro-piston, there seem to be variations in gained actin signal between the compressed and rest stages of each cycle (Fig 4), but these were not statistically significant (p>0.05, see S1 File). Thus, no significant amount of cell artifacts was attached to piston surface.”

Overall, for analysis of actin-GFP signal we had done CTCF calculations and have now improved the graph to show the statical significances in Fig 4 in the revised manuscript. When there was statistically significant loss in actin-GFP signal and as seen on S2 Movie, actin disruption happens and that indicates for plastic response of the cells at those higher pressures. We added a discussion in the last paragraph of section “Live cell actin profiles of cancer cells during sequential cyclic compression”: “The observed disruption of actin cytoskeleton in these experiments indicates for cell plasticity during a cyclic compression in ovarian cancer when varying higher pressures occur in the microenvironment. There is need for more studies in literature to compare our results on cell plasticity after cyclic or dynamic compression applied sequentially up to high pressures. For cell actin deformation at milder pressures, however, a distinct change in actin cytoskeleton morphology was observed by Asem et al. for mesothelial cells, interacting with ovarian cancer cells, under compression that is statically applied at ∼3 kPa for 24 h in a bulk compression system. As a result, the interaction between ovarian cancer cells and mesothelial cells by enhanced formation of actin-based tunneling nanotubes (TNTs), and hence metastatic progression in ovarian cancer, was promoted under compression [7].” 

In the section Conclusions it has been also added that “While actin deformation at mild pressures (e.g. 15.6 kPa) in cyclic mode indicates for cell elasticity, the resultant actin disruption indicates for cell plasticity that can emerge when ovarian cancer cells are exposed to varying and higher pressures in a microenvironment.”

Minor points:

1/The paper is too affirmative at parts (for instance: p.9, “the results illustrate how the actin cytoskeleton […] changes in response to compressive stress” is overstated as the authors show a correlation, and do not provide a clear mechanism). The authors need to be more careful on their claims.

We have gone through the entire manuscript and have done revisions and additions in multiple parts, as highlighted in the revised manuscript. 

We have revised Fig 4 and better referenced the associated Fig 1(d) and S1 File in text. We have further explained the associated graphs and significance values and interpreted and discussed the results. Thus, we have extensively revised the associated section “Live cell actin profiles of cancer cells during sequential cyclic compression” as highlighted in the revised manuscript.

2/ Could the authors put all pressure values in the same unit? We sometimes read kPa, mbar or psi (which is by its very name not a pressure unit and not in the international system of units), and a harmonization would help the reader.

We used mbar to indicate the externally applied pressures as per the settings and calibrations on the pressure pump, pressure sensor and sensor readers used while running this work. This would help towards repeatability of the work and the reader to choose the appropriate pump, sensor, reader and their settings, should they like to use similar methods. We used kPa to indicate for the piston contact pressures as conversion from pascal by N/m2 computed on the mechanical modelling. Our piston contact pressures given in kPa were compared to the literature values on contact pressures which are also given in kPa in the associated references.

mbar and kPa are the two units that seem to be commonly used in the publications in cell compression field. 

A note was added in the methods section “Device operation for sequential cyclic compression” to guide the reader on how it is a better way to read the units:

“Pressure values in this work are given in mbar for externally applied pressures, as per calibrated settings of the pressure pump, pressure sensor and sensor reader, while kPa is used for the piston contact pressures, which are based on conversion from values computed using mechanical modelling.” The following sentence has been also added to the end of this section for a better guidance: “Our piston contact pressures given in kPa were compared to the literature values on contact pressures which are also given in kPa in the associated references.”

As for psi, typically, we do not use psi in our work. psi values mentioned in our manuscript belongs to Ho et al. (2021, Frontiers in Bioeng and Biotech) as externally applied pressures and we also added their conversion to kPa in parentheses. We generally intend to not change the units described in our references from the literature. 

3/ Part at the end of page 8 read more like introduction and the end of page 14 / beginning of page 15 read more like a discussion. Maybe they would be better placed in these specific sections to avoid confusion.

The following part has been moved to ‘Introduction’ from ‘Results and discussion’:

“To date, dynamic compression experiments in literature include frequencies of 0.1 – 30 Hz with an applied stress of 5.1, 9.3, 12.9 and 18.7 kPa on breast cancer cells over short durations (30 – 300 s), as used by Takao et al., who observed a mixed mode of apoptosis and necrosis-dominant cell death in mechanically compressed monolayers in a bulk compression platform [15]. In contrast, work by Novak et al. used a frequency of 0.05 Hz for cyclic compression applied at 3.9 – 6.5 kPa for 24 hours [3]. They observed an increase in proliferation capacity and decrease in apoptosis for OVCAR3 and OVSAHO ovarian cancer cells cultured in 3D hydrogel components in a bioreactor at millimeter scales. In another example, Ho et al. applied cyclic compression by alternating between external pressures of 10 and 15 psi (68.9 and 103.4 kPa) at 0.5 Hz for 6 minutes [2]. Using a single cell microfluidic compression platform, their application resulted in full recovery and thus no permanent plastic deformation in MCF10A normal breast epithelial cells after compression [2]. In the following we add to these examples by using the capabilities of our micro-piston platform to apply cyclic compression with a low frequency in microfluidic settings. In particular, we utilize this functionality to provide a first investigation of the effects of cell compression, deformation and recovery after compression on SKOV-3 ovarian cancer cells. As part of this we provide evidence that the amount of the pressure, duration of the application and number of the cycles clearly affects morphology and compression in this cell type. While compression frequencies in an in vivo tumor microenvironment are not yet fully known, cyclic compression is important for maintenance of cell culture during mechanical compressions in an in vitro experimental setup, in particular as it enhances media flow underneath the micro-pistons.”

Some sentences are revised to

“…….. Using a single cell microfluidic compression platform, their application resulted in full recovery and thus no permanent deformation in MCF10A normal breast epithelial cells after compression [2]. In our work we add to these examples by using the capabilities of our compression platform to apply cyclic compression with a low frequency in microfluidic settings. ……. As part of this we provide evidence that the amount of the pressure and duration of the application clearly affects morphology during cell compression and recovery.”

The following part has been moved to discussion part of the section ‘Comparison of the experimental and computational results in micro-piston device’ from the end of ‘Results and discussion’ (the end of page 14 / beginning of page 15):

“Cyclic compression in the study of Ho et al. was applied only for 6 minutes in total, significantly shorter than the entire duration of a cyclic compression in this work, and results presented here suggest that cell recovery after compression depends on the temporal length of the compression on cells. The overall extent of recovery of compressed cells represented by cell membrane bulges, the actin cytoskeleton and measurements of shape descriptors of cell nuclei can yield further insight into the plasticity of the cancer cells. Particularly observable in S1 Movie, cell plasticity depends on the amount of pressure applied, as well as the timing. Cell bulge formation slowly developed during the compression stage while cells were responding to contact pressure at the interface with the PDMS piston. This was despite the fact that the piston was actuated onto the cells rapidly as directed by the external pressure controller. As shown by the pressure sensor readings in the same movie, PDMS membrane and piston were consistently responding to amount, duration and mode of the applied pressure. Thus, the gradual deformation of cells and bulge formation during compression stages must originate in the response type of the cells to mechanical load when in solid contact with the PDMS piston. During rest stages, where the compression was lifted, cell bulges did not fully recover right after compression was removed, suggesting that the cells require a certain amount of time to recover compressed cellular parts (S1 Movie). Based on these observations of temporal evolution of the cell compression application, and comparison of application durations in this study with the study by Ho et al. [2], we hypothesize that this recovery also depends on the duration and number of the cycles.”

Revised to

“Cyclic compression in the study of Ho et al. was applied only for 6 minutes in total [2], significantly shorter than the entire duration of a cyclic compression in this work, and results presented here suggest that cell deformation at compression stage and recovery at rest stage depends on the temporal length of the compression on cells. Particularly observable in S1 Movie, cell membrane bulge formation slowly developed during the compression stage while cells were responding to contact pressure at the interface with the PDMS piston. This was despite the fact that the piston was actuated onto the cells rapidly during cyclic compression as directed by the external pressure controller. As shown by the pressure sensor readings in the same movie, PDMS membrane and piston were consistently responding to amount, duration and mode of the applied pressure. Thus, the gradual deformation of cells and bulge formation during compression stages must originate in the response type of the cells to mechanical load when in solid contact with the PDMS piston. During rest stages, where the compression was lifted, cell bulges did not fully recover right after compression was removed, suggesting that the cells require a certain amount of time to recover compressed cellular parts (S1 Movie). Based on these observations of temporal evolution of the cell compression application, and comparison of application durations in this study with the study by Ho et al. [2], we hypothesize that this recovery also depends on the duration and number of the cycles.”

The following sentence that was at the end of the paragraph above has been slightly revised and moved to last paragraph of “Results and discussion”:

“Based on this we conclude that the presented sequential cyclic compression and cell recovery after compression method should be employed in the future to study the effects of compression and decompression in intratumor vessels or endothelial cells to help understand the structure and functionality of decompressed vessels, importance of which was shown in a study by Padera et al. [25].”

Revised to 

“For instance, we propose that the presented sequential cyclic compression and cell recovery after compression method should be employed in the future to study the effects of compression and decompression in intratumor vessels or endothelial cells to help understand the structure and functionality of decompressed vessels, importance of which was shown in a study by Padera et al. [34].”

---

## [Decision Letter · Decision Letter 1]

19 Sep 2022

PONE-D-22-08675R1Application of sequential cyclic compression on cancer cells in a flexible microdevicePLOS ONE

Dear Dr. Onal,

Thank you for submitting your manuscript to PLOS ONE. After careful consideration, we feel that it has merit but does not fully meet PLOS ONE’s publication criteria as it currently stands. Therefore, we invite you to submit a revised version of the manuscript that addresses the points raised during the review process.

Please, address all the comments made by reviewer 1. Notice that more clarifications are required as it is not clear if the results are an artifact of the cyclic forces.

We look forward to receiving your revised manuscript.

Kind regards,

Antonio Riveiro Rodríguez, PhD

Academic Editor

PLOS ONE

Reviewers' comments:

Reviewer's Responses to Questions

**Comments to the Author**

1. If the authors have adequately addressed your comments raised in a previous round of review and you feel that this manuscript is now acceptable for publication, you may indicate that here to bypass the “Comments to the Author” section, enter your conflict of interest statement in the “Confidential to Editor” section, and submit your "Accept" recommendation.

Reviewer #1: (No Response)

Reviewer #2: (No Response)

2. Is the manuscript technically sound, and do the data support the conclusions?

Reviewer #1: Partly

Reviewer #2: Partly

3. Has the statistical analysis been performed appropriately and rigorously? 

Reviewer #1: Yes

Reviewer #2: I Don't Know

4. Have the authors made all data underlying the findings in their manuscript fully available?

Reviewer #1: Yes

Reviewer #2: Yes

5. Is the manuscript presented in an intelligible fashion and written in standard English?

Reviewer #1: Yes

Reviewer #2: Yes

6. Review Comments to the Author

Reviewer #1: There are still some issues with the manuscript, and I have a few follow-up comments:

Original comment 1) In the introduction, it is mentioned that "...cells are exposed to chronic compressive stress from different sources such as tumour growth, displacement within stromal tissue and hydrostatic pressure out of ascitic fluid in peritoneal cavity." This is not completely accurate, as hydrostatic pressure will not deform the cell in the same way as anisotropic compression. The forces would be isotropic for fluid pressure.

Authors' response: Asem et al. (2020) Scientific Reports, show that ascites-induced compression exist in the peritoneal microenvironment due to the ascites volume that can reach >2L and cause intraperitoneal pressure (IPP) to rise from normal values of 5 mmHg to as high as 22 mmHg (~3 kPa). Ovarian cancer cell adhesion to peritoneum increased due to the impact of ascites-induced compression, shown in in vivo assay.

Asem et al. also applied a pressure of ∼3 kPa on OVCAR5 or OVCAR8 cells added atop the mesothelial surface of murine peritoneal explants ex vivo for 24 hours in a bulk compression system. As a result, they observed enhanced interaction between peritoneal mesothelial cells and cancer cells via induction of tunneling nanotubes (TNT), which later lead to metastatic ovarian cancer progression. These compression-induced TNTs are actin based and aid in trafficking of mitochondria between ovarian cancer cells and mesothelial cells.

The other consequence of the ascites-induced compression through hydrostatic pressure in the peritoneal cavity has been a more linear anisotropic alignment of peritoneal collagen fibers. Such alignment of collagen is a common indication for enhanced invasion of solid tumors.

Overall, Asem et al. have found that ascites-induced compression promotes metastatic ovarian cancer progression.

This information has been summarized and added in the first paragraph of the Introduction: “…….Ascites-induced compression exists in the peritoneal microenvironment due to the ascites volume that can reach >2L and increases ovarian cancer cell adhesion to peritoneum, shown by Asem et al. using in vivo assay [7]. This peritoneal microenvironment is continuously affected by the movements of the surrounding organs, musculoskeletal dynamics, and gravity [6].…… ”

Follow-up comment: It is not physically possible for isotropic fluid pressure to "compress" a cell against the peritoneal wall. So the in vitro experiments in the cited paper (which used solid anisotropic compression, not fluid pressure) did not mimic the in vivo situation. I agree that the observed increase in cell adhesion in vivo was likely due to strain-induced changes in the collagen network in the peritoneal wall. But this is a result of the fluid pressure stretching the containing tissue and not a direct effect on the cancer cells.

Original comment 3) Does the disruption of the cytoskeleton depend on the rate of application of the force? If the piston is applied more slowly, do the cells have time to adjust and avoid damage?

Authors' response: Regarding deformation and disruption of actin cytoskeleton, Figure 3 and Figure 4 show how live cell actin cytoskeleton behave under dynamic compression applied over time in cyclic mode at a physiological value and sequentially in varying applied pressures in ascending order, and are out of at least 3 independent experiments. In text, while explaining the associated figures, the representative pressure profile for sequential cyclic compression process was indicated to be in Figure 1(d): “Similar as to the sequence shown in Fig 1(d), first, a pressure amount ladder was formed up to a pressure in the mild compression range, represented by an externally applied pressure of 350 mbar and resulting in a piston contact pressure of 15.6 kPa, during which the piston reached the cells at the bottom surface and cells started to distinctly deform.”

We think that this pressure application is gradual enough for cells to adjust to the varying pressures during sequential cyclic compression application. Mechanical cell damage that we see at higher pressures is intrinsic to cell response to higher amounts of applied pressure. The following explanation has been added at the end of the 1st paragraph in section “Live cell actin profiles of cancer cells during sequential cyclic compression”:

“Throughout the application of sequential cyclic compression, cells were allowed to adjust to each next higher applied pressure. As shown in the pressure amount ladder in Fig 1(d), pressure was applied gradually in 30 s among rising steps of the ladder until the piston reached the focal plane of the cells and started cell deformation. After observing that cells were distinctly deformed at a mild compression level but not to a disruption or lysis level, compression was set to be in cyclic mode. Although the pressure was changing suddenly along the cycles as per the settings of the selected cyclic mode in the pressure controller, cells were not damaged when compressed as they resulted in being resistant at this physiological value. Next, pressure application was continued with gradual increase from 350 to 370 mbar, from 370 to 400 mbar and from 400 to 640 mbar in 30 s for each, as shown in Fig 1(d). Thus, cells could have the time to adjust during the pressure increase and show their intrinsic response to the pressurized stage of each compression level.”

Follow-up comment: The question was whether 30s is "gradual" enough. Tumor growth would be expected to cause increases in solid stress much more slowly than this, and cells may have sufficient time to reorganize mechanical structures to avoid damage.

Original comment 11) Further justification should be provided for the cyclic stress schedule chosen. Musculo-skeletal tissues would be expected to experience such forces, but this is not clear for ovarian cancer.

Authors' response: The cyclic stress schedule (i.e. 5-min compressed stage followed by 5-min rest stage in each cycle) was chosen to allow cells enough time during each stage to complete bulge formation or recovery, processes which were considered important for the investigation of cell deformation and recovery after compression. Membrane bulge formation and recovery processes are clearly observable in S1 Movie. This was explained in the following paragraph in section “Comparison of the experimental and computational results in micro-piston device”

“While inferring these results, the cell compression process was imaged in detail in Matrigel-coated microchannels at a speed of 1 frame per 100 ms, as shown in S1 Movie. As illustrated by this time-lapse video, the platform allowed for the temporal evolution of the dynamic cell compression to be investigated in each cycle while the piston was contacting cells and applying compression. Distinct cell membrane bulges could be observed to form during such compression at piston contact pressure of 15.6 kPa and recovery of cell membranes from these bulges was visible when pistons were lifted off during rest stages. Although the platform is capable of temporal control of cell compression (e.g. to decrease, increase or remove the pressure at any certain time interval), duration of each stage in a cycle was set to 5 minutes, corresponding to a 5-min compressed stage followed by 5-min rest stage in each cycle. This was chosen to allow cells enough time during each stage to complete bulge formation or recovery, processes which were considered important for the investigation of cell deformation and recovery after compression. Sensor readings in S1 Movie showed that the micro-piston actuated according to externally applied pressures with reliable control and could be operated in short durations gradually or suddenly when needed.”

Further explanation and discussion have been also added in this section as in the following paragraph:

“Cyclic compression in the study of Ho et al. was applied only for 6 minutes in total [2], significantly shorter than the entire duration of a cyclic compression in this work, and results presented here suggest that cell deformation at compression stage and recovery at rest stage depends on the temporal length of the compression on cells. Particularly observable in S1 Movie, cell membrane bulge formation slowly developed during the compression stage while cells were responding to contact pressure at the interface with the PDMS piston. This was despite the fact that the piston was actuated onto the cells rapidly during cyclic compression as directed by the external pressure controller. As shown by the pressure sensor readings in the same movie, PDMS membrane and piston were consistently responding to amount, duration and mode of the applied pressure. Thus, the gradual deformation of cells and bulge formation during compression stages must originate in the response type of the cells to mechanical load when in solid contact with the PDMS piston. During rest stages, where the compression was lifted, cell bulges did not fully recover right after compression was removed, suggesting that the cells require a certain amount of time to recover compressed cellular parts (S1 Movie). Based on these observations of temporal evolution of the cell compression application, and comparison of application durations in this study with the study by Ho et al. [2], we hypothesize that this recovery also depends on the duration and number of the cycles.”

Regarding sources of compression ovarian cancer cells are exposed to, further information has been added in Introduction:

“While some cells experience permanent plastic deformation after a repetitive mechanical tensile loading and unloading, the impact of such repetitive compression on deformation of cells remains largely unknown [1,2]. As such, the ability to apply cyclic compression is crucial for any experimental setup aimed at the study of mechanical compression taking place in cell and tissue microenvironments [3–5]. For instance, in ovarian cancer, cells are exposed to chronic compressive stress from different sources such as tumor growth, displacement within stromal tissue and hydrostatic pressure out of ascitic fluid in peritoneal cavity [3,6]. Ascites-induced compression exists in the peritoneal microenvironment due to the ascites volume that can reach >2L and increases ovarian cancer cell adhesion to peritoneum, shown by Asem et al. using in vivo assay [7]. This peritoneal microenvironment is continuously affected by the movements of the surrounding organs, musculoskeletal dynamics, and gravity [6]. Thus, ovarian cancer cells might experience cyclic compression profiles during chronic exposure to compression from different sources and due to anatomical location of the ovary. Mimicking such compression profiles and physiological pressure values in an in vitro dynamic compression system is necessary to further study impact of compressive stress in cancer.

Compressive stress is estimated to reach 18.9 kPa for human tumors and can exceed 20 kPa based on the experimental data from murine tumors, while those values can be even higher in situ with the impact of the surrounding extracellular matrix in tumor microenvironment [3,8]. Compared to other cancer types such as breast and metastatic bone niches, compression studies have been limited for ovarian cancer, including for investigating the impact of various amounts of applied pressures [6]. Pressure values existing in literature that are applied for ovarian cancer have been in a range from ∼3 kPa to 6.5 kPa [3,7,9]. Ovarian cancer cell response at higher applied pressures that can occur in the body need further investigation.”

Furthermore, it was noted in our manuscript that “While compression frequencies in an in vivo tumor microenvironment are not yet fully known, cyclic compression is important for maintenance of cell culture during mechanical compressions in an in vitro experimental setup, in particular as it enhances media flow underneath the micro-pistons.”

Follow up comment: If the study is meant to investigate ovarian cancer, it should be designed to mimic the in vivo mechanical environment. The authors should either provide evidence that ovarian cancer cells are exposed to cyclic stresses in vivo, or use their device to produce different (longer term) stress application. As presented, it is not clear whether the results are an artefact of the cyclic forces.

Reviewer #2: The authors revised the introduction and others places in the manuscript to include additional background comparing similar studies and their compression parameters with this study, and further clarifying the novelty of this study based on a previously developed device. Sufficient information is now provided to clarify Figure 2 and the corresponding results. However, some of the results require further clarification.

- Page 8, line 285: It is not clear what is the significance and relevance of cell membrane bulge formation under compression. How it is related to actin redistribution and existing literature should be discussed.

- Page 10, line 324- 359: this section is mainly describing parameters and process of the cyclic compression, therefore would better fit in Methods section.

- Page 10, line 333-334, “similar as to the sequence shown in fig 1(d)…”: Why not use the exact sequence for generating results in fig 3 in fig 1(d) instead of a similar sequence?

- Figure 3: What is the total imaging time throughout compression? Is photobleaching of GFP a concern?

- Page 11, line 411-412: The authors should be careful and more specific when describing observed changes in actin during cyclic compression. Deformation/damage implies structural changes of actin. It is not clear whether the authors observed structural changes in actin or simply redistribution of actin inside a cell. If actin is deformed, how a lower intensity of GFP corresponds to actin deformation should be explained.

- Page 11, line 376-378: Given that cell has a certain height and upon compression, part of the cell that was not in the focal plane can be compressed to the focal plane and add to the actin signal. The authors should clarify whether this was observed (maybe during one cycle of quick compression) and was an issue for the quantification. Similarly, are compressed cells and control cells in the same plane when compressed?

- Page 11, line 379-382: it is not clear why the focal plane needs to be at the piston level (above the glass) at rest stages, as cells were cultured on the glass slide.

- Page 11, line 386-395: Given that Ho et al used a much higher pressure compared to this study but observed actin formation instead of disruption, how much of the continuous decrease of actin signal in this study is due to insufficient rest time? Have the authors tested how long it would take cells to recover actin levels post one cycle of compression?

- Page 13, line 423-426: It would be helpful to have a zoomed in image for control cells as well, so the difference of actin distribution could be observed more directly. Additionally, in Fig 5(a) compressed cells seem to have stronger actin compared to control cells? Did actin level restore in compressed cells after 24 hours or just nucleus morphology?

- Page 15, line 475-477: based on figure 6 it seems like the actin profile was altered after 24h recovery compared to compressed cells? It would be helpful to point out cells that did not recover in the image.

- Page 17, line 496-498: specifying the difference, i.e., decrease/increase, would help improve the readability.

7. PLOS authors have the option to publish the peer review history of their article (what does this mean?). If published, this will include your full peer review and any attached files.

Reviewer #1: No

Reviewer #2: No

---

## [Author Response · Author response to Decision Letter 1]

5 Nov 2022

Reviewer #1: There are still some issues with the manuscript, and I have a few follow-up comments:

Original comment 1) In the introduction, it is mentioned that "...cells are exposed to chronic compressive stress from different sources such as tumour growth, displacement within stromal tissue and hydrostatic pressure out of ascitic fluid in peritoneal cavity." This is not completely accurate, as hydrostatic pressure will not deform the cell in the same way as anisotropic compression. The forces would be isotropic for fluid pressure.

Authors' response: Asem et al. (2020) Scientific Reports, show that ascites-induced compression exist in the peritoneal microenvironment due to the ascites volume that can reach >2L and cause intraperitoneal pressure (IPP) to rise from normal values of 5 mmHg to as high as 22 mmHg (~3 kPa). Ovarian cancer cell adhesion to peritoneum increased due to the impact of ascites-induced compression, shown in in vivo assay.

Asem et al. also applied a pressure of ∼3 kPa on OVCAR5 or OVCAR8 cells added atop the mesothelial surface of murine peritoneal explants ex vivo for 24 hours in a bulk compression system. As a result, they observed enhanced interaction between peritoneal mesothelial cells and cancer cells via induction of tunneling nanotubes (TNT), which later lead to metastatic ovarian cancer progression. These compression-induced TNTs are actin based and aid in trafficking of mitochondria between ovarian cancer cells and mesothelial cells.

The other consequence of the ascites-induced compression through hydrostatic pressure in the peritoneal cavity has been a more linear anisotropic alignment of peritoneal collagen fibers. Such alignment of collagen is a common indication for enhanced invasion of solid tumors.

Overall, Asem et al. have found that ascites-induced compression promotes metastatic ovarian cancer progression.

This information has been summarized and added in the first paragraph of the Introduction: “…….Ascites-induced compression exists in the peritoneal microenvironment due to the ascites volume that can reach >2L and increases ovarian cancer cell adhesion to peritoneum, shown by Asem et al. using in vivo assay [7]. This peritoneal microenvironment is continuously affected by the movements of the surrounding organs, musculoskeletal dynamics, and gravity [6].…… ”

Follow-up comment: It is not physically possible for isotropic fluid pressure to "compress" a cell against the peritoneal wall. So the in vitro experiments in the cited paper (which used solid anisotropic compression, not fluid pressure) did not mimic the in vivo situation. I agree that the observed increase in cell adhesion in vivo was likely due to strain-induced changes in the collagen network in the peritoneal wall. But this is a result of the fluid pressure stretching the containing tissue and not a direct effect on the cancer cells.

Indeed, Asem et al. has done in vivo compression assays in addition to in vitro and ex vivo. We would like to share the information from Asem et al. 2020 Scientific Reports in the way it is summarized by themselves:

“The majority of women with recurrent ovarian cancer (OvCa) develop malignant ascites with volumes that can reach > 2 L. The resulting elevation in intraperitoneal pressure (IPP), from normal values of 5 mmHg to as high as 22 mmHg, causes striking changes in the loading environment in the peritoneal cavity. The effect of ascites-induced changes in IPP on OvCa progression is largely unknown. Herein we model the functional consequences of ascites-induced compression on ovarian tumor cells and components of the peritoneal microenvironment using a panel of in vitro, ex vivo and in vivo assays. Results show that OvCa cell adhesion to the peritoneum was increased under compression. Moreover, compressive loads stimulated remodeling of peritoneal mesothelial cell surface ultrastructure via induction of tunneling nanotubes (TNT). TNT mediated interaction between peritoneal mesothelial cells and OvCa cells was enhanced under compression and was accompanied by transport of mitochondria from mesothelial cells to OvCa cells. Additionally, peritoneal collagen fibers adopted a more linear anisotropic alignment under compression, a collagen signature commonly correlated with enhanced invasion in solid tumors. Collectively, these findings elucidate a new role for ascites-induced compression in promoting metastatic OvCa progression.”

We also would like to share examples where compression can be induced on tumor cells via isotropic pressure, these have been discussed in an upcoming (accepted) review article by Onal et al. 2022 iScience:

“Apart from compression produced by a mechanical loading unit, such as a piston, and confinement through physical constraints, indirect restriction of the volume of multicellular spheroids (MCSs) can also be used to apply compression on cells. To achieve this, Delarue et al. cultured MCSs of various cell lines including colon carcinoma, breast cancer, and sarcoma cells and supplemented the culture medium of MCSs with dextran which is a biopolymer that does not penetrate single cells [Delarue et al. 2014 Biophysical J.]. Dextran addition into media exerted a moderate osmotic stress directly on the outermost layer of cells in spheroids. This osmotic stress was transmitted to the inner cells of spheroids as mechanical compressive stress, which in turn reduced the overall volume of the MCSs. Compressive stress applied through such volume limitation inhibited cell proliferation in tumor MCSs. A similar setup was constructed by interposing a dialysis membrane between the MCSs and media with dextran, upon which the MCSs of CT26 mouse colon carcinoma cells were mechanically compressed by stress transmitted from the dialysis membrane to inner cells via osmotic stress at the outermost layer of the cells [Montel et al. 2011 Physical Rev Letters and 2012 New J of Physics]. In either method, with or without dialysis membrane, the volume of the MCS was reduced by the applied stress. Although this osmotic effect by dextran did not originate from high osmolarity, such as produced by salts, compressive stress in these setups emerged as a network stress by an osmotic origin, which impacted the tumor growth rate and MCS volume. The resultant compressive stress was estimated to be 5 kPa or 10 kPa based on the concentration of dextran added to the culture medium. Thus, it can be said that the compressive stress transmission on the cells was not the product of a contacting physical surface, for example a flexible solid polymer (e.g. PDMS). Furthermore, this type of compression was also statically applied in bulk in 48-well plates, unlike the controlled and dynamic manner that can be achieved in microfluidic devices.”

Such a concentration change and transmission of compressive stress onto cells can happen via ascites in the peritoneal cavity based on the volume and composition of the ascitic fluid that is further discussed and referenced in the following in response to 3rd follow-up comment of the reviewer.

Original comment 3) Does the disruption of the cytoskeleton depend on the rate of application of the force? If the piston is applied more slowly, do the cells have time to adjust and avoid damage?

Authors' response: Regarding deformation and disruption of actin cytoskeleton, Figure 3 and Figure 4 show how live cell actin cytoskeleton behave under dynamic compression applied over time in cyclic mode at a physiological value and sequentially in varying applied pressures in ascending order, and are out of at least 3 independent experiments. In text, while explaining the associated figures, the representative pressure profile for sequential cyclic compression process was indicated to be in Figure 1(d): “Similar as to the sequence shown in Fig 1(d), first, a pressure amount ladder was formed up to a pressure in the mild compression range, represented by an externally applied pressure of 350 mbar and resulting in a piston contact pressure of 15.6 kPa, during which the piston reached the cells at the bottom surface and cells started to distinctly deform.”

We think that this pressure application is gradual enough for cells to adjust to the varying pressures during sequential cyclic compression application. Mechanical cell damage that we see at higher pressures is intrinsic to cell response to higher amounts of applied pressure. The following explanation has been added at the end of the 1st paragraph in section “Live cell actin profiles of cancer cells during sequential cyclic compression”:

“Throughout the application of sequential cyclic compression, cells were allowed to adjust to each next higher applied pressure. As shown in the pressure amount ladder in Fig 1(d), pressure was applied gradually in 30 s among rising steps of the ladder until the piston reached the focal plane of the cells and started cell deformation. After observing that cells were distinctly deformed at a mild compression level but not to a disruption or lysis level, compression was set to be in cyclic mode. Although the pressure was changing suddenly along the cycles as per the settings of the selected cyclic mode in the pressure controller, cells were not damaged when compressed as they resulted in being resistant at this physiological value. Next, pressure application was continued with gradual increase from 350 to 370 mbar, from 370 to 400 mbar and from 400 to 640 mbar in 30 s for each, as shown in Fig 1(d). Thus, cells could have the time to adjust during the pressure increase and show their intrinsic response to the pressurized stage of each compression level.”

Follow-up comment: The question was whether 30s is "gradual" enough. Tumor growth would be expected to cause increases in solid stress much more slowly than this, and cells may have sufficient time to reorganize mechanical structures to avoid damage.

Here first we would like to present some examples from the literature:

- Lee et al. 2018 LOC applied 1 hour (14 kPa air external pressure, static and 1 Hz dynamic, microfluidic) on alginate–chondrocyte constructs. They observed significant changes in chondrocyte deformability, no significant change in viability under dynamic compression and decrease in viability under static compression.

- Asem et al. 2020 Scientific Rep applied 1 hour (~3 kPa; ~22 mmHg, BioPress Flexcell, static) on murine peritoneal explants including mesothelial cells (that interacts with ovarian cancer cells). They observed significant changes in extracellular extensions including tunneling nanotubes (TNTs). It should be noted that the pressure here was static for 1 hour but the pressure amount ~3 kPa is relatively low compared to pressures applied in our study.

- Ho et al. 2018 Frontiers Bioeng Biotech applied 6-min cyclic compression alternating between external pressures of 10 and 15 psi (68.9 and 103.4 kPa) at 0.5 Hz on single MCF10A normal breast epithelial cells. They observed no significant change in cell height between before and after 6-min compression. 

- Takao et al. 2019 Biology Open applied cyclic compression stress for short durations of 30-60 s at 0.1 - 1.0 kPa (smaller loading) at 1 - 30 Hz, and of 210-300 s 5.1 - 18.7 kPa (larger loading) at 30 Hz. They observed mechanical stress-induced cell death (MSICD) in breast cancer cells, with mixed mode of 5–35% apoptosis and 5–95% necrosis dominant (smaller loading), and of 0.1–4.4% apoptosis and 5–98% necrosis dominant (larger loading), respectively.

- Hosmane et al. 2011 LOC applied varying compressive loads (0-250 kPa, contact pressures) to single axons for ~5 s. They observed growth (or regrowth) of axons in mild (<55 kPa), moderate (55-95 kPa) and severe (>95 kPa) levels of mechanical injury.

Comparing to some of the literature studies that are applied in in vitro compression platforms, we think that compression rate applied as one step pressure increase per 30 s (e.g. from 300 mbar to 320 mbar in 30s) that is applied during pressure ladder is gradual enough (in the Supplementary Movie 1, the gradual response of the cells can be observed distinctly in accordance with the pressure graph). This was followed by 5 min compressed – 5 min at rest cyclic compression, and total duration which is well beyond 1 hour in sequential cyclic compression applications that were run in this work are relevant in this context as we have also seen significant differences in deformation of cellular structures before and after compression or between control (non-compressed) and compressed ovarian cancer cells (similar to that we have seen significant differences in cancer cell viability, as well, in these compression rate and durations in Onal et al. 2021 Frontiers in Physics). Indeed, the damage that we observed in cells was related more to the magnitude of the applied stress not the rate. This is presented in our results throughout the manuscript. Cells at milder applied pressures (e.g 15.6 kPa) have cellular deformations but not damage. When applied pressures were increased to Intermediate 2 (37.8–41.7 kPa) and Severe (127.8–140 kPa) piston contact pressures, cells started to get damage and those results were as expected based on both in vitro compression devices and in vivo applied pressures reported in literature. For instance, compressive stresses up to 120 mmHg (~16 kPa) can be exerted in tumor microenvironment and would be sufficient to locally collapse tumor blood vessels in vivo which commonly happens in tumors (tumor microvascular pressures are reported to range from 6 to 17 kPa), as mentioned in study results by Helmlinger et al. 1997 Nature. Compressive stresses beyond these are also expected to happen in tumor microenvironments, and tumor cells could be resistant to those as mentioned in the Introduction of our manuscript:

“Compressive stress is estimated to reach 18.9 kPa for human tumors and can exceed 20 kPa based on the experimental data from murine tumors, while those values can be even higher in situ with the impact of the surrounding extracellular matrix in tumor microenvironment [3,8].”

Original comment 11) Further justification should be provided for the cyclic stress schedule chosen. Musculo-skeletal tissues would be expected to experience such forces, but this is not clear for ovarian cancer.

Authors' response: The cyclic stress schedule (i.e. 5-min compressed stage followed by 5-min rest stage in each cycle) was chosen to allow cells enough time during each stage to complete bulge formation or recovery, processes which were considered important for the investigation of cell deformation and recovery after compression. Membrane bulge formation and recovery processes are clearly observable in S1 Movie. This was explained in the following paragraph in section “Comparison of the experimental and computational results in micro-piston device”

“While inferring these results, the cell compression process was imaged in detail in Matrigel-coated microchannels at a speed of 1 frame per 100 ms, as shown in S1 Movie. As illustrated by this time-lapse video, the platform allowed for the temporal evolution of the dynamic cell compression to be investigated in each cycle while the piston was contacting cells and applying compression. Distinct cell membrane bulges could be observed to form during such compression at piston contact pressure of 15.6 kPa and recovery of cell membranes from these bulges was visible when pistons were lifted off during rest stages. Although the platform is capable of temporal control of cell compression (e.g. to decrease, increase or remove the pressure at any certain time interval), duration of each stage in a cycle was set to 5 minutes, corresponding to a 5-min compressed stage followed by 5-min rest stage in each cycle. This was chosen to allow cells enough time during each stage to complete bulge formation or recovery, processes which were considered important for the investigation of cell deformation and recovery after compression. Sensor readings in S1 Movie showed that the micro-piston actuated according to externally applied pressures with reliable control and could be operated in short durations gradually or suddenly when needed.”

Further explanation and discussion have been also added in this section as in the following paragraph:

“Cyclic compression in the study of Ho et al. was applied only for 6 minutes in total [2], significantly shorter than the entire duration of a cyclic compression in this work, and results presented here suggest that cell deformation at compression stage and recovery at rest stage depends on the temporal length of the compression on cells. Particularly observable in S1 Movie, cell membrane bulge formation slowly developed during the compression stage while cells were responding to contact pressure at the interface with the PDMS piston. This was despite the fact that the piston was actuated onto the cells rapidly during cyclic compression as directed by the external pressure controller. As shown by the pressure sensor readings in the same movie, PDMS membrane and piston were consistently responding to amount, duration and mode of the applied pressure. Thus, the gradual deformation of cells and bulge formation during compression stages must originate in the response type of the cells to mechanical load when in solid contact with the PDMS piston. During rest stages, where the compression was lifted, cell bulges did not fully recover right after compression was removed, suggesting that the cells require a certain amount of time to recover compressed cellular parts (S1 Movie). Based on these observations of temporal evolution of the cell compression application, and comparison of application durations in this study with the study by Ho et al. [2], we hypothesize that this recovery also depends on the duration and number of the cycles.”

Regarding sources of compression ovarian cancer cells are exposed to, further information has been added in Introduction:

“While some cells experience permanent plastic deformation after a repetitive mechanical tensile loading and unloading, the impact of such repetitive compression on deformation of cells remains largely unknown [1,2]. As such, the ability to apply cyclic compression is crucial for any experimental setup aimed at the study of mechanical compression taking place in cell and tissue microenvironments [3–5]. For instance, in ovarian cancer, cells are exposed to chronic compressive stress from different sources such as tumor growth, displacement within stromal tissue and hydrostatic pressure out of ascitic fluid in peritoneal cavity [3,6]. Ascites-induced compression exists in the peritoneal microenvironment due to the ascites volume that can reach >2L and increases ovarian cancer cell adhesion to peritoneum, shown by Asem et al. using in vivo assay [7]. This peritoneal microenvironment is continuously affected by the movements of the surrounding organs, musculoskeletal dynamics, and gravity [6]. Thus, ovarian cancer cells might experience cyclic compression profiles during chronic exposure to compression from different sources and due to anatomical location of the ovary. Mimicking such compression profiles and physiological pressure values in an in vitro dynamic compression system is necessary to further study impact of compressive stress in cancer.

Compressive stress is estimated to reach 18.9 kPa for human tumors and can exceed 20 kPa based on the experimental data from murine tumors, while those values can be even higher in situ with the impact of the surrounding extracellular matrix in tumor microenvironment [3,8]. Compared to other cancer types such as breast and metastatic bone niches, compression studies have been limited for ovarian cancer, including for investigating the impact of various amounts of applied pressures [6]. Pressure values existing in literature that are applied for ovarian cancer have been in a range from ∼3 kPa to 6.5 kPa [3,7,9]. Ovarian cancer cell response at higher applied pressures that can occur in the body need further investigation.”

Furthermore, it was noted in our manuscript that “While compression frequencies in an in vivo tumor microenvironment are not yet fully known, cyclic compression is important for maintenance of cell culture during mechanical compressions in an in vitro experimental setup, in particular as it enhances media flow underneath the micro-pistons.”

Follow up comment: If the study is meant to investigate ovarian cancer, it should be designed to mimic the in vivo mechanical environment. The authors should either provide evidence that ovarian cancer cells are exposed to cyclic stresses in vivo, or use their device to produce different (longer term) stress application. As presented, it is not clear whether the results are an artefact of the cyclic forces.

For ovarian tumours and in progression to cancer, compression is sourced by tumour growth, so it is on inner cells in a tumour, the counteracting compression by stromal tissue and hydrostatic pressure from ascitic fluid and it is affected by the other body movements due to the anatomical region of ovarian tissue.

Although we don’t have a substantial data showing cyclic compression profile in vivo in ovarian cancer patients, it would be hard to ignore the impact of the movements of the surrounding organs such as gastrointestinal movements (Rietveld, 2018, Supportive Care in Cancer), contractions and musculoskeletal dynamics that can take place in a cyclic form, which can alter the hydrostatic pressure from ascitic fluid.

When it comes to ascitic fluid and ascites-induced compression, this exists in ovarian cancer patients as it was also mentioned by Klymenko et al. 2018 Disease Models & Mechanisms, Bregenzer et al. 2019 Cancers (review article), Asem et al. 2020 Scientific reports, Novak et al. 2020 Cancers, and others. 

Klymenko et al. 2018 Disease Models & Mechanisms:

“Ascites-induced increases in intraperitoneal pressure have negative implications at the clinical level. Development of tense ascites, usually associated with stage III or IV disease, is associated with severe discomfort, poor prognosis and fatality in women with EOC.”

Their model systems in bulk settings, as well as the platforms by Asem et al. 2020 (bulk, ex vivo and in vivo) and Novak et al. 2020 (bioreactor at millimetre scale), have also intended to provide in vitro platforms, in comparison to microfluidic platform and method we have developed in our work, to study future mechanistic analyses of compression-induced mechanotransduction related to enhanced intraperitoneal dissemination.

We have also queried this with researchers in the Department of Obstetrics and Gynaecology, University of Otago, New Zealand, who have expertise in ovarian cancer. Here is their information they have shared with us:

“Advanced ovarian cancer has a large volume of ascitic fluid, which has high viscosity relative to water. Increased density resulting from protein components in the ascitic fluid could mimic the mechanical pressure and influence tumour cells that grow on the surface of the peritoneal membrane. The fluid could put pressure on tumour cells to change the protein cytoskeleton and reshape cell morphology and cell signalling, leading to cell survival, growth, and drug resistance.”

Klymenko et al. also mentions in their paper that ascites includes bioactive lipids, growth factors and cytokines and the role of ascites-induced changes in peritoneal mechanobiology needs to be examined.

Apart from such in vivo patient-related information about ascites-induced compression which can be impacted by the cyclic movements and pressures of the surrounding organs, that is transmitted to peritoneal microenvironment, 

particularly for cyclic compression profiles we insist on that this is a more realistic way of exerting compressive forces on cells whether for short or long term in an in vitro experimental setup.

Common examples for the in vitro systems that used cyclic compression at different scales:

- Takao et al. 2019 Biology Open - bulk cyclic compression device applied cyclic pressures on breast cancer cells

- Novak et al. 2020 Cancers - 3D bioreactor at millimetre scale, applied cyclic compressive forces and compared to static compression, applied on ovarian cancer models

- Ho et al. 2018 Front. Bioeng. Biotech. - microfluidic single cell compression device at micrometre scale, applied cyclic compressive forces on single normal breast epithelial cells

Briefly exemplified here, these cyclic compression applications were detailed in Introduction of our manuscript.

For more detailed information, we have introduced compressive force types and further reviewed cyclic compression profiles in our upcoming (accepted) review article by Onal et al. 2022 iScience. 

Thus, unlike bulk systems which have extensive access to cell media / nutrients and a continuous compression on cells can be kept for long term which we call static compression,

when it comes to using microfluidics to study the impact of compression on cells in a microchannel (mechanical compression at micro scale), we need to do that in a cyclic manner to allow the cells to reach nutrients efficiently and so to eliminate the impact of nutrient deprivation which would happen if we had kept the compression for long term with no rest(uncompressed) cycles in between, as it was mentioned in Introduction of our manuscript:

“While compression frequencies in an in vivo tumor microenvironment are not yet fully known, cyclic compression is important for maintenance of cell culture during mechanical compressions in an in vitro experimental setup, in particular as it enhances media flow underneath the micro-pistons.”

Reviewer #2: The authors revised the introduction and others places in the manuscript to include additional background comparing similar studies and their compression parameters with this study, and further clarifying the novelty of this study based on a previously developed device. Sufficient information is now provided to clarify Figure 2 and the corresponding results. However, some of the results require further clarification.

- Page 8, line 285: It is not clear what is the significance and relevance of cell membrane bulge formation under compression. How it is related to actin redistribution and existing literature should be discussed.

Here, the reviewer touches upon an important topic which will be focus of another piece of work. We think that cell membrane where bulge formation can happen under stress (here we expect that that bulge formation would also affect cell extensions/edges which are actin-based) and redistribution of the actin cytoskeleton would be important to investigate cell adhesion and migration(locomotion), integration of cadherin adhesion and cytoskeleton at adherens junctions at cell-cell contacts and interactions between different cell types such as ovarian cancer cells and mesothelial cells. New references (Kalli et al. 2022 Molecular Cancer Research; Mege et al. 2017 Cold Spring Harbor Perspectives in Biology; Heisenberg et al. 2013 Cell) and a comment have been added for the associated part as below:

“Changes in cell membrane integrity and redistribution of the actin cytoskeleton under stress could affect cell adhesion and migration, integration of cadherin adhesion and cytoskeleton at adherens junctions at cell-cell contacts [27–29] and interactions between different cell types such as ovarian cancer cells and mesothelial cells [7].”

Page 9, line 322-326 in the revised manuscript.

- Page 10, line 324- 359: this section is mainly describing parameters and process of the cyclic compression, therefore would better fit in Methods section.

As per the reviewer’s suggestion, the relevant section “The actin-GFP transduced cancer cells were dynamically stimulated and compressed with various intermediate actuation pressures in a sequential cyclic manner,…………..……Thus, cells could have the time to adjust during the pressure increase and show their intrinsic response to the pressurized stage of each compression level.” has been moved to the Methods section “Live imaging of GFP-tagged actin cytoskeleton of cells”.

Page 6-7, line 207-241 in the revised manuscript.

The last sentence in the current version of the first paragraph of the Results and discussion section “Live cell actin profiles of cancer cells during sequential cyclic compression” has been slightly revised to “Specifically, the actin cytoskeleton profile was quantified while cells were in compressed and rest stages (Fig 3(b-c) and Fig 4).”

Page 10-11, line 362-363 in the revised manuscript.

- Page 10, line 333-334, “similar as to the sequence shown in fig 1(d)…”: Why not use the exact sequence for generating results in fig 3 in fig 1(d) instead of a similar sequence?

Since the results in Fig 3 and Fig 4 are out of at least three independent experiments, it was meant that the sequence in Fig 1(d) is representative. It has been thus revised from “Similar as to the sequence shown in Fig 1(d)….” to “As shown by the representative sequence in Fig 1(d)….”.

Page 6, line 215-216 in the revised manuscript.

- Figure 3: What is the total imaging time throughout compression? Is photobleaching of GFP a concern?

Per one sequential cyclic compression application on cell actin-GFP, as shown by the events/stages on the x-axis of the graph in Fig 4, namely from first stage ‘Before compression start (0 kPa)’ to the final stage ‘At rest’, 9 actin-GFP frames were taken. So, the total imaging itself was for 9 frames, 2 seconds exposure per frame. However, the entire of sequential cyclic compression application itself took the total time shown on the x-axis of the graph in Fig 1(d), ~8000 seconds = ~2.2 hours. 

The calculated actin signals were compared between the region under piston and control region around the piston. Actin signal in the control regions were also compared to each other across the cycles and remained statistically insignificant (S1 File) as it was indicated in the manuscript in Results and discussion in section “Live cell actin profiles of cancer cells during sequential cyclic compression”: “The change in actin signal for compressed cells under micro-piston was statistically different compared to non-compressed cells in control regions around the micro-piston at all applied compression levels (p <0.05, see S1 File). Actin signal changes in the control regions on the other hand, remained statistically insignificant (p >0.05).”

Page 12, line 369-373 in the revised manuscript.

The following discussion was added in the same paragraph: “Thus, the GFP tagged actin signal in control regions was robust during the entire process, as expected [30]. This can be reference to that fluorescence changes in the region under micro-piston (i.e. decrease in actin signal) was due to the impact of mechanical compression.” 

Page 12, line 373-376 in the revised manuscript.

Thus, given that there was no statistically significant difference between the control groups throughout the application, we do not have signs indicating for photobleaching. 

- Page 11, line 411-412: The authors should be careful and more specific when describing observed changes in actin during cyclic compression. Deformation/damage implies structural changes of actin. It is not clear whether the authors observed structural changes in actin or simply redistribution of actin inside a cell. If actin is deformed, how a lower intensity of GFP corresponds to actin deformation should be explained.

Indeed “deformation” here has meant remodelling of the actin inside a cell, similar to the term used at “nuclei deformation” or “cell deformation”. When we say a cell or cellular structure is deformed, we don’t necessarily mean that the cell or structure is damaged. It could be interpreted as a change in shape but not necessarily damaged shape. Compression would induce a change on cells and cellular structures, such as actin cytoskeleton and nuclei, at different levels from shape deformation to damage, depending on the strength of the applied stress. Thus, the relevant sentence has been revised as in the following:

“Thus, the cell actin was deformed and damaged only partially in the compressed part depending on the magnitude of the applied pressure, whereas it remained more intact in the non-compressed part.”

Revised to 

“Thus, the cell actin was either deformed or damaged in the compressed part depending on the magnitude of the applied pressure, whereas it retained more of its original shape in the non-compressed part.”

Page 13, line 417-419 in the revised manuscript.

The next sentence has been also slightly revised:

“This functionality which is unique to our platform by having control and compressed groups in the same microdevice, is useful to mimic partial cellular deformation and damage that can occur in vivo in presence of a localized mechanical loading.”

Revised to 

“This functionality which is unique to our platform by having control and compressed groups in the same microdevice, is useful to mimic partial cellular deformation or damage that can occur in vivo in presence of a localized mechanical loading.”

Page 13, line 420-422 in the revised manuscript.

- Page 11, line 376-378: Given that cell has a certain height and upon compression, part of the cell that was not in the focal plane can be compressed to the focal plane and add to the actin signal. The authors should clarify whether this was observed (maybe during one cycle of quick compression) and was an issue for the quantification. Similarly, are compressed cells and control cells in the same plane when compressed?

In the relevant sentence while we have written it as “the focus was kept on one focal plane on the glass surface where cells were initially adhered to” we did not mean exactly the glass surface itself. Before we started the sequential cyclic compression process and time-lapse imaging, we tried to have cells as many as possible in a certain focal plane (into focus) to the extent that wide-field imaging technique allows. The focal plane of cells is slightly upper than the glass surface where they are adhered to, so this was where we had the focus of most of the cells as a compromise between the very top of an adhered cell height (e.g. on top of nucleus region of a cell or a slightly more rounded cell) and the very bottom (e.g. cell extensions/edges adhered to the glass surface). Usually, unless there is a temperature change on the sample, cells stay in focus throughout the time-lapse imaging process. We have considered this and before we started the compression and imaging process, we allow the cells to come to an equilibrium in temperature on the microscope stage for 20-30 minutes. 

To prevent any confusion whether the focus was on the glass surface itself or the cells themselves, the relevant sentence has been revised:

“During the entire process, wide-field imaging was used and the focus was kept on one focal plane on the glass surface where cells were initially adhered to.”

Revised to 

“During the entire process, wide-field imaging was used and the focus was kept on one focal plane on the cells adhered to the glass surface.”

Page 12, line 380-382 in the revised manuscript.

Compressed cells and control(uncompressed) cells were in the same focal plane when compressed.

Regarding technical side of imaging cells in a dynamic compression application, an experiment by 3D imaging of the cells (taking cell height into consideration) could be useful, but also there are limitations even with the confocal imaging technique, considering how fast cells are able to react under dynamic compression and then at rest stages (once applied pressure is lifted). Related to this, we noted the following in our manuscript with a good number of references, including our own previous report, who tried confocal imaging in their compression assays and found vertical and temporal resolution and long image acquisition times required for 3D visualization are limiting factors to use confocal imaging compared to wide-field imaging:

“………. Measurement of the viscoelastic relaxation of the cells is difficult however, especially when vertical and temporal resolution in imaging are limited, and long image acquisition times required for 3D visualization might prevent capturing the changes in cell height at the time of compression. ………….. wide-field imaging is considered fast enough (e.g. milliseconds long) to capture cell states during the steps of the compression process compared to real-time volumetric imaging (e.g. minutes long) using confocal microscopy [2,5,33–36].”

Page 16, line 520-527 in the revised manuscript.

- Page 11, line 379-382: it is not clear why the focal plane needs to be at the piston level (above the glass) at rest stages, as cells were cultured on the glass slide.

The section including the relevant sentence has been trying to indicate that we did not bring the focal plane to the piston level (non-compressed form, above the glass) at rest stages: 

“During the entire process, wide-field imaging was used and the focus was kept on one focal plane on the glass surface where cells were initially adhered to. At compressed stages, glass and PDMS piston surfaces were in contact and at the same focal plane. At the rest stages when pressure is released and piston is retracted back, to be able to gain the signal from piston surface there would be need to bring the focal plane to the suspended piston level that is on average 108 �m above the glass surface. In the region under the micro-piston, there seem to be variations in gained actin signal between the compressed and rest stages of each cycle (Fig 4), but these were not statistically significant (p>0.05, see S1 File). Thus, no significant amount of cell artifacts was attached to piston surface.”

The relevant sentence has been revised and developed for further clarification:

“At the rest stages when pressure is released and piston is retracted back, to be able to gain the signal from piston surface there would be need to bring the focal plane to the suspended piston level that is on average 108 �m above the glass surface, which we did not employ in this application. Thus, we retained the same focal plane and wide-field imaging condition across all stages of the sequential compression on GFP-tagged actin of cells.”

one focal plane on the cells adhered to the glass surface.”

Page 12-13, line 383-388 in the revised manuscript.

Mentioned as response to a previous question, the first sentence has been also revised:

“During the entire process, wide-field imaging was used and the focus was kept on one focal plane on the cells adhered to the glass surface.”

Page 12, line 380-382 in the revised manuscript.

- Page 11, line 386-395: Given that Ho et al used a much higher pressure compared to this study but observed actin formation instead of disruption, how much of the continuous decrease of actin signal in this study is due to insufficient rest time? Have the authors tested how long it would take cells to recover actin levels post one cycle of compression?

Ho et al. 2018 Frontiers Bioeng Biotech applied 6-min cyclic compression though control valve alternating between external pressures of 10 and 15 psi (68.9 and 103.4 kPa) at 0.5 Hz on single MCF10A normal breast epithelial cells and observed no significant change in cell height between before and after 6-min compression. In other words, these are the pressures applied externally to operate the microfluidic device. In their study Ho et al. do not specify the actual (contact) pressure cells are experiencing inside microchannels under the blocks, which is needed for a complete comparison to our piston contact pressures and the impact on cells. 10 and 15 psi (68.9 and 103.4 kPa) Ho et al. applied externally is not what cells are experiencing as contact (internal) pressures inside microchannels. Epithelial cells would hardly stay intact/undamaged at such high pressures. Thus, compressive stress transmission onto cells must be less than their externally applied pressures. 

Additionally, Ho et al. in their paper (2018 Frontiers Bioeng Biotech) mention that they have observed a limited number of cells in their single cell compression platform (in their supplementary file, there is an image of thickening of a pre-existing actin structure shown on one cell). Our study results which are an average of multiple cells out of at least three independent experiments of sequential cyclic compression application with actin-GFP are in comparison to this and they were observed for a wide range of contact (internal) pressures from Mild to Intermediate to Severe pressures (Fig 3 and 4), as it was noted in the associated part in our manuscript:

“As such, these results expand on the work by Ho et al., who used a more limited pressure and frequency range, and observed formation of new actin stress fibers and thickening of a pre-existing actin structure in a very limited number of cells [2].”

Page 13, line 396-399 in the revised manuscript.

In our study, as shown in the representative graph in Fig 1(d), there are rest stages after the compression stages were applied and we were taking the actin-GFP frame within these resting times while the cells are at rest stages. While removing the pressure from the cells (retracting the piston to its initial position) some recovery of the cells was starting. After the pressure is totally brought to zero, while there is no pressure on cells, we were waiting at least 2 minutes for the cells to recover more, and then initiating the fluorescence imaging at the wavelength and exposure setting for GFP (2 seconds per frame). We have done that consistently for the rest stages after each cycle of compression at the indicated pressure amount shown in Fig 3 and 4. 

We have thought that waiting at least 2 minutes after the pressurized stage, before any GFP image was taken for the rest stage, would be somewhat enough for the cells to recover but technically it is also possible to wait more times with our sequential cyclic compression and time-lapse imaging method presented in this manuscript. The developed device and method are as flexible as it could allow for a wide range of possibilities of durations for compression and/or rest stages and for imaging. 

Furthermore, in our end point staining assays of cell actin of the cyclically compressed cells at mild (15.6 kPa, for >1 hour in cyclic manner) and upper mild (20.8 kPa, for >1 hour in cyclic manner) pressures, we also see thickening of actin structures at the edges of multiple cells. We also see loss of actin structures in some cells exposed to upper mild (20.8 kPa, for >1 hour in cyclic manner) pressures as Phalloidin did not bind well to those cells while a good Hoechst signal was gained on nuclei of the same cells, shown in Figure 6(a) at zero time after compression. In the recovery process after compression, in these set of experiments, we waited 24 hours for cells to recover (keeping cells in cell culture conditions without applied pressure), after cyclic compression application was ended. The results of this are also shown in Fig 6(a) for the recovered group (at 24 h-recovery after compression).

- Page 13, line 423-426: It would be helpful to have a zoomed in image for control cells as well, so the difference of actin distribution could be observed more directly. Additionally, in Fig 5(a) compressed cells seem to have stronger actin compared to control cells? Did actin level restore in compressed cells after 24 hours or just nucleus morphology?

Zoomed in image for control cells was given in S5 Fig.

This figure has been referenced in the associated sentence:

“Fig 5(a) shows the distinct actin deformation by the increased fluorescence signals at the edges of the cells in the compressed groups under the micro-piston at zero time after compression, compared to the control (non-compressed) groups around the micro-piston (S5 Fig).”

Page 13, line 429-432 in the revised manuscript.

The quantitation demonstrated in Fig 5(b-d) is for nucleus morphology only, as nucleus has more defined shape making image analysis more convenient/standardized on a good number of cells that were interacting and forming a denser culture on Matrigel coating under a 300 mm x 300 um piston in this platform. In such a culture distinguishing actin structure for each cell would be harder and in these set of experiments cells were not transduced with actin-GFP BacMam constructs. Instead, the results were from end point staining assays, with Phalloidin staining on fixed cells at the indicated time points. For recovery of actin at a mild applied pressure amount such as 15.6 kPa, fluorescent tracking of actin of compressed cells during 24 h-recovery would give more information for quantitation of the actin structures.

However, we can say that distinct actin deformation at the edges of the cells in the compressed groups under the micro-piston at zero time after compression were no longer visible on the 24 h-recovered compressed groups in Fig 5(a). Additionally, we can say that at 24 h-recovery after compression cell actin deformation seems recovered in the group under the micro-piston as appeared similar to the control group. We mentioned these observations and slightly revised by adding in-text reference of the figures in the associated part following the sentence that has been just revised above:

“………. Thus, membrane bulges formed during cell compression at 15.6 kPa as observed in S1 Movie and fixed directly after compression appeared as increased actin signal at cell edges (Fig 5(a)). Conversely, at 24 h-recovery after compression, cell actin deformation seems recovered in the group under the micro-piston as appeared similar to the control group (Fig 5(a) and S5 Fig).”

Page 13-14, line 432-436 in the revised manuscript.

- Page 15, line 475-477: based on figure 6 it seems like the actin profile was altered after 24h recovery compared to compressed cells? It would be helpful to point out cells that did not recover in the image.

Arrows (yellow) have been added onto Figure 6.

Caption of Fig 6(a) has been revised:

“Representative arrows (white) show distinct actin deformation in form of increased fluorescence signals at the edges of the cells in the compressed groups under the micro-piston at zero time after compression.”

Revised to

“Representative white arrows show distinct actin deformation in form of increased fluorescence signals at the edges of the cells in the compressed groups under the micro-piston at zero time after compression, while yellow arrows indicate no full recovery and thus an altered actin profile of cells at 24 h-recovery after compression.”

- Page 17, line 496-498: specifying the difference, i.e., decrease/increase, would help improve the readability.

The associated sentence has been revised to improve the readability:

“On the other hand, circularity of the compressed and 24 h-recovered compressed cells was statistically different both between each and respective control groups (Fig 6(c)).”

Revised to 

“On the other hand, circularity of the compressed and 24 h-recovered compressed cells statistically significantly decreased compared to their respective control groups (Fig 6(c)). Circularity of the 24 h-recovered compressed cells was also statistically less than the compressed group at zero-time after compression (Fig 6(c)).”

Page 16, line 502-505 in the revised manuscript.

---

## [Decision Letter · Decision Letter 2]

23 Nov 2022

PONE-D-22-08675R2Application of sequential cyclic compression on cancer cells in a flexible microdevicePLOS ONE

Dear Dr. Onal,

Thank you for submitting your manuscript to PLOS ONE. After careful consideration, we feel that it has merit but does not fully meet PLOS ONE’s publication criteria as it currently stands. Therefore, we invite you to submit a revised version of the manuscript that addresses the points raised during the review process. Please, add a potential explanation on the reduction in GFP (CTCF) signal as requested by reviewer 2.

We look forward to receiving your revised manuscript.

Kind regards,

Antonio Riveiro Rodríguez, PhD

Academic Editor

PLOS ONE

Journal Requirements:

Reviewers' comments:

Reviewer's Responses to Questions

**Comments to the Author**

1. If the authors have adequately addressed your comments raised in a previous round of review and you feel that this manuscript is now acceptable for publication, you may indicate that here to bypass the “Comments to the Author” section, enter your conflict of interest statement in the “Confidential to Editor” section, and submit your "Accept" recommendation.

Reviewer #1: All comments have been addressed

Reviewer #2: (No Response)

2. Is the manuscript technically sound, and do the data support the conclusions?

Reviewer #1: (No Response)

Reviewer #2: Partly

3. Has the statistical analysis been performed appropriately and rigorously? 

Reviewer #1: (No Response)

Reviewer #2: I Don't Know

4. Have the authors made all data underlying the findings in their manuscript fully available?

Reviewer #1: (No Response)

Reviewer #2: Yes

5. Is the manuscript presented in an intelligible fashion and written in standard English?

Reviewer #1: (No Response)

Reviewer #2: (No Response)

6. Review Comments to the Author

Reviewer #1: (No Response)

Reviewer #2: I have the following comment regarding changes in GFP signal:

Page 10, line 367-368: It is still not clear what a reduction in GFP (CTCF) signal means biologically, whether it is due to changes in actin production or actin deformation, or something else.

Apart from this, the reviewer has no further comments and believes the manuscript is acceptable for publication.

7. PLOS authors have the option to publish the peer review history of their article (what does this mean?). If published, this will include your full peer review and any attached files.

Reviewer #1: No

Reviewer #2: No

---

## [Author Response · Author response to Decision Letter 2]

30 Nov 2022

Journal Requirements:

We have reviewed our reference list. 

Journal names in references Fabry et al. 2001 Physical Review Letters and Fu et al. 2012 Lab on a Chip have been slightly edited.

The reference Fitzpatrick M. Measuring cell fluorescence using ImageJ. The Open Lab Book; 2014 has been also improved for its BibTeX type. 

Reviewers' comments:

6. Review Comments to the Author

Reviewer #1: (No Response)

Reviewer #2: I have the following comment regarding changes in GFP signal:

Page 10, line 367-368: It is still not clear what a reduction in GFP (CTCF) signal means biologically, whether it is due to changes in actin production or actin deformation, or something else.

Apart from this, the reviewer has no further comments and believes the manuscript is acceptable for publication.

The relevant sentence has been revised and developed as below:

“…… As such, the results illustrate how the actin cytoskeleton of cancer cells changes in response to compressive stress under live-cell conditions.”

Revised to

“…… As such, the results illustrate how the actin cytoskeleton of cancer cells changes in response to compressive stress under live-cell conditions, from deformation to disruption, depending on the strength of compression (also see S2 Movie). The actin cytoskeleton of cells, after being flattened under the micro-piston during compression stages (Fig 3(c)), was undergoing a process of recovery during rest stages (Fig 3(b)). Throughout a cycle of compression and rest stages, actin structures rearranged themselves and a few new extensions could be observed forming, for instance, during the first cycle of applied mild pressures (15.6-15.9 kPa), while disruption and ruptures became apparent during cycles of higher pressures such as Intermediate 2 (37.8–41.7 kPa) and Severe (127.8–140 kPa) levels (see S2 Movie and Fig 3(b-c)). As such, Fig 3(b) clearly exemplifies cell actin changes from deformation to disruption over the indicated pressure ranges.”

Page 10, line 367-378 in the revised manuscript.

Arrows have been added in Fig 3(b) to show cell actin changes from deformation to disruption depending on the strength of compression. 

Fig 3(b) caption has been also updated as below:

“…… Yellow arrows indicate example cells exhibiting rearrangement of actin structures with formation of new extensions, blue arrows for actin deformation with rearrangement of existing actin structures, green arrows for actin disruption with raptures and pink arrows for complete actin disruption.”

Thus, in our sequential cyclic compression method, the interpretation of the reduction in GFP (CTCF) signal depends on the applied pressure range and how significant the reduction in signal is. Measuring those signals, for instance, we found that “CTCF during the rest stage after upper mild compression at Intermediate 2 level was statistically significantly different from the stage before compression (p=0.0229) and the rest stage after first cycle of Mild compression (p=0.0352), which indicates that Intermediate 2 level caused a disruption of cell actin. This disruption was significantly higher again at Severe level compared to other levels (p <0.05, see S1 File)”, as it was indicated in text in the relevant section and in Fig 4. 

The part in the following of the relevant revised/developed sentence which reviewer pointed out further detailed and discussed those results by the fluorescently measured actin signals and this part has been split into a new paragraph, starting with a small revision “In Fig 4, ……”. Next paragraph provided biological interpretation comparing to other studies. A new sentence “Such a plastic response could originate from the cytoskeletal bond ruptures, as suggested by Bonakdar et al. who applied mechanical tensile loading on cells [1].” has been added in this paragraph as shown below:

“In Fig 4, the change in actin signal for compressed cells under micro-piston was statistically different compared to non-compressed cells in control regions around the micro-piston at all applied compression levels (p <0.05, see S1 File). Actin signal changes in the control regions on the other hand, remained statistically insignificant (p >0.05). Thus, the GFP-tagged actin signal in control regions was robust during the entire process, as expected [30]. This can be reference to that fluorescence changes in the region under micro-piston (i.e. decrease in actin signal) was due to the impact of mechanical compression. At the higher pressures of Intermediate 2 and Severe, actin of most of the cells was disrupted compared to in the initial stages of Mild compression (p <0.05, see S1 File). Interestingly, a few cells on the glass surface usually remained resistant to the increasing pressure ranges, as evidenced by their retained fluorescent signal while compressed under the micro-piston. During the entire process, wide-field imaging was used and the focus was kept on one focal plane on the cells adhered to the glass surface. At compressed stages, glass and PDMS piston surfaces were in contact and at the same focal plane. At the rest stages when pressure is released and piston is retracted back, to be able to gain the signal from piston surface there would be need to bring the focal plane to the suspended piston level that is on average 108 μm above the glass surface, which we did not employ in this application. Thus, we retained the same focal plane and wide-field imaging condition across all stages of the sequential compression on GFP-tagged actin of cells. In the region under the micro-piston, there seem to be variations in gained actin signal between the compressed and rest stages of each cycle (Fig 4), but these were not statistically significant (p >0.05, see S1 File). Thus, no significant amount of cell artifacts was attached to piston surface. 

CTCF during the rest stage after upper mild compression at Intermediate 2 level was statistically significantly different from the stage before compression (p=0.0229) and the rest stage after first cycle of Mild compression (p=0.0352), which indicates that Intermediate 2 level caused a disruption of cell actin. This disruption was significantly higher again at Severe level compared to other levels (p <0.05, see S1 File). As such, these results expand on the work by Ho et al., who used a more limited pressure and frequency range, and observed formation of new actin stress fibers and thickening of a pre-existing actin structure in a very limited number of cells [2]. In contrast, the current work observed live-cell actin in the cytoskeleton from deformation to ruptures at a wider range of applied pressures across several stages of the sequential cyclic compression. The observed disruption of actin cytoskeleton in these experiments indicates for cell plasticity during a cyclic compression in ovarian cancer when varying higher pressures occur in the microenvironment. Such a plastic response could originate from the cytoskeletal bond ruptures, as suggested by Bonakdar et al. who applied mechanical tensile loading on cells [1]. There is need for more studies in literature to compare our results on cell plasticity after cyclic or dynamic compression applied sequentially up to high pressures. For cell actin deformation at milder pressures, however, a distinct change in actin cytoskeleton morphology was observed by Asem et al. for mesothelial cells, interacting with ovarian cancer cells, under compression that is statically applied at ∼3 kPa for 24 h in a bulk compression system. As a result, the interaction between ovarian cancer cells and mesothelial cells by enhanced formation of actin-based tunneling nanotubes (TNTs), and hence metastatic progression in ovarian cancer, was promoted under compression [7].

Page 11, line 379-424 in the revised manuscript, revisions in line 379 and 414-416.

Fig 3 has been edited to include arrows on the images in the panel (b). 

Fig 5 and Fig 6 have been slightly adjusted for matching figure dimensions. 

All main figures have been checked through PACE tool. PACE corrected figure files have been uploaded to the EM submission system.

---

## [Decision Letter · Decision Letter 3]

19 Dec 2022

Application of sequential cyclic compression on cancer cells in a flexible microdevice

PONE-D-22-08675R3

Dear Dr. Onal,

We’re pleased to inform you that your manuscript has been judged scientifically suitable for publication and will be formally accepted for publication once it meets all outstanding technical requirements.

Kind regards,

Antonio Riveiro Rodríguez, PhD

Academic Editor

PLOS ONE

Reviewers' comments:

Reviewer's Responses to Questions

**Comments to the Author**

1. If the authors have adequately addressed your comments raised in a previous round of review and you feel that this manuscript is now acceptable for publication, you may indicate that here to bypass the “Comments to the Author” section, enter your conflict of interest statement in the “Confidential to Editor” section, and submit your "Accept" recommendation.

Reviewer #2: All comments have been addressed

2. Is the manuscript technically sound, and do the data support the conclusions?

Reviewer #2: Yes

3. Has the statistical analysis been performed appropriately and rigorously? 

Reviewer #2: Yes

4. Have the authors made all data underlying the findings in their manuscript fully available?

Reviewer #2: Yes

5. Is the manuscript presented in an intelligible fashion and written in standard English?

Reviewer #2: Yes

6. Review Comments to the Author

Reviewer #2: The authors have addressed my comments satisfactorily, and this manuscript is now acceptable for publication.

7. PLOS authors have the option to publish the peer review history of their article (what does this mean?). If published, this will include your full peer review and any attached files.

Reviewer #2: No

---

## [Editor Report · Acceptance letter]

26 Dec 2022

PONE-D-22-08675R3 

Application of sequential cyclic compression on cancer cells in a flexible microdevice 

Dear Dr. Onal:

I'm pleased to inform you that your manuscript has been deemed suitable for publication in PLOS ONE. Congratulations! Your manuscript is now with our production department. 

Kind regards, 

on behalf of

Dr. Antonio Riveiro Rodríguez 

Academic Editor

PLOS ONE